# Historical and future changes in air pollutants from CMIP6 models

Steven T. Turnock[1], Robert J. Allen[2], Martin Andrews[1], Susanne E. Bauer[3,4], Makoto Deushi[5], Louisa Emmons[6], Peter Good[1], Larry Horowitz[7], Jasmin G. John[7], Martine Michou[8], Pierre Nabat[8], Vaishali Naik[7], David Neubauer[9], Fiona M. O'Connor[1], Dirk Olivié[10], Naga Oshima[5], Michael Schulz[10], Alistair Sellar[1], Sungbo Shim[11], Toshihiko Takemura[12], Simone Tilmes[6], Kostas Tsigaridis[3,4], Tongwen Wu[13], Jie Zhang[13]

[1]Met Office Hadley Centre, Exeter, UK
[2]Department of Earth and Planetary Sciences, University of California Riverside, Riverside, California, USA
[3]Center for Climate Systems Research, Columbia University, New York, NY, USA
[4]NASA Goddard Institute for Space Studies, New York, NY, USA
[5] Meteorological Research Institute, Tsukuba, Japan
[6]Atmospheric Chemistry Observations and Modelling Lab, National Center for Atmospheric Research, Boulder, CO, USA
[7]NOAA Geophysical Fluid Dynamics Laboratory, Princeton, USA
[8]Centre National de Recherches Météorologiques (CNRM), Université de Toulouse, Météo-France, CNRS, Toulouse, France
[9]Institute of Atmospheric and Climate Science, ETH Zurich, Zurich, Switzerland
[10]Division for Climate Modelling and Air Pollution, Norwegian Meteorological Institute, Oslo, Norway
[11] National Institute of Meteorological Sciences, Seogwipo-si, Jeju-do, Korea
[12]Research Institute for Applied Mechanics, Kyushu University, Fukuoka, Japan
[13]Beijing Climate Center, China Meteorological Administration, Beijing, China

*Correspondence to*: Steven Turnock (steven.turnock@metoffice.gov.uk)

**Abstract.**

Poor air quality is currently responsible for large impacts on human health across the world. In addition, the air pollutants, ozone ($O_3$) and particulate matter less than 2.5 microns in diameter ($PM_{2.5}$), are also radiatively active in the atmosphere and can influence Earth's climate. It is important to understand the effect of air quality and climate mitigation measures over the historical period and in different future scenarios to ascertain any impacts from air pollutants on both climate and human health. The 6th Coupled Model Intercomparison Project (CMIP6) presents an opportunity to analyse the change in air pollutants simulated by the current generation of climate and Earth system models that include a representation of chemistry and aerosols (particulate matter). The shared socio-economic pathways (SSPs) used within CMIP6 encompass a wide range of trajectories in precursor emissions and climate change, allowing for an improved analysis of future changes to air pollutants. Firstly, we conduct an evaluation of the available CMIP6 models against surface observations of $O_3$ and $PM_{2.5}$. CMIP6 models consistently overestimate observed surface $O_3$ concentrations across most regions and in most seasons by up to 16 ppb, with a large diversity in simulated values over northern hemisphere continental regions. Conversely, observed surface $PM_{2.5}$ concentrations are consistently underestimated in CMIP6 models by up to 10 µg m$^{-3}$, particularly for the northern hemisphere winter months, with the largest model diversity near natural emission source regions. The biases in CMIP6 models when compared to observations of $O_3$ and $PM_{2.5}$ are similar to those found in previous studies. Over the historical period (1850-2014) large increases in both surface $O_3$ and $PM_{2.5}$ are simulated by the CMIP6 models across all regions, particularly over the mid to late 20th Century when anthropogenic emissions increase markedly. Large regional historical changes are simulated for both pollutants, across East and South Asia, with an annual mean increase of up to 40 ppb for $O_3$ and 12 µg m$^{-3}$ for $PM_{2.5}$. In future scenarios containing strong air quality and climate mitigation measures (ssp126), annual mean concentrations of air pollutants are substantially reduced across all regions by up to 15 ppb for $O_3$ and 12 µg m$^{-3}$ for $PM_{2.5}$. However, for scenarios that encompass weak action on mitigating climate and reducing air pollutant emissions (ssp370), annual mean increases of both surface $O_3$ (up 10 ppb) and $PM_{2.5}$ (up to 8 µg m$^{-3}$) are simulated across most regions, although, for regions like North America and Europe small reductions in $PM_{2.5}$ are simulated due to the regional reduction of precursor emissions in this scenario. A comparison of simulated regional changes in both surface $O_3$ and $PM_{2.5}$ from individual CMIP6 models highlights important regional differences due to the simulated interaction of aerosols, chemistry, climate and natural emission sources

within models. The projection of regional air pollutant concentrations from the latest climate and Earth system models used within CMIP6 shows that the particular future trajectory of climate and air quality mitigation measures could have important consequences for regional air quality, human health and near-term climate. Differences between individual models emphasises the importance of understanding how future Earth system feedbacks influence natural emission sources e.g. response of biogenic emissions under climate change.

## 1 Introduction

Air pollutants are important atmospheric constituents as they have large impacts on human health (Lelieveld et al., 2015), damage ecosystems (Fowler et al., 2009) and can also influence climate through changes in the Earth's radiative balance (Boucher et al., 2013; Myhre et al., 2013). Two major components of air pollution at the surface are ozone ($O_3$) and particulate matter less than 2.5 microns in diameter ($PM_{2.5}$). Exposure to present day ambient concentrations of these two air pollutants was estimated as causing up to 4 million premature deaths per year (Apte et al., 2015; Malley et al., 2017). Over recent decades, the impact on human health from exposure to air pollutants has been increasing (Butt et al., 2017; Cohen et al., 2017). Additionally, elevated levels of air pollutants over recent decades have also been responsible for ecosystem damage to crops and vegetation, although there have been recent improvements in environmental health (de Wit et al., 2015).

In terms of climate impact, tropospheric $O_3$ has a positive radiative forcing on climate over the industrial period and is the third most important greenhouse gas in terms of radiative forcing (Myhre et al., 2013). However, depletion of $O_3$ in the stratosphere has resulted in a net negative top of atmosphere radiative forcing over recent decades (Checa-Garcia et al., 2018). Particulate matter (PM), also referred to as aerosols, has an overall negative radiative forcing on climate, both directly and indirectly through the modification of cloud properties (Boucher et al., 2013). Both $O_3$ and PM are relatively short lived in the troposphere, with a typical lifetime of less than 2 weeks in the lower atmosphere and are commonly referred to as Short-lived Climate Forcers (SLCFs). Future air pollutant concentrations and distributions are driven by changes to both precursor emissions and climate. Emission control measures on a national and international level can both influence future changes to air pollutants, with global increases in $CH_4$ abundance potentially offsetting benefits to surface $O_3$ from local emission reductions (Fiore et al., 2002; Shindell et al., 2012; Wild et al., 2012). For $PM_{2.5}$, changes in concentrations are dependent on both emission rates and levels of atmospheric oxidants, although changes in specific aerosol components can be more directly related to emissions, e.g. black carbon. In a warming world, background $O_3$ concentrations over remote locations are likely to decrease (Johnson et al., 1999; Isaksen et al., 2009; Fiore et al., 2012; Doherty et al., 2013), whereas over anthropogenic source regions, which have higher average surface $O_3$ concentrations, an increase is anticipated (Rasmussen et al., 2013; Colette et al., 2015). The climate impact on $PM_{2.5}$ is much more uncertain and variable across regions, with both increases and decreases predicted due to the uncertainty of future meteorological effects (Jacob and Winner, 2009; Allen et al., 2016; Shen et al., 2017). However, any such climate change impacts on $PM_{2.5}$ are considered to be smaller than the effect from implementing emission mitigation measures (Westervelt et al., 2016).

Experiments conducted as part of the 5th Coupled Model Intercomparison Project (CMIP5; Taylor et al., 2012) and the Atmospheric Chemistry and Climate Model Intercomparison Project (ACCMIP, Lamarque et al., 2013) contributed to a multi-model assessment of future trends in air pollutants. Global annual mean surface $O_3$ concentrations were predicted to increase by up to 5 ppb in 2100 using RCP8.5 (Representative Concentration Pathway with an anthropogenic radiative forcing of 8.5 W m$^{-2}$ in 2100); the RCP with largest increases in methane ($CH_4$) abundances and the largest climate change signal used in CMIP5 (Kirtman et al., 2013). The other RCPs used in CMIP5 had a lower climate forcing and smaller changes in $CH_4$ abundance with models predicting global annual mean surface $O_3$ concentrations that showed little change in the short term (up to 2050) but decreased by around 5 ppb in 2100. The scenario differences in the global mean response for surface $O_3$ were generally reflected across other regions, although with a larger magnitude of change over the northern hemisphere continental

regions. The predicted range of future surface $O_3$ concentrations was previously found to be dominated by changes in precursor emissions (Fiore et al., 2012). However, in regions remote from pollution sources (low-NOx) future climate change was shown to result in a small reduction in surface $O_3$ concentrations. For $PM_{2.5}$, results from CMIP5 and ACCMIP models showed annual mean concentrations declining in most regions and across all scenarios due to the reduction in aerosol emissions. Globally, $PM_{2.5}$ concentrations reduced by ~1 $\mu g\ m^{-3}$ by 2100, whereas larger regional reductions of up to 6 $\mu g\ m^{-3}$ were predicted by 2100. Exceptions to this occurred over South and East Asia where $PM_{2.5}$ concentrations increased by up to 3 $\mu g\ m^{-3}$ in the near-term (up to 2050), after which concentrations reduced by 2100. The largest difference in the response of $PM_{2.5}$ across the scenarios was also shown across East and South Asia due to differences in the carbonaceous and sulphur dioxide ($SO_2$) emission trajectories (Fiore et al., 2012). Future $PM_{2.5}$ concentrations over Africa and the Middle East were shown to be quite noisy due to the large meteorological variability that influences dust emissions over these regions.

The current set of experiments conducted for the 6th Coupled Model Intercomparison Project (CMIP6; Eyring et al., 2016) represent an opportunity to update the assessment of current and future levels of air pollutants using the latest generation of Earth system and climate models. A new set of future scenarios have been generated for CMIP6, the Shared Socio-economic Pathways (SSPs), which combine different trends in social, economic and environmental developments (O'Neill et al., 2014). Varying amounts of emission mitigation to SLCFs are applied on top of the baseline social and economic developments to meet predefined climate and air quality targets in the future, allowing for a wider range of future air pollutant trajectories to be assessed than occurred in CMIP5 (Rao et al., 2017; Riahi et al., 2017). Initial assessments have been made of future changes to air pollutants in the SSPs using simplified models (Reis et al., 2018; Turnock et al., 2018, 2019). The sustainability pathway (SSP1) leads to improvements in both air quality and climate, whereas SSP3 (regional rivalry) is not compatible with achieving air quality and climate goals, and the conventional fuels (SSP5) pathway improves air quality at the expense of climate (Reis et al., 2018). Strong climate and air pollutant mitigation measures in SSP1 were shown to reduce global annual mean surface $O_3$ concentrations by more than 3.5 ppb, whereas for SSP3 $O_3$ concentrations over Asia were predicted to increase by 6 ppb (Turnock et al., 2019). These studies highlighted the potential large regional variability in the response of air pollutants to the different assumptions in the future pathways and also the need for a full model assessment using the current generation of Earth System Models (ESMs) that take into account both changes in emissions and climate.

In this study, we use results from experiments conducted as part of CMIP6 to make a first assessment of historical and future changes in air pollutants. First, we assess the performance of CMIP6 models in simulating present day air pollutants by conducting an evaluation against observations of $O_3$ and $PM_{2.5}$. Regional changes in surface $O_3$ and $PM_{2.5}$ are computed over the historical period (1850-2014) to provide context with future changes. We are then able to show future projections of air pollutants over different world regions under different Shared Socio-economic Pathways used in the CMIP6 experiments. Finally, a comparison is made of individual CMIP6 models for a single future scenario (ssp370) to identify potential reasons for model discrepancies.

## 2 Methods

### 2.1 Air Pollutant Emissions

A new set of historical and future anthropogenic air pollutant emissions has been developed and used as part of CMIP6. The historical anthropogenic emissions are from the Community Emissions Data System (CEDS) and a new dataset was developed for biomass burning emissions, both of which provides information on emissions from 1750 to 2014 (van Marle et al., 2017; Hoesly et al., 2018). The SSPs used in future CMIP6 experiments represent an update from the RCPs used in CMIP5, as they combine pathways of socio-economic development with targets to achieve a certain level of climate mitigation (O'Neill et al., 2014; van Vuuren et al., 2014; Riahi et al., 2017). The SSPs are divided into the following 5 different pathways depending on their social, economic and environmental development: SSP1 – sustainability, SSP2 - middle-of-the-road, SSP3 – regional

rivalry, SSP4 - inequality, SSP5 – fossil fuel development. An assumption about the degree of air pollution control (strong, medium or weak) is included on top of the baseline pathway, with stricter air pollution controls assumed to be tied to economic development (Rao et al., 2016). Weak air pollution controls occur in SSP3 and SSP4, with medium controls in SSP2 and strong air pollution controls in SSP1 and SSP5 (Gidden et al., 2019). A particular climate mitigation target, in terms of an anthropogenic radiative forcing by 2100, and the range of emission mitigation measures associated with achieving it are included in addition to the existing policy measures within each baseline SSP scenario. Climate mitigation targets vary from a weak mitigation scenario with an anthropogenic radiative forcing of 8.5 W m$^{-2}$ by 2100, comparable with a 5 °C temperature change (Riahi et al., 2017), to a strong mitigation scenario with a radiative forcing of 1.9 W m$^{-2}$ by 2100, in accordance with the Paris agreement for keeping temperatures below 2 °C (United Nations, 2016). Some climate mitigation targets are comparable with those of the RCPs used in CMIP5 (2.6, 4.5 and 6.0), whilst others are new, e.g. ssp534-over is included as a delayed mitigation scenario. A scenario specific to the Aerosol and Chemistry Model Intercomparison Project (AerChemMIP), ssp370-lowNTCF, is also included to study the impact of mitigation measures to specifically control SLCFs on top of ssp370. Future biomass burning emissions vary in each scenario, depending on the particular land-use assumptions (Rao et al., 2017). Whilst future anthropogenic and biomass burning emissions are prescribed in each CMIP6 model from the same dataset, other natural emissions, e.g. dust, biogenic volatile organic compounds (BVOCs) etc., will be different and depend on the individual model configuration.

Figure 1 shows the future changes in global total (anthropogenic and biomass) emissions of the major air pollutant precursors across all of the CMIP6 scenarios, provided as input to the CMIP6 models. The overlying feature is that global air pollutant emissions are predicted to reduce across the majority of scenarios by 2100. The exception to this is that global and regional emissions increase or remain at present day levels for ssp370 (Figs. 1 and 2). Some air pollutant emissions increase in the near-term in other scenarios e.g. nitrogen oxides (NOx) in ssp585 (by up to 15%), but by 2100 these have been reduced. Future CH$_4$ abundances show the largest diversity amongst the SSPs. Large increases in global CH$_4$ abundances of more than 50% are predicted for the fossil fuel dominated pathways of ssp370 and ssp585, whereas large reductions of ~50% are predicted to occur in the strong mitigation scenarios of SSP1.

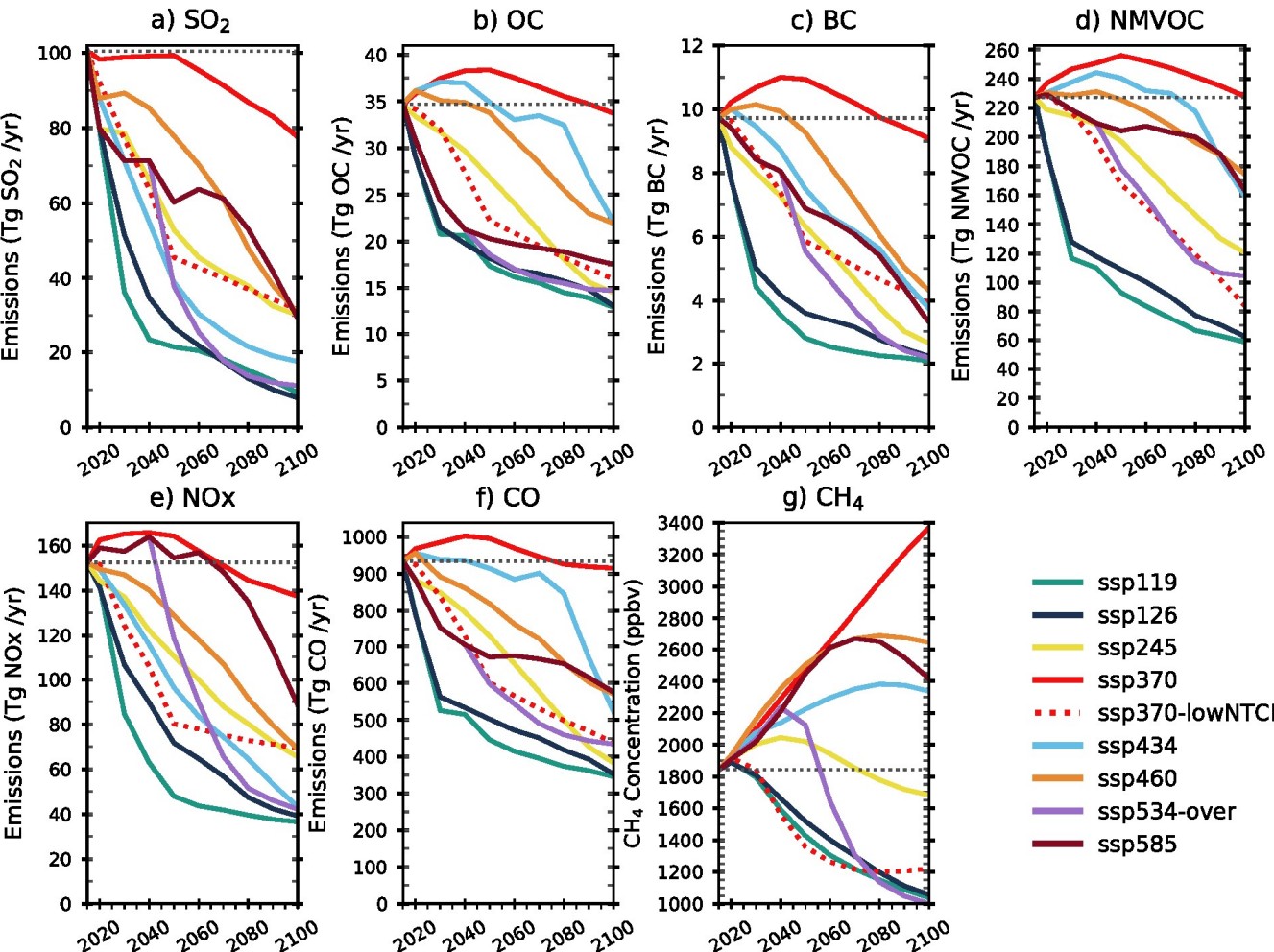

**Figure 1: Changes in annual total (anthropogenic and biomass) global air pollutant emissions (relative to 2015) of sulphur dioxide (SO₂), organic carbon (OC), black carbon (BC), non-methane volatile organic compounds (NMVOCs), nitrogen oxides (NOx), carbon monoxide (CO) and global methane (CH₄) abundances in the future CMIP6 scenarios used as input to CMIP6 models. The dashed black line represents the 2015 value. Global CH₄ abundances are not reduced in the AerChemMIP ssp370-lowNTCF simulations used here.**

For SO₂, large reductions of more than 50% are shown for most scenarios and across most regions (Figure 2), apart from Africa and Asia in ssp370. Near-term (2050) increases in SO₂ occur over South Asia and other developing regions, which are then reduced in the latter half of the 21ˢᵗ Century. Over Europe and North America consistent decreases are predicted across all scenarios. The other major aerosol emissions, OC and BC, show similar reductions to SO₂ across all scenarios and regions. For all aerosol and aerosol precursors, a reduction of 80-100% (relative to 2015) in regional emissions is predicted by 2100 in the strong mitigation scenarios. Changes in the emissions of the O₃ precursors, NOx, CO and non-methane volatile organic compounds (NMVOCs), show a similar increase across most regions for ssp370 but a general decrease in other scenarios. The change in these emissions are particularly diverse across all the scenarios in South Asia with large relative increases in ssp370 (of up to 50%), in contrast to the large decreases in ssp126 (up to 40%). Across East Asia there is a 20% increase in NOx emissions for ssp370 in 2050 but a long term reduction across all scenarios.

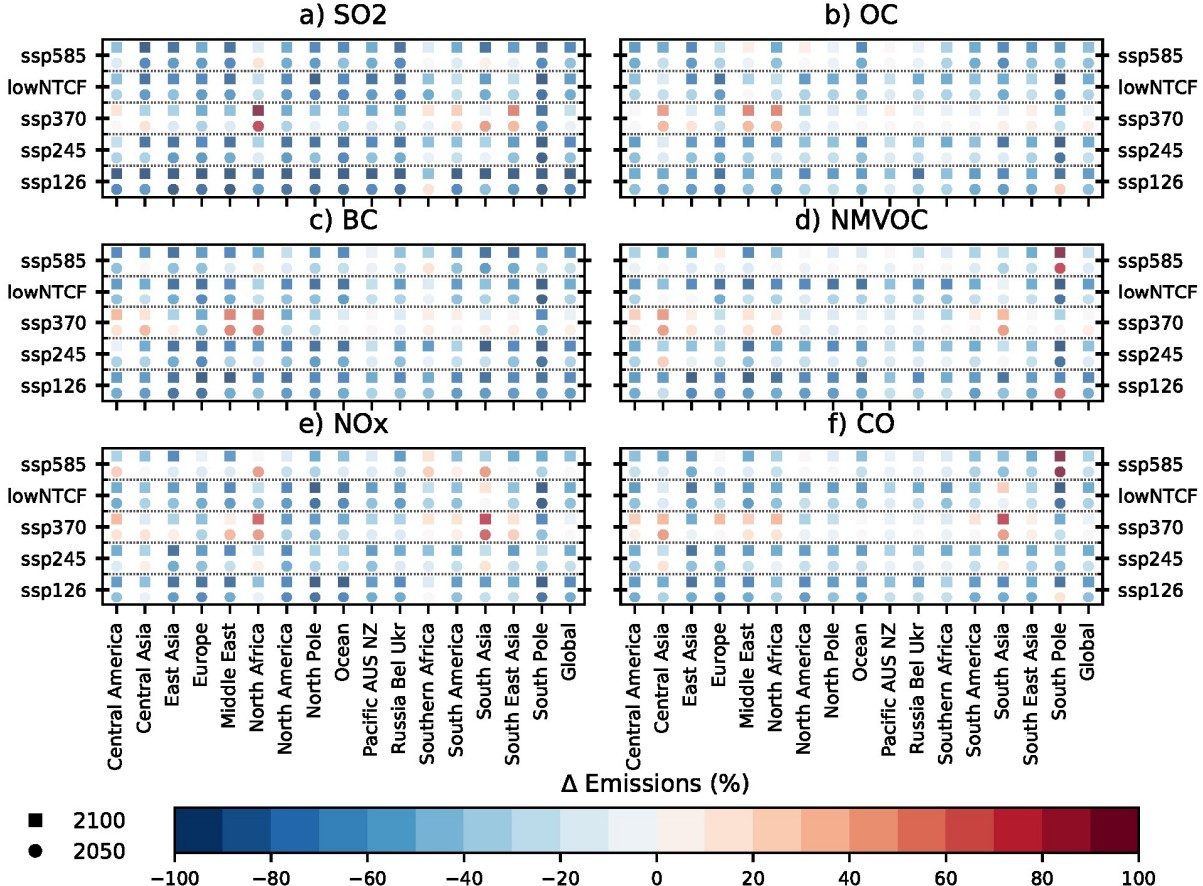

**Figure 2: Percent change in 2050 (circles) and 2100 (squares), relative to 2015, for annual mean total (anthropogenic and biomass) air pollutant emissions of a) SO₂, b) OC, c) BC, d) NMVOCs, e) NOx and f) CO across different world regions in the 4 Tier 1 future CMIP6 scenarios and the ssp370-lowNTCF scenario (identified as lowNTCF). Regions are defined in Figure S1.**

## 2.2 CMIP6 Simulations

Surface concentrations of $O_3$ and $PM_{2.5}$ have been obtained from all the CMIP6 models that made appropriate data available on the Earth System Grid Federation (ESGF) at the time of writing. To study changes in surface air pollutants over the industrial period data has been obtained from the coupled historical simulations (Eyring et al., 2016) over the period 1850 to 2014 from all of the available ensemble members of each available CMIP6 model. For each model, a mean is taken using all available ensemble members prior to the calculation of multi-model mean. For model evaluation purposes, 10 years of data from historical simulations has been used over the period that is relevant to the particular observational dataset (2000-2010 for ground-based $PM_{2.5}$, 2004-2014 for $PM_{2.5}$ reanalysis product and 2005-2014 for ground-based $O_3$). To investigate future changes in air pollutants, all available data has been obtained over the period 2015 to 2100 for each of the different future coupled atmosphere-ocean model experiments, conducted as part of ScenarioMIP (O'Neill et al., 2016). CMIP6 model data has also been obtained for the AerChemMIP specific ssp370-lowNTCF scenario, which was only required to be conducted over the period 2015-2055 (Collins et al., 2017).

Concentrations of both pollutants at the surface have been obtained by extracting the lowest vertical level of the full 3D field output on the horizontal and vertical grid of each model (the "AERmon" CMIP6 table ID). For $O_3$, this is supplied as a separate diagnostic which can be used directly. However, models contributing to CMIP6 will not all directly output $PM_{2.5}$ and the calculation of $PM_{2.5}$ will not be consistent across individual models due to the different treatment of aerosols and their components. For example only a few CMIP6 models include the simulation of ammonium nitrate in their aerosol scheme (currently, only GISS-E2-1-G and GFDL-ESM4 have provided nitrate mass mixing ratios on the ESGF database). Therefore, to use a consistent definition across all models, we calculated $PM_{2.5}$ offline. In this study surface $PM_{2.5}$ is defined as the sum of the individual dry aerosol mass mixing ratios of black carbon (BC), total organic aerosol (OA – both primary and secondary

sources), sulphate ($SO_4$), sea salt (SS) and dust (DU) from the lowest model level extracted from the full 3D model fields. All BC, OA and $SO_4$ aerosol mass is assumed to be present in the fine size fraction (< 2.5 µm), whereas a factor of 0.25 for SS and 0.1 for DU has been used to calculate the approximate contribution from these components to the fine aerosol size fraction (Eq. 1).

$$PM_{2.5} = BC + OA + SO_4 + (0.25 \times SS) + (0.1 \times DU) \hspace{3cm} (1)$$

The factors used to calculate the contribution of SS and DU concentrations to the $PM_{2.5}$ size fraction are likely to depend on the individual aerosol scheme and the simulated aerosol size distribution within a particular model. The calculation of an approximate $PM_{2.5}$ concentration using Eq. (1) is therefore likely to introduce some errors but it does provide an estimate that is consistent across models and also with that previously used in CMIP5 and ACCMIP (Fiore et al., 2012; Silva et al., 2013, 2017). For the CNRM-ESM2-1 model, anomalously large concentrations were obtained from the sea salt mass mixing ratios. Sensitivity tests with this model suggested that a much smaller factor of 0.01 was more appropriate to use for SS, which takes into account the non-dry nature of the sea salt aerosols and the large possible size range, up to 20 µm in diameter, of sea salt particles within the CNRM-ESM2-1 model (P Nabat 2019, personal communication, 27[th] November).

Details of the data used in this study from different CMIP6 models, in both the historical and future scenarios, is presented below in Table 1. For the historical period, data was available from 5 different CMIP6 models for $O_3$ and 10 models for $PM_{2.5}$. The future scenario with the most data available was ssp370, with 4 models suppling data for $O_3$ and 7 models for $PM_{2.5}$. For the other Tier 1 CMIP6 scenarios (ssp126, ssp245 and ssp585), data was only available for 2 models for $O_3$ and 4 for $PM_{2.5}$ (all components). It was decided to focus the analysis on ssp370 and other Tier 1 scenarios due to the limited availability of model data for Tier 2 scenarios (ssp119, ssp434, ssp460 and ssp534-over). The results from an $O_3$ parameterisation (Turnock et al., 2018, 2019), referred to in this study as HTAP_param, have also been included in the analysis of surface $O_3$ from CMIP6 models for both the historical and future scenarios. The HTAP_param was previously developed based upon the source-receptor relationships of $O_3$ derived from perturbation experiments of regional precursor emissions and global $CH_4$ abundances (Wild et al., 2012; Turnock et al., 2018). The HTAP_param applies the fractional change in global $CH_4$ abundance and regional emission precursors (NOx, CO and NMVOCs) for a particular scenario to the ozone response from each individual model used in the parameterisation. The total $O_3$ response is obtained by summing up the response from each of the individual models to all precursor changes across all source regions. The surface $O_3$ response previously calculated from the HTAP_param in both the historical and future CMIP6 scenarios is compared to that from the CMIP6 models (Turnock et al., 2019). The $O_3$ parameterisation does not take into account the effects of climate change on surface $O_3$ concentrations and therefore provides an estimate of the emission-only driven changes to surface $O_3$, which we compare to the climate and Earth System models.

**Table 1 –Number of ensemble members used for the historical and future scenarios experiments from each model in the analysis of surface $O_3$ and $PM_{2.5}$ in this study**

| Model | Pollutant | historical | ssp126 | ssp245 | ssp370 | ssp370-lowNTCF | ssp585 | Model Refs | Data Citation |
|---|---|---|---|---|---|---|---|---|---|
| BCC-ESM1 | $O_3$, $PM_{2.5}$ | 3 | | | 3 | 3 | | (Wu et al., 2019, 2020) | (Zhang et al., 2018, 2019) |
| CESM2-WACCM | $O_3$, $PM_{2.5}$ | 3 | | | 1 | 1 | | (Gettelman et al., 2019; Tilmes et al., 2019; Emmons et al., 2020) | (Danabasoglu, 2019b, 2019c, 2019a) |
| CNRM-ESM2-1 | $PM_{2.5}$ | 3 | | | 3 | 3 | | (Michou et al., 2019; Séférian et al., 2019) | (Seferian, 2018, 2019; Voldoire, 2019) |
| GFDL-ESM4 | $O_3$, $PM_{2.5}$ | 1 | 1 | 1 | 1 | 1 | 1 | (Horowitz et al., 2019; Dunne et al., 2020) | (Horowitz et al., 2018; John et al., 2018; Krasting et al., 2018) |
| HadGEM3-GC31-LL | $PM_{2.5}$ | 4 | 1 | 1 | | | 1 | (Kuhlbrodt et al., 2018) | (Ridley et al., 2018; Good, 2019) |
| MIROC6 -ES2L | $PM_{2.5}$ | 3 | 1 | 1 | 1 | | 1 | (Takemura, 2012; Hajima et al., 2019) | (Hajima and Kawamiya, 2019; Tachiiri and Kawamiya, 2019) |
| MPI-ESM1.2-HAM | $PM_{2.5}$ | 1 | | | 1 | 1 | | (Tegen et al., 2019) | (Neubauer et al., 2019) |
| MRI-ESM2-0 | $O_3$, | 5 | 1 | 1 | 3 | 1 | 1 | (Yukimoto et al., 2019d; Oshima et al., 2020) | (Yukimoto et al., 2019b, 2019c, 2019a) |
| | $PM_{2.5}$ | 5 | 1 | 1 | 3 | 1 | 1 | | |
| GISS-E2-1-G | $O_3$, | 5 | 1 | 5 | 1 | | 1 | (Bauer et al., 2020) | (NASA Goddard Institute For Space Studies (NASA/GISS), 2018) |
| | $PM_{2.5}$ | 4 | 1 | 5 | 1 | | 1 | | |
| NorESM2-LM | $PM_{2.5}$ | 1 | 3 | 3 | 3 | 3 | 3 | (Karset et al., 2018; Kirkevåg et al., 2018) | (Norwegian Climate Center (NCC), 2018) |
| UKESM1-0-LL | $O_3$, $PM_{2.5}$ | 5 | 5 | 5 | 5 | 3 | 5 | (Sellar et al., 2019) | (Good et al., 2019; Tang et al., 2019) |
| Total Number of models | $O_3$ | 6 | 4 | 4 | 6 | 5 | 4 | | |
| | $PM_{2.5}$ | 11 | 7 | 7 | 10 | 8 | 7 | | |

**2.3 Surface Observations**

Present day surface $O_3$ and $PM_{2.5}$ simulated by all of the CMIP6 models is evaluated against surface observations to ascertain model biases and inter-model discrepancies. Surface $O_3$ observations are obtained from the database of the Tropospheric Ozone Assessment Report (TOAR) (Schultz et al., 2017). The TOAR database provides a gridded product of surface $O_3$ observations over the period 1970 to 2015. The majority of measurement sites are located in North America and Europe, with a smaller number of other sites in East Asia, Australia, New Zealand, South America, Southern Africa, Antarctica and remote ocean locations. Here we compile a monthly mean climatology of all available $O_3$ observations over the period 2005-2014 from measurement locations that are classified as rural in the TOAR database (Schultz et al., 2017). The rural locations were selected to be representative of background (i.e. non-urban) $O_3$ concentrations and are considered to be more appropriate in evaluating the simulated values obtained at the relatively coarse horizontal resolution of the global ESMs. Simulated surface $O_3$ concentrations from the CMIP6 models are re-gridded onto the same resolution of the observational product (2° x 2°) for evaluation purposes.

Surface $PM_{2.5}$ observations have been obtained from all of the locations compiled in the database of the Global Aerosol Synthesis and Science Project (GASSP: http://gassp.org.uk/data/, Reddington et al., 2017) to evaluate CMIP6 models. Background, non-urban, $PM_{2.5}$ data is compiled in the GASSP database from three major networks: the Interagency Monitoring of Protected Visual Environments (IMPROVE) network in North America, the European Monitoring and Evaluation Programme (EMEP) and Asia-Pacific Aerosol Database (A-PAD). Again, like for $O_3$, the networks/observations for $PM_{2.5}$ were selected to be representative of non-urban environments, which are more appropriate for the evaluation of global ESMs. With the exception of the IMPROVE network, most measurements of $PM_{2.5}$ began after the year 2000. Like for $O_3$, we compile a monthly mean climatology of $PM_{2.5}$ but now over the period of 2000 to 2010, selected as the GASSP database contained the most observations within this period. Simulated surface $PM_{2.5}$ was computed from CMIP6 models over the same time period as the observations and linearly interpolated to each measurement location. Whilst the surface observations measure total $PM_{2.5}$ mass, the computed $PM_{2.5}$ from CMIP6 models use Eq. 1 and does not include all observable $PM_{2.5}$ aerosol components (e.g. nitrate aerosol). Therefore, it is anticipated that the CMIP6 models will underrepresent the $PM_{2.5}$ observations in this comparison.

To address the anticipated disparity between the observed ground based $PM_{2.5}$ and the approximate $PM_{2.5}$ from CMIP6 models, a further comparison has been made between the CMIP6 models and the Modern-Era Retrospective Analysis for Research and Applications, version 2 (MERRA-2), aerosol reanalysis product (Buchard et al., 2017; Randles et al., 2017). The MERRA-2 aerosol product assimilates observations of Aerosol Optical Depth (AOD) from ground based and satellite remote sensing platforms into model simulations that use the GEOS-5 atmospheric model coupled to the GOCART aerosol module. The data assimilation used in MERRA-2 generally improves comparisons of $PM_{2.5}$ with observations but there are still overestimations due to dust and sea salt and underestimations over East Asia (Buchard et al., 2017; Provençal et al., 2017). Separate mass mixing ratios for BC, OA, $SO_4$, SS and DU aerosol components are provided from MERRA-2, which are then combined using the formula in Eq. 1 to make an approximate $PM_{2.5}$. Monthly mean approximate $PM_{2.5}$ concentrations are then computed over the period 2005-2014 from the MERRA-2 reanalysis product to provide a more direct comparison and enhanced spatial coverage against the approximate $PM_{2.5}$ concentrations calculated from the CMIP6 models calculated over the same time period.

# 3 Present-day Model Evaluation of Air Pollutants

## 3.1 Surface Ozone

The 6 CMIP6 models with data available for the historical experiments are evaluated against surface $O_3$ observations from the TOAR database over the period 2005-2014. A long-term evaluation of surface $O_3$ concentrations from CMIP6 models using observations compiled over the 20th Century is presented separately in Griffiths et al., (2020). Figure 3 shows the annual and seasonal multi-model mean in surface $O_3$ over the period 2005-2014 and the standard deviation across the 6 CMIP6 models. The annual and seasonal mean surface $O_3$ concentrations and evaluation against observations for individual CMIP6 models are shown in Figures S2–S7. Higher surface $O_3$ concentrations are simulated in the northern hemisphere summer (June, July, August- JJA) when $O_3$ formation is enhanced by increased photolytic activity and levels of oxidants, as well as larger biogenic emissions. The hemispheric difference in surface $O_3$ is smaller in December, January and February (DJF) when $O_3$ production is less in the northern hemisphere but higher in the southern hemisphere. However, model diversity is larger in DJF (Fig. 3b) due to individual models simulating different seasonal cycles of $O_3$, particularly UKESM1-0-LL which has the most pronounced seasonal cycle of all 6 models (Fig. S2).

The multi-model mean of CMIP6 models overestimates surface $O_3$ concentrations by up to 16 ppb annually and in both seasons when compared to observations from the TOAR database, although they do capture the broad hemispheric gradient in $O_3$ concentrations (Fig. 3c, 3f and 3i). The model observational comparison of CMIP6 models to the TOAR observations are consistent across all models and with the previous evaluation of ACCMIP models (Young et al., 2018). This indicates a common source of error within models for example uncertainties in emission inventories, deposition processes or vertical mixing (Wild et al., 2020). In addition, the coarse resolution of the ESMs could lead to an overproduction of $O_3$ across polluted regions, with finer resolutions exhibiting improvements in the simulation of surface $O_3$ (Wild and Prather, 2006; Neal et al., 2017). Smaller model biases exist in DJF (<5 ppb) than in JJA (5-15 ppb), mostly attributed to the strong seasonal cycle simulated by UKESM1-0-LL. In contrast to other models (Fig. S2 – S7), UKESM1-0-LL underpredicts surface $O_3$ in DJF over most continental northern hemisphere locations, potentially indicating there is excessive NOx titration of $O_3$ in this model, which is also shown by the large sensitivity of $O_3$ formation to NOx concentrations over the historical period (Fig. S17).

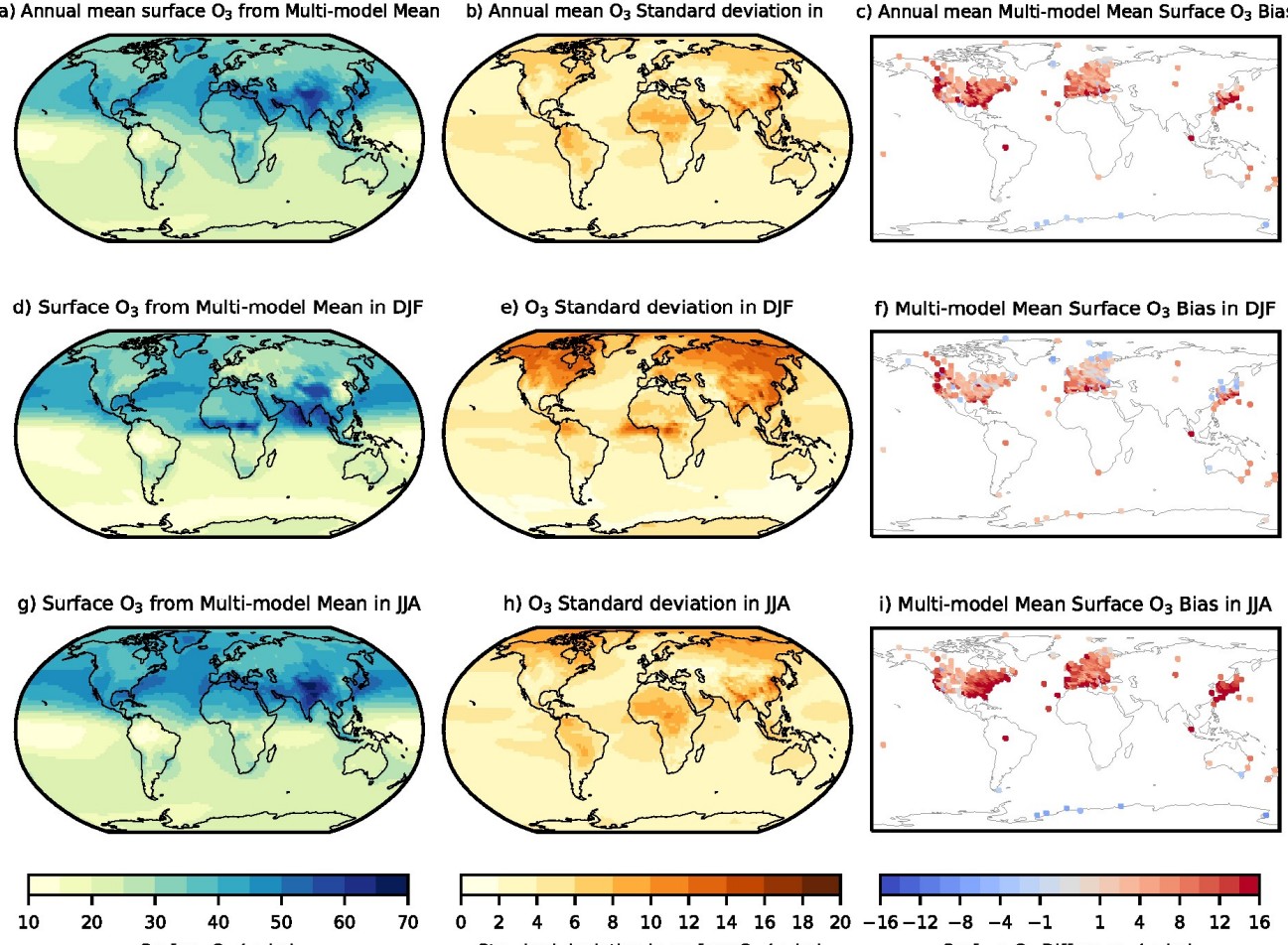

**Figure 3 – Multi-model (6 CMIP6 models) annual and seasonal mean surface O$_3$ concentrations in a) Annual mean, d) December January, February (DJF) and g) June, July, August (JJA) over the 2005-2014 period. The standard deviation in the multi-model mean in b) Annual mean, e) DJF and h) JJA. The difference between the multi-model mean and TOAR observations in c) Annual mean, f) DJF and i) JJA (colour bar saturates).**

The observed annual cycle in surface O$_3$ averaged across measurement locations within different regions is compared to that simulated by CMIP6 models (Figure 4). Across most regions, the mean annual cycle from CMIP6 models compares relatively well to that observed. The overprediction of surface O$_3$ values in JJA is evident across most regions, as is the large concentrations in BCC-ESM1 and GISS-E2-1-G and the strong seasonal cycle in UKESM1-0-LL across northern hemisphere continental regions. Additionally, the timing of peak O$_3$ over continental northern hemisphere locations occurs earlier in the observations (springtime) than in the CMIP6 models (spring and summer), which is consistent with that from ACCMIP models (Young et al., 2018). At oceanic observation locations, surface O$_3$ is overestimated in CMIP6 models by up to 20 ppb across all seasons, indicating that O$_3$ deposition rate could be underestimated here (Clifton et al., 2020). There is also a large overestimation (~20 ppb) in all models at the one observation location in South East Asia, potentially due to difficulty in simulating O$_3$ in the maritime continental boundary layer using lower resolution global ESMs. In contrast to this, CMIP6 models, particularly UKESM1-0-LL and GISS-E2-1-G, tend to underpredict the observed surface O$_3$ concentrations at locations in the South Pole region in JJA by ~5 ppb. This could be due to lack of long range transport of O$_3$ to these sites, inaccuracies in southern hemisphere precursor emissions, or because of the difficulty in simulating O$_3$ concentrations at the appropriate elevation of measurement sites located on the Antarctic ice sheet.

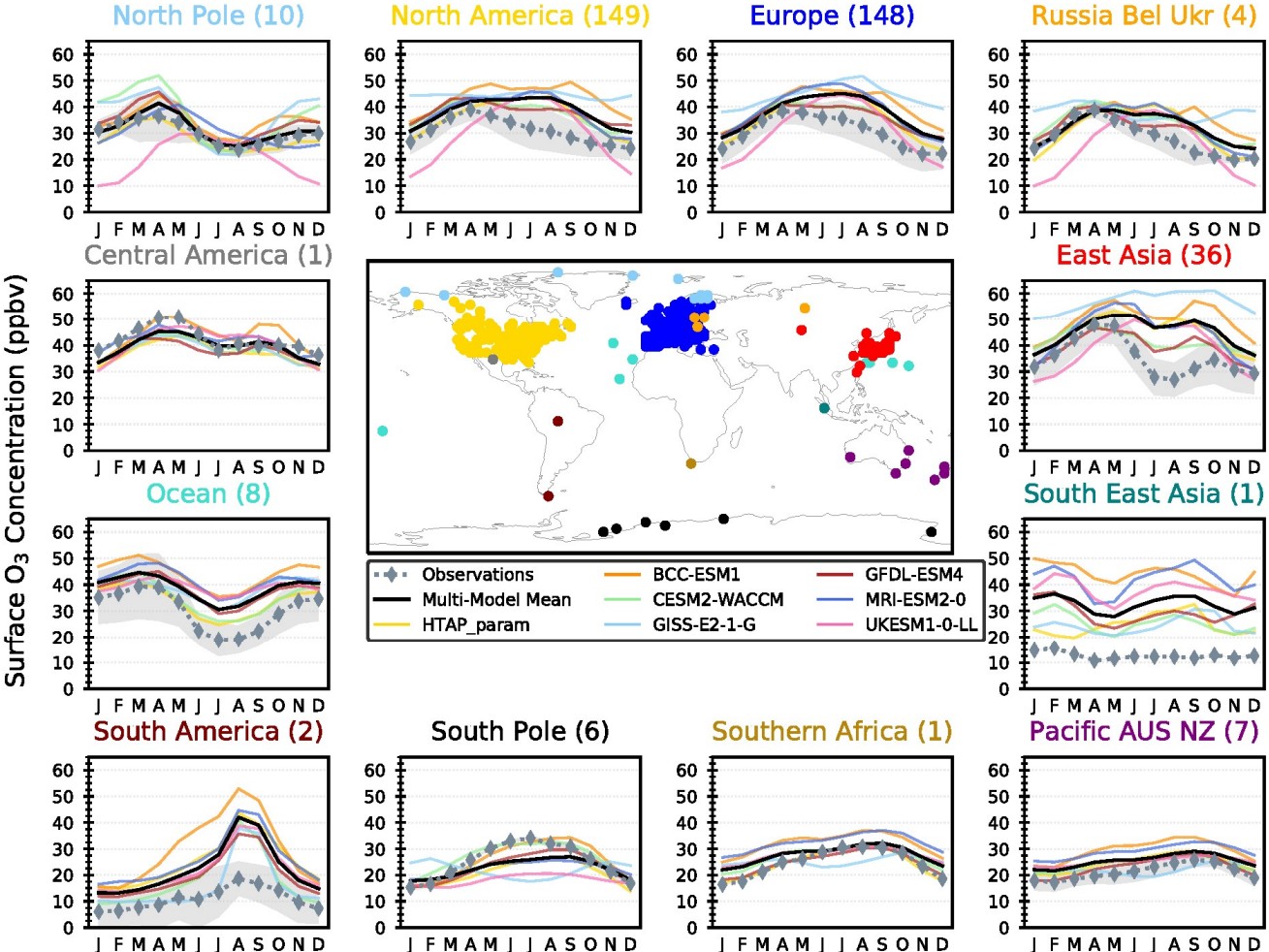

**Figure 4 – Individual and multi-model (6 CMIP6 models and HTAP_param) monthly mean surface O₃ concentrations across different world regions compared with the regional monthly values from all the TOAR observations within the region for the period 2005-2014. The number of observations within a region is shown in parenthesis. The shading shows variability in observations across all sites within the region.**

### 3.2 Surface PM₂.₅

### 3.2.1 Ground Based Observations

A similar comparison is made for annual and seasonal mean surface PM₂.₅ concentrations from CMIP6 models against ground based surface observations (Figure 5). The annual and seasonal multi-model mean from CMIP6 models shows that elevated PM₂.₅ concentrations (>50 μg m⁻³) occur close to the large dust emission source regions of the Sahara and Middle East in both DJF and JJA over 2000-2010. These natural source regions are also one of the largest areas of diversity in PM₂.₅ concentrations (up to 20 μg m⁻³) between the different CMIP6 models (Fig. 5b, 5e, 5h and S8). High concentrations of PM₂.₅ (>40 μg m⁻³) are also simulated over the large anthropogenic source regions of South and East Asia, particularly in DJF when there is enhanced variability across CMIP6 models due to the different contribution from anthropogenic PM₂.₅ components (Fig. S9-S11). The diversity in CMIP6 model is particularly evident in the organic aerosol concentrations across Asia, with higher present day values simulated by CESM2-WACCM and UKESM1-0-LL and lower values in CNRM-ESM2-1 and MIROC-ES2L (Fig. S11). Lower PM₂.₅ concentrations (<10 μg m⁻³) are predicted across both North America and Europe, with more agreement between CMIP6 models. Across the biomass burning regions of South America and Southern Africa, PM₂.₅ concentrations are elevated in JJA with larger diversity in the CMIP6 models due to the differing contributions of the BC and OA components, particularly shown in NorESM2-LM, GISS-E2-1-G and GFDL-ESM4 (Fig. S10 and S11). Relatively consistent PM₂.₅ concentrations of <10 μg m⁻³, with small model diversity (<5 μg m⁻³), are shown across oceanic regions, mainly from emissions

of sea salt (Fig. S12). Apart from the natural sources of aerosol, which are subject to meteorological variability, the CMIP6 models are relatively consistent when simulating PM$_{2.5}$ concentrations across most regions.

Compared to the ground based observations from the GASSP database, the CMIP6 multi-model mean underpredicts the observed PM$_{2.5}$ values by up to 10 μg m$^{-3}$ in both seasons, with a slightly larger underestimation in DJF than JJA. As discussed in section 2.3, an underestimation was anticipated from comparing approximate PM$_{2.5}$ concentrations, derived from CMIP6 models, to observed values. Nevertheless, the evaluation highlights that fine particulate matter (PM$_{2.5}$) is generally underrepresented in the CMIP6 models across North America, Europe and parts of Asia for which observations are available;

a similar result to other studies evaluating global and regional models (Tsigaridis et al., 2014; Pan et al., 2015; Glotfelty et al., 2017; Solazzo et al., 2017; Im et al., 2018). Numerous reasons potentially exist for the model observation discrepancy shown here and in other studies including uncertainties in emission inventories (e.g. local dust sources), errors in the wet/dry deposition schemes, the absence/underrepresentation of aerosol formation processes (e.g. secondary organic aerosols) and the coarse resolution of global models leading to errors in emissions and simulated meteorology. Understanding the causes of

model observational discrepancies is an area of active research and should be explored in further research, for example in a global multi-model sensitivity study that examines model uncertainties.

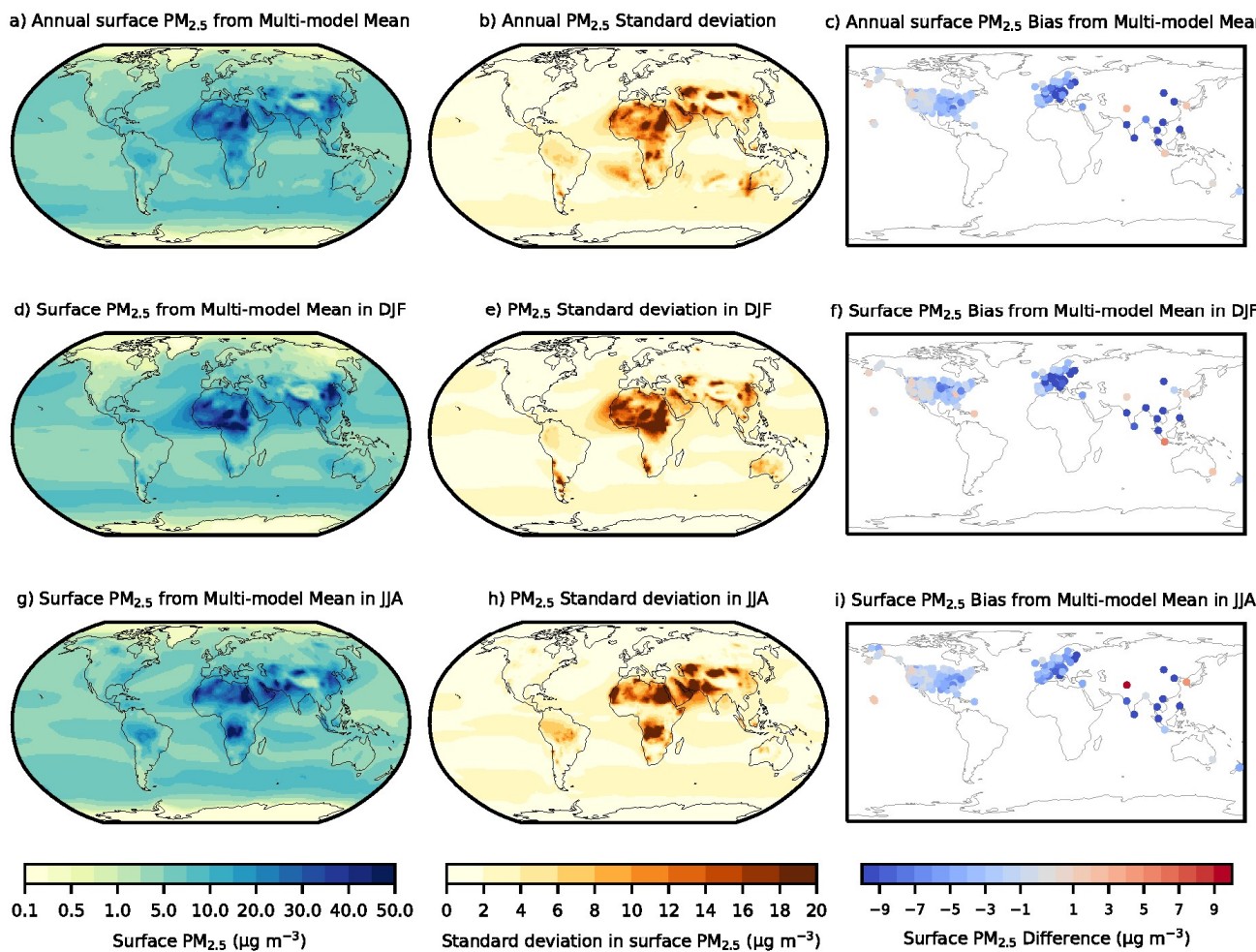

**Figure 5 – Multi-model (11 CMIP6 models) annual and seasonal mean surface PM$_{2.5}$ concentrations in a) annual mean, d) December January, February (DJF) and g) June, July, August (JJA) over the 2000-2010 period. The standard deviation in the multi-model**

**mean in b) annual mean, e) DJF and h) JJA. The difference between the multi-model mean and PM$_{2.5}$ observations in c) annual mean, f) DJF and i) JJA (colour bar saturates).**

The simulated regional mean annual cycle in surface PM$_{2.5}$ from different CMIP6 models against observations is shown in Figure 6. The low model bias in PM$_{2.5}$ concentrations is highlighted across all regions, except for the ocean region where there is a relatively large diversity in model simulations, particularly MIROC-ES2L and NorESM2-LM, at these observation

locations. Across North America, the region with most observations, the annual cycle is simulated relatively well with a peak

in concentrations in JJA and a lower model bias, although a larger model bias (factor of ~1.5 to 2) occurs in winter and spring. Across Europe, there is a larger underestimation of observed PM$_{2.5}$ concentrations by CMIP6 models in DJF (factor > 2) than JJA. Nitrate aerosols are observed and modelled (from two CMIP6 models in Fig. S13) to contribute between 1 and 5 µg m$^{-3}$ of the total aerosol mass over Europe (Fagerli and Aas, 2008; Pozzer et al., 2012), explaining  part, but not all, of the model

observational discrepancy here. Additionally, on Fig. 6 the CMIP6 models also underestimate the MERRA-2 reanalysis product (which does not include nitrate aerosols), indicating that other aerosol sources/processes are underrepresented across Europe and other regions in the models. The limited number of observations across other regions makes it difficult to infer particular model/observational biases. However, over Asia CMIP6 PM$_{2.5}$ concentrations tend to be within a factor of 2 of the observations and represent the seasonal cycle relatively well at these locations. Over Asia, larger PM$_{2.5}$ concentrations are

simulated in the CMIP6 models CESM2-WACCM, HadGEM3-GC31-LL and UKESM1-0-LL, mainly due to the larger OA component (Fig. S11). Across South Asia, concentrations are relatively well simulated in JJA but a larger discrepancy (15 µg m$^{-3}$) exists in DJF between the model and observations.

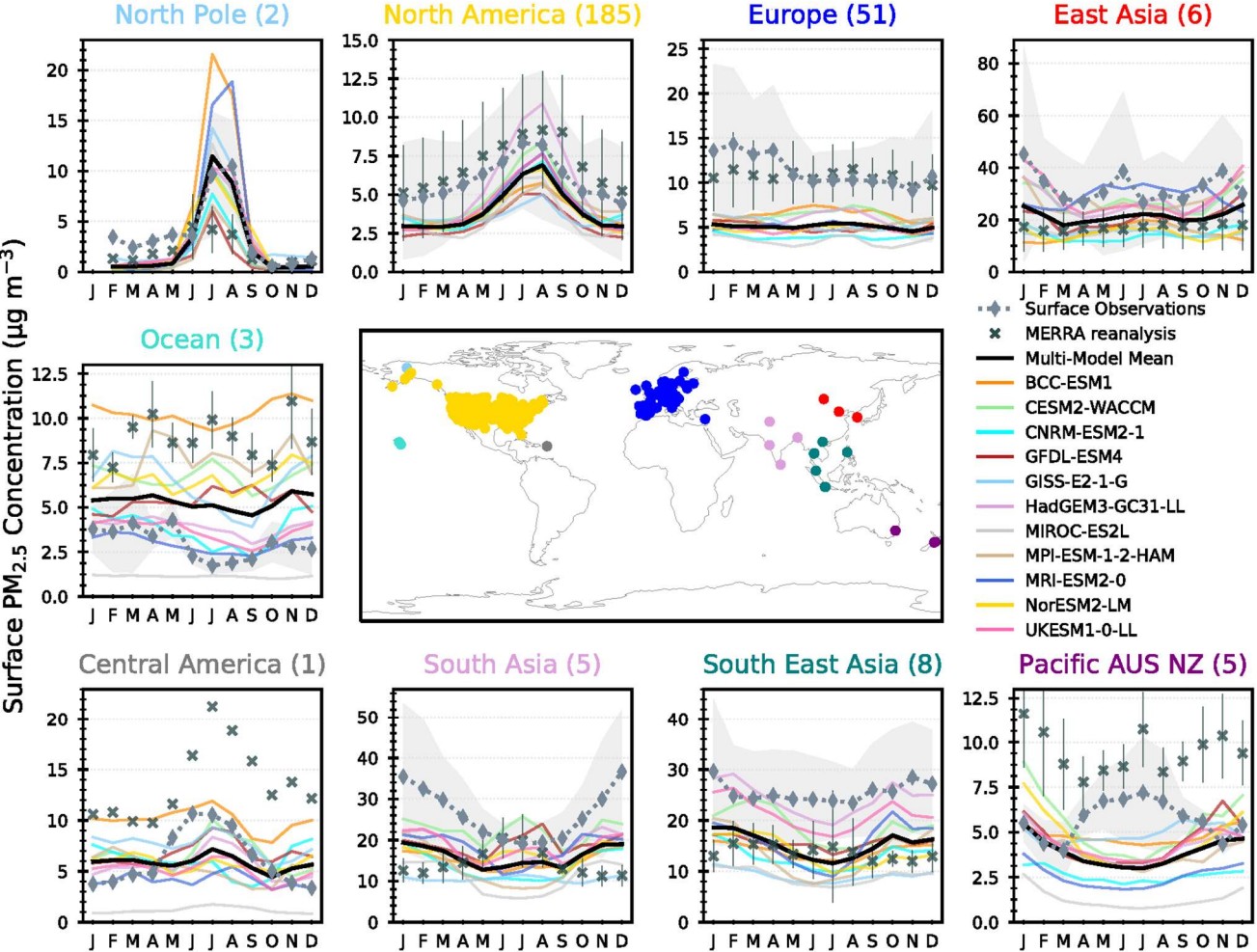

**Figure 6 – Individual and multi-model (11 CMIP6 models) monthly mean surface PM$_{2.5}$ concentrations across different world regions compared with the regional monthly values from all the PM$_{2.5}$ observations (◊) and the MERRA-2 reanalysis product (x) within the region for the period 2000-2010. The number of observations within the region is shown in parenthesis. The shading and errors bars show variability in observations and the reanalysis product across all sites within the region.**

### 3.2.2 MERRA Reanalysis Product

An additional comparison of surface PM$_{2.5}$ concentrations from the MERRA-2 aerosol reanalysis product is made with that simulated by the CMIP6 models to improve the spatial coverage and provide a more consistent evaluation of the approximate PM$_{2.5}$ concentrations. Figure 7 shows the same comparison as in Fig. 5 but now using the approximate PM$_{2.5}$ obtained from the MERRA-2 reanalysis product over the period 2005-2014. In comparison to MERRA-2, the CMIP6 models are shown to

underpredict PM$_{2.5}$ concentrations across North America, Europe and Eurasia, but by a smaller amount than in comparison to

ground-based observations. A similar seasonal cycle comparison is shown for Europe and North America (regions with most ground based observations) in both Fig. 6 and 8, providing confidence that the underestimation of PM$_{2.5}$ by CMIP6 models is robust over these regions. Across all other regions, the MERRA-2 reanalysis product provides much greater spatial coverage for each region and therefore the features shown in the site-level comparison (Fig. 6) will not necessarily apply here. A large overestimation of the MERRA-2 reanalysis product by the CMIP6 multi-model mean is shown across East and South Asia.

Figure 8 shows that on a regional mean basis most CMIP6 models are within the spread of the MERRA-2 concentrations for East Asia, although MERRA-2 was previously shown to underestimate PM$_{2.5}$ concentrations across East Asia (Buchard et al., 2017; Provençal et al., 2017) and also on Fig. 6. CESM2-WACCM and MRI-ESM2-0 are the exceptions to this with distinctly higher PM$_{2.5}$ concentrations over East Asia, potentially due to larger OA concentrations and more dust aerosols within the western side of this region (Fig. S8 and S11). Across the South Asian region, CMIP6 models consistently overestimate

MERRA-2 by more than 10 µg m$^{-3}$ in certain months. UKESM1-0-LL, MRI-ESM2-0 and CESM2-WACCM simulate particularly high monthly PM$_{2.5}$ concentrations of 20-40 µg m$^{-3}$ over South Asia, due to large contributions from SO$_4$, dust and OA. Across North Africa there is considerable variability in PM$_{2.5}$ within this region, as CMIP6 models both under and over-estimate the MERRA-2 PM$_{2.5}$ concentrations, although this results in a relatively good regional mean representation (Fig. 7 and 8). The annual mean cycle in MERRA-2 PM$_{2.5}$ concentrations across South America is well represented by the CMIP6

models, although the peak in the biomass burning season is underestimated by 5-10 µg m$^{-3}$ in some models. A more pronounced annual cycle is exhibited by UKESM1-0-LL across Southern Africa, due to the larger contributions from the OA fraction (Fig. S11), potentially from enhanced biogenic emissions that result in secondary OA formation (SOA). Across oceanic locations all of the CMIP6 models underestimate the MERRA-2 PM$_{2.5}$ concentrations by 5 µg m$^{-3}$, although MERRA-2 was previously shown to overestimate sea-salt concentrations (Buchard et al., 2017; Provençal et al., 2017), accounting for some of this

discrepancy. Overall, comparisons of CMIP6 models with the MERRA-2 reanalysis product show biases across Europe and North America that are consistent with the comparison to ground-based observations. Additionally, similar comparisons are shown in annual mean cycles across other regions, for which appropriate ground based data is lacking.

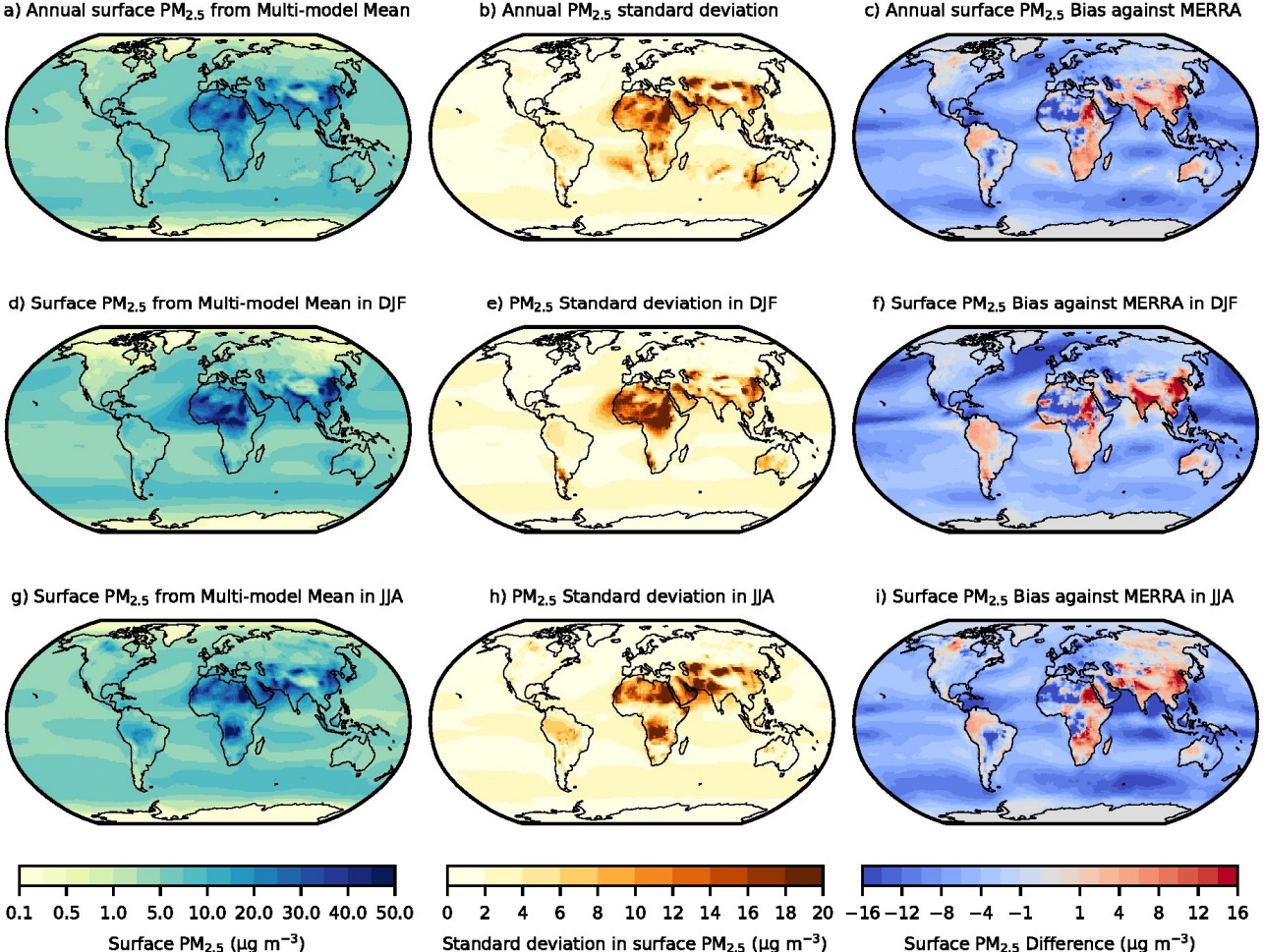

**Figure 7 – Multi-model (11 CMIP6 models) annual and seasonal mean surface PM$_{2.5}$ concentrations in a) annual mean, d) December January, February (DJF) and g) June, July, August (JJA) over the 2005-2014 period. The standard deviation in the multi-model mean in b) annual mean, e) DJF and h) JJA. The difference between the multi-model mean and MERRA-2 reanalysis for c) annual mean, f) DJF and i) JJA.**

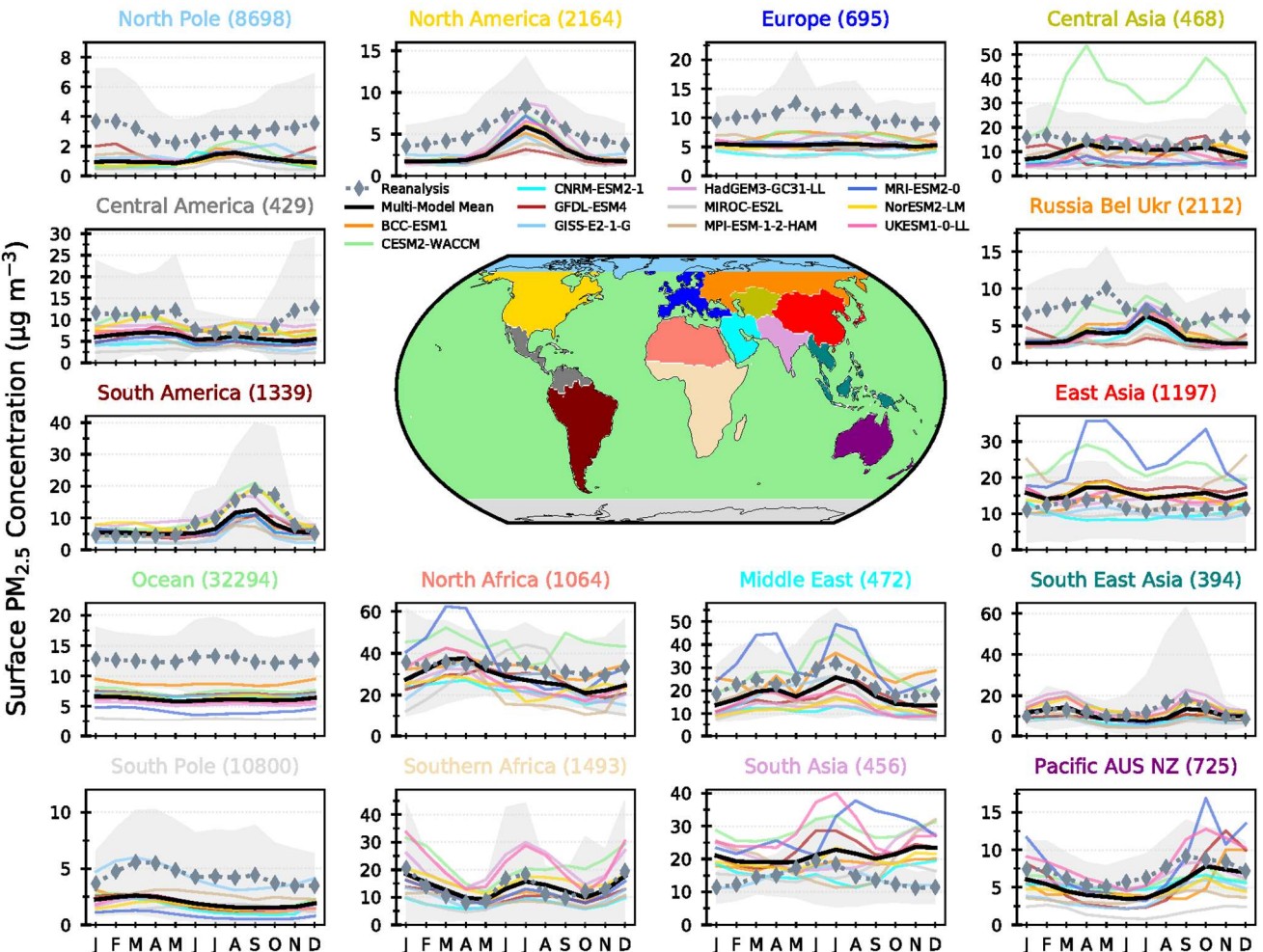

**Figure 8 – Individual and multi-model (11 CMIP6 models) monthly mean surface PM₂.₅ concentrations across different world regions compared with the regional monthly values from the PM₂.₅ MERRA-2 reanalysis within the region for the period 2005-2014. The number of reanalysis points within the region is shown in parenthesis. The shading shows variability in the values of the MERRA-2 reanalysis products across the region.**

## 4 Air Pollutants from Pre-Industrial to Present-day

### 4.1 Surface Ozone

The simulated changes in surface $O_3$ across 6 CMIP6 models and the HTAP_param are shown in Figure 9 and S14-S15 over the historical period of 1850 to 2014. The CMIP6 multi-model mean shows that global annual mean surface $O_3$ has increased by 11.7 +/- 2.3 ppb since 1850 (+/- 1 standard deviation), although the change could be as large as 14 ppb (from BCC-ESM1) or as little as 7 ppb (from UKESM1-0-LL). Globally and over most regions there has been a larger historical increase in surface $O_3$ in JJA than in DJF (Figure S16). The 1850 to 2000 multi-model annual mean change in surface $O_3$ from the CMIP6 models of 10.6 ppb is in good agreement with the 10 +/- 1.6 ppb simulated by the CMIP5 models used in ACCMIP (Young et al., 2013). An evaluation of the long-term changes in surface $O_3$ over the historical period simulated by the CMIP6 models at specific measurement locations is presented separately in the tropospheric $O_3$ CMIP6 companion paper of Griffiths et al., (2020). This shows that CMIP6 models can reasonably represent long term changes in surface ozone since the 1960s, providing a degree of confidence in the future projections of changes in the CMIP6 scenarios. However, long term changes in simulated surface $O_3$ from the previous generation of global coupled chemistry-climate models (used in CMIP5) were found to underestimate the observed trend at northern hemisphere monitoring locations (Parrish et al., 2014). Further comparisons of historical surface $O_3$ simulated by CMIP6 models with long-term historical observations is outside the scope of the current work but will be the subject of future research.

A large diversity in the simulated historical changes is shown across the different regions analysed here, with UKESM1-0-LL tending to simulate the smallest historical change and GISS-E2-1-G or BCC-ESM1 the largest. The large diversity across CMIP6 models in the surface $O_3$ response over the historical period can be attributed to the different magnitude of simulated $O_3$ concentrations in the 1850 period (Figure S14) and the rate of change in regional mean $O_3$ concentrations (Figure S15), which is related to the different chemical sensitivity of $O_3$ formation in each model to changing NOx concentrations (Figure S17). Larger differences between CMIP6 models are shown in the DJF mean historical changes over northern hemisphere regions than occurred in JJA (Figure S16), reflecting the differences shown in the model evaluation (Fig. 4) and the strong seasonality of the changes. Even, though the historical surface $O_3$ response is small in UKESM1-0-LL, it is shown to have larger tropospheric changes in $O_3$ over the historical period compared to other CMIP6 models (Griffiths et al., 2020).

South Asia is the region with the largest diversity in simulated historical changes in surface $O_3$ of between 16 and 40 ppb, with a larger range in DJF (10-40 ppb) than in JJA (19-36 ppb). The large diversity in CMIP6 models is attributed to the large differences in simulated NOx concentrations, and hence chemical sensitivities of $O_3$ formation, occurring across South Asia over the historical period (Figure S17). In addition, the large historical change in $PM_{2.5}$ over this region (Fig. S18) could alter the heterogeneous loss rate of radicals to aerosols and therefore also affect $O_3$ formation. Surface $O_3$ is simulated to have increased by between 10 to 30 ppb on an annual mean basis and by a larger amount in JJA (12 to 37 ppb) over the major northern anthropogenic source regions since 1850, driven mainly by the large increases in anthropogenic precursor emissions of $CH_4$, NOx, CO, and NMVOCs over this period.

A qualitative estimate of the influence of non-emission driven processes (chemistry and climate change) can be ascertained by comparing results from the HTAP_param, an emission-only driven model, to those of the CMIP6-models. Simulated historical changes in surface $O_3$ from UKESM1-0-LL are comparable to those from the HTAP_param, indicating that the magnitude of change simulated by UKESM1-0-LL is similar to that solely from changes in precursor emissions. However, the global annual mean surface $O_3$ response of 7.6 +/- 0.7 ppb from HTAP_param over the historical period is 4.1 ppb lower than the CMIP6 multi-model mean, indicating globally that non-emission driven processes have contributed to approximately 30% of the change in surface $O_3$, although this contribution varies regionally. The different magnitude of response across models could be due to non-emission driven process, e.g. from different chemistry schemes and climate change signals within models.

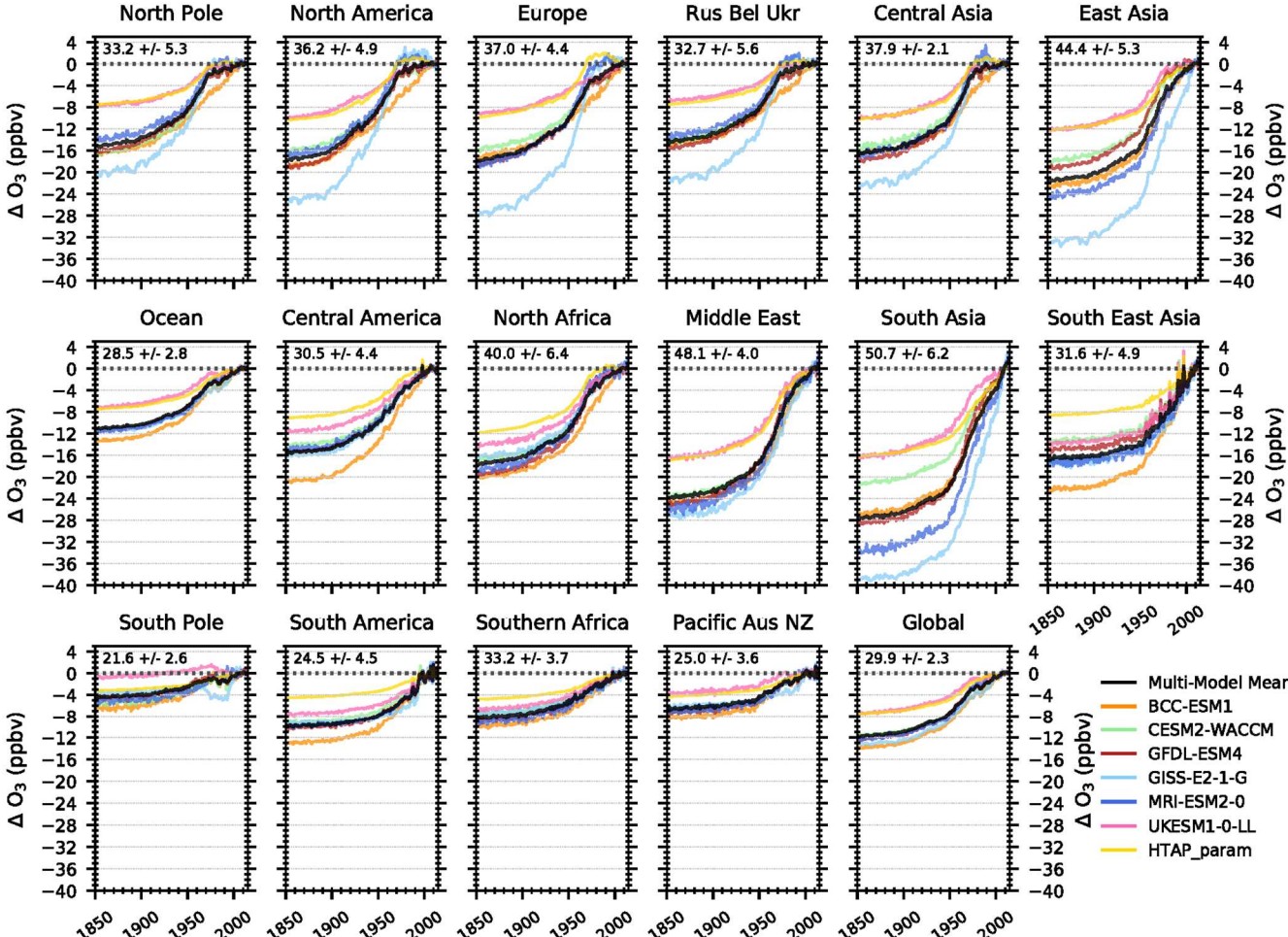

**Figure 9 – Changes in the regional and global annual mean surface O$_3$ concentrations, relative to a 2005-2014 mean value, across 6 CMIP6 models and the HTAP_param. The multi-model annual mean year 2005-2014 surface O$_3$ concentrations (+/- 1 standard deviation) are shown in the top left of each panel. Regions are defined in Figure S1.**

### 4.2 Surface PM$_{2.5}$

The simulated change in annual mean surface PM$_{2.5}$ across 11 CMIP6 models is shown in Figure 10 over the historical period
of 1850 to 2014. Since 1850, CMIP6 models simulated an increase in global annual and seasonal mean surface PM$_{2.5}$ concentrations of <2 µg m$^{-3}$ (15-20%). Larger regional increases of surface annual mean PM$_{2.5}$ of up to 12 µg m$^{-3}$ are simulated across South and East Asia, with changes in DJF (up to 21 µg m$^{-3}$) larger than those in JJA (up to 12 µg m$^{-3}$) (Fig. S16), reflecting the strong seasonality of PM$_{2.5}$ concentrations in these regions. The historical increase in surface PM$_{2.5}$ is primarily driven by the large increase in anthropogenic aerosol and aerosol precursor emissions over the 1850-2014 period (Hoesly et
al., 2018). The largest model diversity is also exhibited over the Asian regions with variations in the response between models of up to 50%, with larger differences between models in DJF than JJA (Figure S16), reflecting the differences shown in the present day model evaluation (Fig. 6). The inter-model differences can be attributed to the different simulation of historical changes in the anthropogenic components sulphate, black carbon and organic aerosols (Figure S18). The largest interannual variability in surface PM$_{2.5}$ concentrations occurs over the North African and Middle East regions as they are located near
large sources of dust, whose emissions are highly dependent on meteorological fluctuations (wind speed). Over Europe, and to a lesser extent Russia, Belarus, Ukraine and North America, the increase in surface PM$_{2.5}$ concentrations since 1850 peaked in the 1980s at 4 µg m$^{-3}$ above the 2005-2014 mean value before decreasing over the last 30 years. There is limited long-term multi-decadal observational data available to assess changes in aerosols simulated by global models. Previous studies using long-term data since the 1980s, mainly over Europe and North America, have found that global models are able to reproduce
the observed multi-decadal changes in aerosols relatively well (Pozzoli et al., 2011; Leibensperger et al., 2012; Tørseth et al.,

2012; Chin et al., 2014; Turnock et al., 2015; Aas et al., 2019). More recently, global composition models, including some CMIP6 models, were shown to be able to reproduce the observed changes in AOD, sulphate and particulate matter over the last two decades (Mortier et al., 2020). The ability of global composition models to reproduce historical changes in aerosols provides a degree of confidence in the future projections under the CMIP6 scenarios. Further model observational comparisons of multi-decadal changes in aerosols will need to be undertaken to improve the understanding of changing aerosol properties and processes.

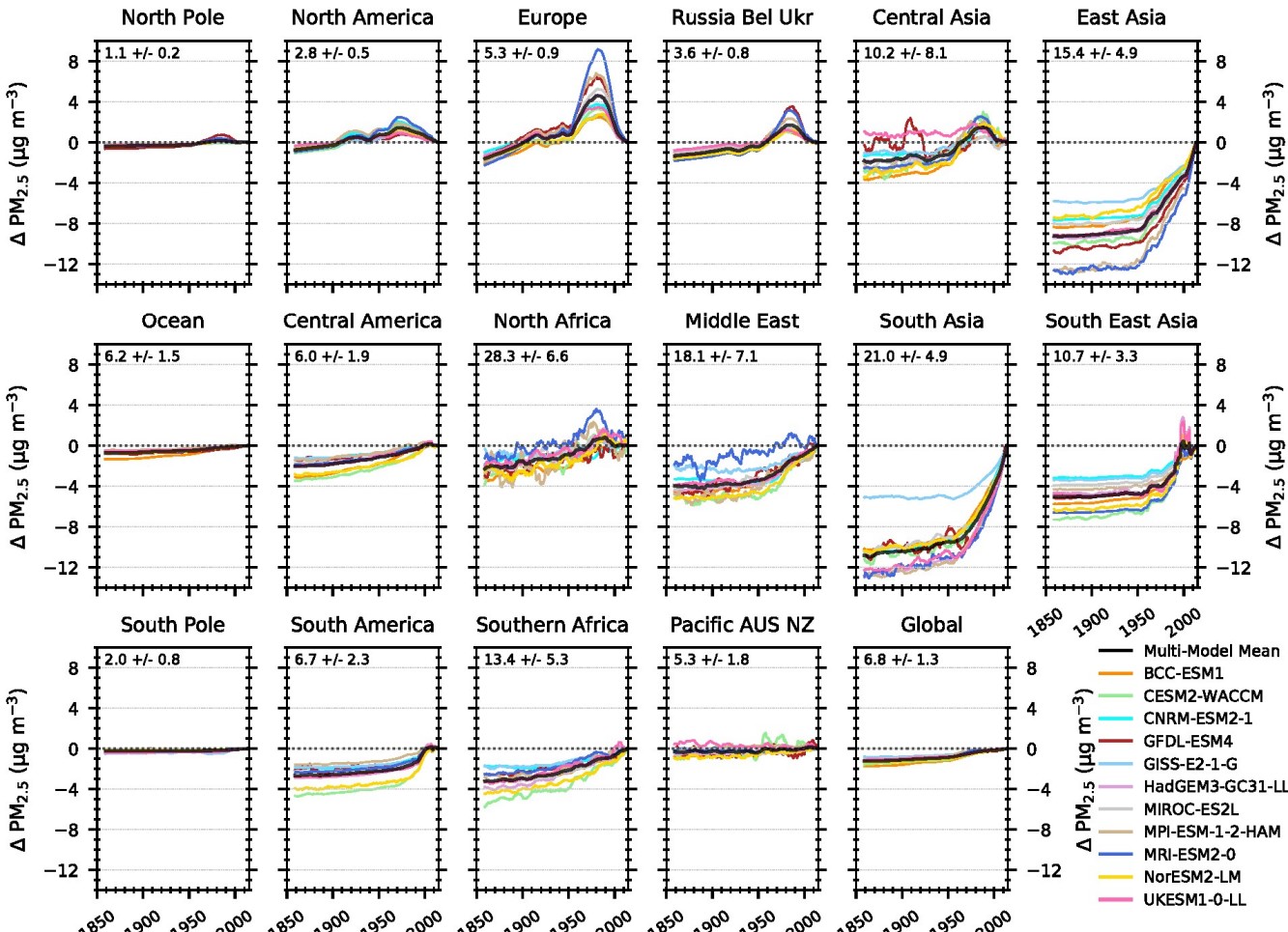

**Figure 10 – Changes in the regional and global annual mean surface PM₂.₅ concentrations, relative to a 2005-2014 mean value, across 11 CMIP6 models. Changes for each region are computed as 10 year running means over the historical period. The multi-model mean 2005-2014 surface PM₂.₅ concentrations (+/- 1 standard deviation) are shown in the top left of each panel. Regions are defined in Figure S1.**

## 5 Air Pollutants from Present-day to 2100

An analysis is now made of the future projections of air pollutants in the CMIP6 Tier 1 scenarios, including ssp370-lowNTCF. A comparison is made of the projected future changes in 2050 and 2100 from the four CMIP6 models (CESM2-WACCM, GFDL-ESM4 and UKESM1-0-LL for both $O_3$ and PM₂.₅, along with BCC-ESM1 for $O_3$ and MIROC-ES2L for PM₂.₅) that had the most data available for the ssp370 scenario.

### 5.1 Surface Ozone

Global annual mean surface $O_3$ is reduced by more than 5 +/- 1.2 ppb (+/- 1 standard deviation value of the multi-model mean) in the near-term (2050) and by 9 +/- 1.6 ppb in 2100 in the strong air pollutant and climate mitigation scenario ssp126 (Figure 11). Smaller reductions in global annual mean surface $O_3$ are predicted for the middle of the road pathway (ssp245) of 4 +/- 1.7 ppb by 2100. Whereas for the weak climate and air pollutant mitigation scenario ssp370, a global annual mean increase in

surface $O_3$ of 1.6 +/- 0.9 ppb in 2050 and 0.6 +/- 1.0 ppb is predicted by 2100. However, implementing strong emission controls for SLCFs on top of a weak climate mitigation scenario (ssp370-lowNTCF) shows that previous increases in global annual mean surface $O_3$ can be substantially reduced to values that are 2.5 +/- 0.5 ppb below the 2005-2014 mean value in 2050, with

benefits to air quality and climate (Allen et al., 2020). For ssp585, which has weak climate mitigation measures but strong air pollution controls, a near-term increase in global annual mean surface $O_3$ of 1.4 +/- 0.8 ppb is predicted in 2050 but by 2100 surface $O_3$ reduces by 2.7 +/- 1.5 ppb, relative to 2005-2014, due to the implementation of air pollutant controls in the latter half of the 21st Century.

The global response in annual mean surface $O_3$ concentrations to the different scenarios is also repeated across the different

world regions, albeit with differing magnitudes. In ssp370 increases in annual mean surface $O_3$ are predicted to occur across North America (+1.6 ppb), Europe (+5.4 ppb) and East Asia (+5.9 ppb), with the largest increase predicted in South Asia of 15.1 +/- 9.6 ppb by 2100. Despite the reductions in $O_3$ precursor emissions across North America, Europe and East Asia by 2100 (Fig. 2) surface $O_3$ concentrations have continued to increase up to the end of this period, indicating the importance of future changes in chemistry, global $CH_4$ abundances and climate on the response of surface $O_3$ in ssp370 (Wild et al., 2012;

Gao et al., 2013; Rasmussen et al., 2013; Young et al., 2013; Colette et al., 2015; Fortems-Cheiney et al., 2017; Li et al., 2019; Turnock et al., 2019). South Asia shows the largest increase in surface $O_3$ as precursor emissions are anticipated to increase across this region on top of the large climate change signal and growth in $CH_4$ abundance. Additionally, the largest diversity in projections between the CMIP6 models is shown over South Asia, indicating that there is some disagreement between the models as to the magnitude and extent of changes over this region. Surface $O_3$ across oceanic regions (background) are

predicted to remain at or near current values in ssp370 due to the increases in water vapour in a warming world leading to more $O_3$ destruction (Johnson et al., 1999; Doherty et al., 2013). The impact of more aggressive near-term reductions to emissions of SLCFs (but not $CH_4$) on top of the ssp370 pathway is shown by the smaller changes in the ssp370-lowNTCF (Fig. 11 and Figures S19-S20 for individual models). In this pathway surface $O_3$ concentrations are reduced globally and across most regions to be at or near 2005-2014 values, a substantial benefit to surface $O_3$ air quality compared to ssp370. Surface $O_3$

concentrations are predicted to have almost halved by 2050 across South Asia in ssp370-lowNTCF. However, across East Asia the additional precursor emission reductions in ssp370-lowNTCF have resulted in smaller benefits to surface $O_3$ concentrations being simulated by the CMIP6 models than in other regions (Figure S20), which is attributed to an increase in surface $O_3$ concentrations over Eastern China (a part of the larger East Asian region shown in Fig. S1). This increase in surface $O_3$ results from the slight increase in NMVOC emissions (Fig. 2) and a reduction in the NOx titration of $O_3$ due to the large decreases in

NOx emissions in ssp370-lowNTCF. In addition, a reduction in the heterogeneous loss of radicals due to decreases in $PM_{2.5}$ concentrations in ssp370-lowNTCF could also lead to increased surface $O_3$ concentrations (Li et al., 2019).

Surface $O_3$ concentrations predicted across northern hemisphere regions in ssp585 are similar to ssp370 due to comparable changes in air pollutant emissions and climate change. However, a notable exception is a reduction in surface $O_3$ across regions towards the latter half of the 21st Century (post 2080) when there are additional reductions in precursor emissions and global

$CH_4$ abundances by 2100. Surface $O_3$ shows a slightly slower increase until the mid-21st Century over South Asia in ssp585 than occurred in ssp370. This can be attributed to a slightly different temporal evolution of NOx emissions over this region, in that they peak earlier (by 2040) and decline more rapidly in ssp585, when compared to the continual increase in NOx emissions in ssp370 (Fig. 2), which results in a different response of $O_3$ formation within CMIP6 models, In addition, there are more CMIP6 models with data available for ssp370 (6 models) than ssp585 (4 models) (Table 1), which could affect the multi-model

mean response shown in Fig. 11.

The future scenario ssp245 (middle-of-the-road) predicts annual mean surface $O_3$ concentrations that tend to remain at or near the 2005-2014 mean values by 2100 across the major anthropogenic source regions of the Northern Hemisphere, whereas for other tropical and southern hemisphere regions surface $O_3$ concentrations are reduced by more than 4 ppb. The changes in ssp245 are driven by larger precursor emission controls, a smaller climate change signal and controlling $CH_4$ so that global

abundances are below 2015 values by 2100 (Fig. 1g). In ssp245 a near-term (up to 2040) increase in surface O₃ is shown across East Asia and South Asia, which could be attributed to the peaking of global CH₄ abundances at this point, prior to then reducing.

The Tier1 future scenario with the strongest climate and air pollutant mitigation measures, ssp126, shows substantial decreases in surface O₃ concentrations across most regions due to the large reduction in precursor emissions, global CH₄ abundances, 
and small climate change signal. Reductions in surface O₃ of more than 10 ppb are predicted across anthropogenic emission source regions of the northern hemisphere, with smaller reductions across southern hemisphere regions.

Projections from the CMIP6 models show that to achieve global benefits for regional surface O₃ it is important to control O₃ precursor emissions (including CH₄) in addition to limiting future climate change. However, scenarios with large climate change signals (ssp370 and ssp585) but different post 2050 controls on O₃ precursors (most notably CH₄ and NOx), show 
different long-term changes in regional surface O₃ concentrations, which could have important consequences for any potential human health impacts.

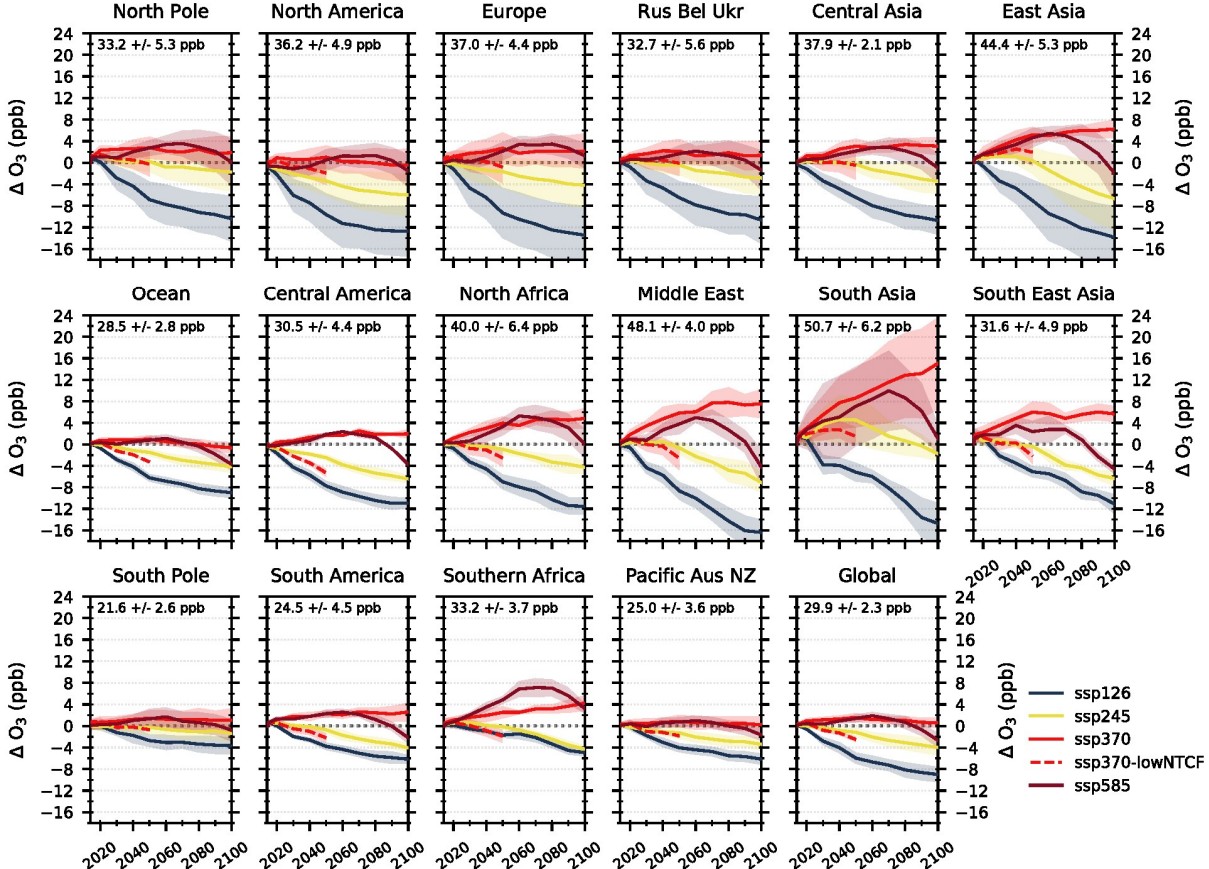

**Figure 11 – Future global and regional changes in annual mean surface O₃, relative to 2005-2014 mean, for the different SSPs used in CMIP6. Each line represents a multi-model mean across the region with shading representing the +/- 1 standard deviation in the** 
**mean. See Table 1 for details of models contributing to each scenario. The multi-model regional mean value (+/- 1 standard deviation) for the year 2005-2014 is shown in the top left corner of each panel.**

A more detailed comparison of future surface O₃ projections between CMIP6 models has been undertaken for ssp370, as it is the scenario with the largest number of available models (Table 1). The regional change in decadal annual and seasonal mean surface O₃, relative to 2005-2014, in 2050 (2045 - 2055 mean) and 2095 (2090 – 2100 mean) for ssp370 from four CMIP6 
models and the HTAP_param is shown in Figure 12. An analysis of the relationships, in terms of correlation coefficients, between future annual mean surface O₃ concentrations and other variables (CH₄ concentrations, surface air temperature, NOx concentrations, emissions of BVOCs and anthropogenic emissions of NMVOCs) is undertaken for CMIP6 models in the ssp370 scenario (Figure 13). Discrepancies in the simulated response of background O₃ across the ocean region (also South Pole and Pacific, Australia and New Zealand) are noticeable between individual models, with UKESM1-0-LL predicting a

decrease in surface $O_3$ compared to the small increase from the HTAP_param and most other models in both 2050 and 2095 (Figure S19). The future surface $O_3$ response in UKESM1-0-LL over the ocean region exhibits a large negative correlation with surface temperature changes (Figure 13), indicating the importance of future climate change in this model over remote regions. UKESM1-0-LL is a model with high equilibrium climate sensitivity (ECS, 5.4 K) compared to other CMIP6 models (Forster et al., 2019; Sellar et al., 2019), and therefore will exhibit a larger climate response (surface temperature and water vapour), leading to enhanced background $O_3$ destruction via water vapour and the hydroxyl radical (OH). Over the North Pole region all models show surface $O_3$ increases that are larger than the HTAP_param, with a larger increase in DJF than JJA. The large future temperature response over the Arctic, as well as changes in NOx concentrations and emissions of NMVOCs are particularly important drivers of surface $O_3$ changes across most CMIP6 models in this region with comparatively low local emissions (Figure 13).

Differences in the predicted surface $O_3$ between models exist across South Asia where CESM2-WACCM (and BCC-ESM1 in 2050) predict a response that is twice as large as UKESM1-0-LL and GFDL-ESM4. The lower annual mean response over South Asia in UKESM1-0-LL and GFDL-ESM4 is driven by a reduction in DJF in these models (Fig. S21), which results in the DJF change in 2050 being lower than the 2005-2014 annual mean value (Fig. 12). The large increase in NOx emissions in ssp370 over South Asia (~80%) has resulted in areas of NOx titration, particularly in DJF, near the Indo-Gangetic plain in both UKESM1-0-LL and GFDL-ESM4, reducing surface $O_3$ concentrations (Fig. S19 and S21). This strong feature of NOx titration of $O_3$ in DJF is absent in both CESM2-WACCM and BCC-ESM1, resulting in larger $O_3$ production over South Asia. The comparison in Fig. 12 shows how the $O_3$ chemistry within models responds differently across a particular area in a future scenario with a large climate change signal and over a region with large increases in local precursor emissions, but that all the drivers related to regional $O_3$ change in South Asia are similarly important across all models (Figure 13).

Over South America and Southern Africa, particularly the tropical areas (Fig. S19), larger future changes in surface $O_3$, particularly by 2100, are predicted by GFDL-ESM4 and UKESM1-0-LL than by CESM2-WACCM. These changes over South America are larger in JJA in all models, with small seasonal differences over Southern Africa. Over this region, biogenic emissions (particularly isoprene) are an important source of $O_3$ formation. Discrepancies in the future response of these BVOC emissions between models could be occurring due to the differing magnitudes of climate and land-use change and how they are coupled within individual CMIP6 models (Table S1), which could affect future surface $O_3$. Future changes in the total emissions of BVOCs) and those solely from isoprene obtained from five CMIP6 models (Figure S22 and S23) show that CESM2-WACCM has larger total BVOC emissions over the period 2005-2014 (due to the inclusion of more BVOCs), which then increase in the future ssp370 scenario, along with isoprene emissions, resulting in a smaller increase (and even decrease in some parts of the region) in $O_3$. Whereas, UKESM1-0-LL shows a larger increase in $O_3$ and a reduction in BVOC emissions, mainly from isoprene (Fig. 23), over parts of South America and tropical Africa. Figure 13 shows that there are differing relationships between future surface $O_3$ concentrations, BVOC emissions and NOx concentrations across CMIP6 models over South America and Southern Africa. Over Southern Africa, UKESM1-0-LL shows a different relationship between BVOC emissions and surface $O_3$ concentrations than other CMIP6 models, indicating that this could be leading to the different future $O_3$ response in this model over this region. Similarly, Figure 13 shows that over South America, CESM2-WACCM has a different relationship between surface $O_3$ and the variables considered here than in other CMIP6 models, particularly for BVOCs, leading to the different future responses in this model over this region. Figure 13 shows that there are differences between models in the surface $O_3$ response over regions such as South America and Southern Africa, which are potentially linked to the land-surface response and are important to understand more in future work.

Whilst there are disagreements between models over some regions, there is also substantial consistency in the predicted increase to annual mean surface $O_3$ in ssp370 over North America, Europe and East Asia, which is larger than that from HTAP_param. However, BCC-ESM1 tends to predict a larger increase than the other three models, potentially due to the coarser resolution of this ESM. There are differences in simulated seasonal response across these regions, with all models

showing a smaller increase in JJA than DJF across North America and Europe, whilst across East Asia there tends to a be a larger future surface O₃ increase in JJA than DJF. Figure 13 shows that there is a negative correlation between surface O₃ and NOx concentrations, as well as between O₃ and NMVOC emissions, for most CMIP6 models across these regions, reflecting that as most anthropogenic precursor emissions (including NOx) decrease in this scenario (Fig. 2) then surface O₃ is simulated to increase. An exception to this is across East Asia, where the increase in NMVOC emissions in ssp370 (Fig. 2) are positively correlated with surface O₃, indicating different chemical drivers of future O₃ across this region. In addition, there are positive correlations between the other variables (temperature, CH₄ and BVOCs) for most CMIP6 models indicating that changes in climate and global CH₄ abundances are also important drivers of surface O₃ increases over these regions.

The differences between the individual CMIP6 models highlight the importance of further understanding how future O₃ chemistry is affected by changes to precursor emissions and climate. The predicted differences in models can be quite pronounced over regions like South Asia where changes in one model can be double that of another model, which could have important consequences for the ability of models to simulate future regional air quality.

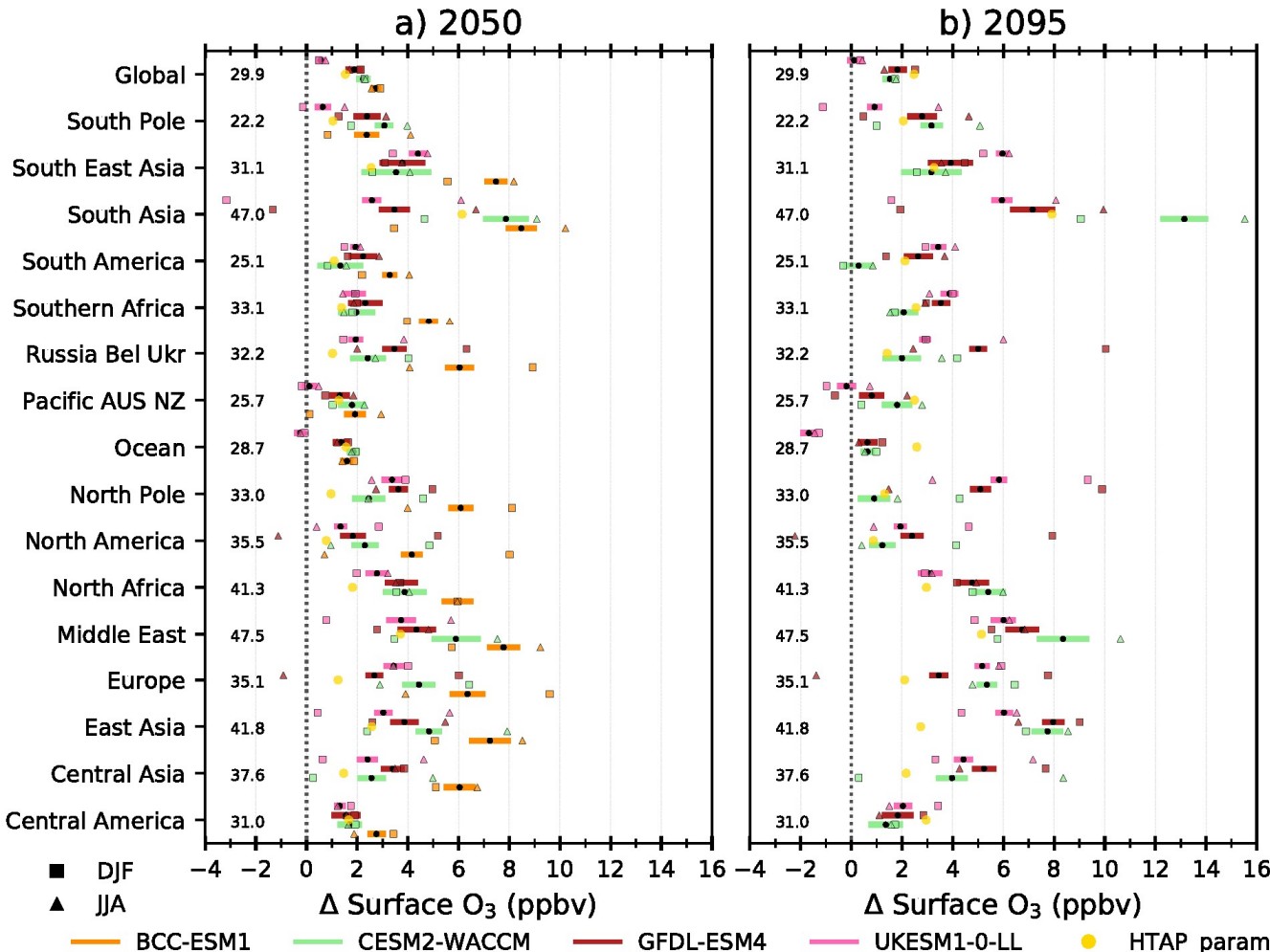

**Figure 12 – Future global and regional changes in the decadal annual and seasonal mean surface O₃, relative to the 2005-2014 mean, for the ssp370 pathway used in CMIP6. Each black circle represents the decadal annual mean response for an individual model in a) 2045-2055 and b) 2090-2100, with the coloured bars showing the standard deviation across the decadal annual mean. The DJF and JJA seasonal mean response averaged over the relevant 10 year period is shown by squares and triangles respectively. The multi-model regional mean over the period 2005- 2014 is given towards the left of each panel. The response from the HTAP_param in each time period is shown by the separate gold circle.**

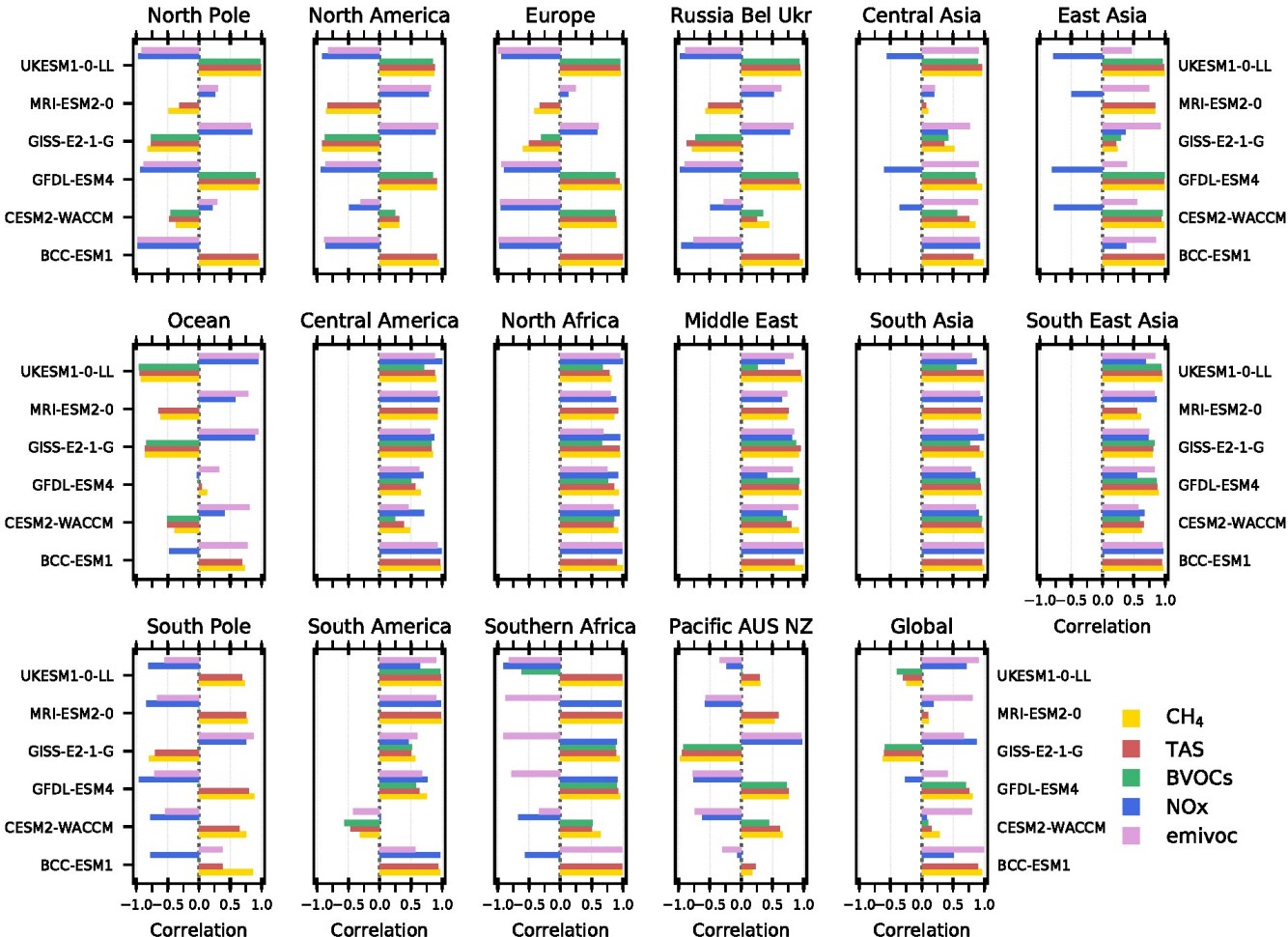

**Figure 13 – Correlation coefficients calculated when comparing future annual mean surface O₃ concentrations against individual variables of surface CH₄ concentrations, surface air temperature (TAS), emissions of biogenic volatile organic compounds (BVOCs), NOx (NO + NO₂) concentrations and anthropogenic emissions of non-methane volatile organic compounds (NMVOCs) from individual CMIP6 models over the period 2015 to 2100 in the ssp370 scenario.**

### 5.2 Surface PM₂.₅

Relatively small global changes in annual mean surface PM₂.₅ are predicted for all CMIP6 models across all scenarios (Figure 14), with an increase in ssp370 and a reduction in the others. Small reductions in PM₂.₅ are predicted for all scenarios across Europe (0.3 to 3 μg m⁻³) and North America (0.0 to 1.3 μg m⁻³) due to the reduction in aerosol and aerosol precursor emissions. Differences in PM₂.₅ between scenarios are highlighted across a number of regions.

For the weak climate and air pollutant mitigation scenario ssp370, increases in annual mean surface PM₂.₅ are predicted across South Asia (7.3 +/- 4.1 μg m⁻³ by 2050 and 3.1 +/- 3.1 μg m⁻³ by 2100), South East Asia (2.7 +/- 4.7 μg m⁻³ by 2100), Southern Africa (1.6 +/- 3.7 μg m⁻³ by 2100), Central (2.8 +/- 3.2 μg m⁻³ by 2100) and South America (2.9 +/- 3.6 μg m⁻³ by 2100). The increases in PM₂.₅ are driven mainly by the increase in aerosol and aerosol precursor emissions in this scenario (Fig. 2), shown by the positive correlations between emissions and surface PM₂.₅ in CMIP6 models across these regions (Figure 16). However, there is a degree of uncertainty associated with all of these future projections indicated by the large diversity across the CMIP6 models. Some of the largest predicted increases in surface PM₂.₅ occur across South Asia in ssp370, a region already with high present day PM₂.₅ concentrations. The increase in PM₂.₅ peak in 2050 across this region, which coincides with the increase of SO₂, BC and OC emissions, before declining to 2100 when emissions reduce. Over East Asia, annual mean PM₂.₅ concentrations are simulated to remain at or near 2005-2014 values until the latter half of the 21ˢᵗ Century when the decrease in emissions reduce PM₂.₅ concentrations by 2.5 +/- 2.7 μg m⁻³. The impact of reductions in SLCFs on top of the ssp370 scenario act to constrain any increases of PM₂.₅ concentrations to near present day values across most regions. However, substantial reductions in PM₂.₅ concentrations of 5.6 +/- 2.0 μg m⁻³ and 5.3 +/- 2.1 μg m⁻³ below 2005-2014 values are achieved

650    by 2050 across East and South Asia respectively, by implementing these measures. Due to the short lifetime of aerosols in the atmosphere $PM_{2.5}$ concentrations respond rapidly to the large cuts in emissions that occur in ssp370-lowNTCF and show the benefits to targeting these emissions, although there could be a potential climate impact (Allen et al., 2020).

Reductions in annual mean surface $PM_{2.5}$ are simulated across all regions for ssp126, ssp245 and ssp585. Differences exist in the magnitude and timing of $PM_{2.5}$ reductions across regions linked to the changes in emissions. The largest reductions in

655    $PM_{2.5}$ occur over South Asia in 2100 and range from 11.1 +/- 2.8 $\mu g$ $m^{-3}$ in ssp126 to 8.6 +/- 2.9 $\mu g$ $m^{-3}$ in ssp585, a substantial benefit to regional air quality. Similar benefits to $PM_{2.5}$ are achieved over East Asia by 2100 although the more rapid improvements occur over this region in the first part of the 21$^{st}$ Century.

The response of $PM_{2.5}$ concentrations is more variable, with a larger diversity across CMIP6 models within regions that are close to natural aerosol emission sources. This is particularly noticeable over North Africa where the variability across CMIP6

660    models in dust emissions from the Saharan source region (Fig. S8) results in an uncertain $PM_{2.5}$ response across this region. A similar response is also exhibited across the Middle East and Central Asia. The potential influence of BVOCs on SOA formation (Fig. S22 and S26) could also be contributing to the diversity in the CMIP6 model responses across the South America and Southern Africa regions.

The CMIP6 models show that future reductions in aerosols and aerosol precursors will lead to a decrease in surface $PM_{2.5}$

665    concentrations across most world regions and a benefit to regional air quality (and human health), consistent with that from CMIP5. However, if emissions are not controlled over economically developing regions such as South America, Asia and Africa then surface $PM_{2.5}$ is anticipated to increase and worsen future regional air quality. Targeting emission reductions of SLCFs in the short-term shows the potential for rapid improvements in surface $PM_{2.5}$ and air quality.

670

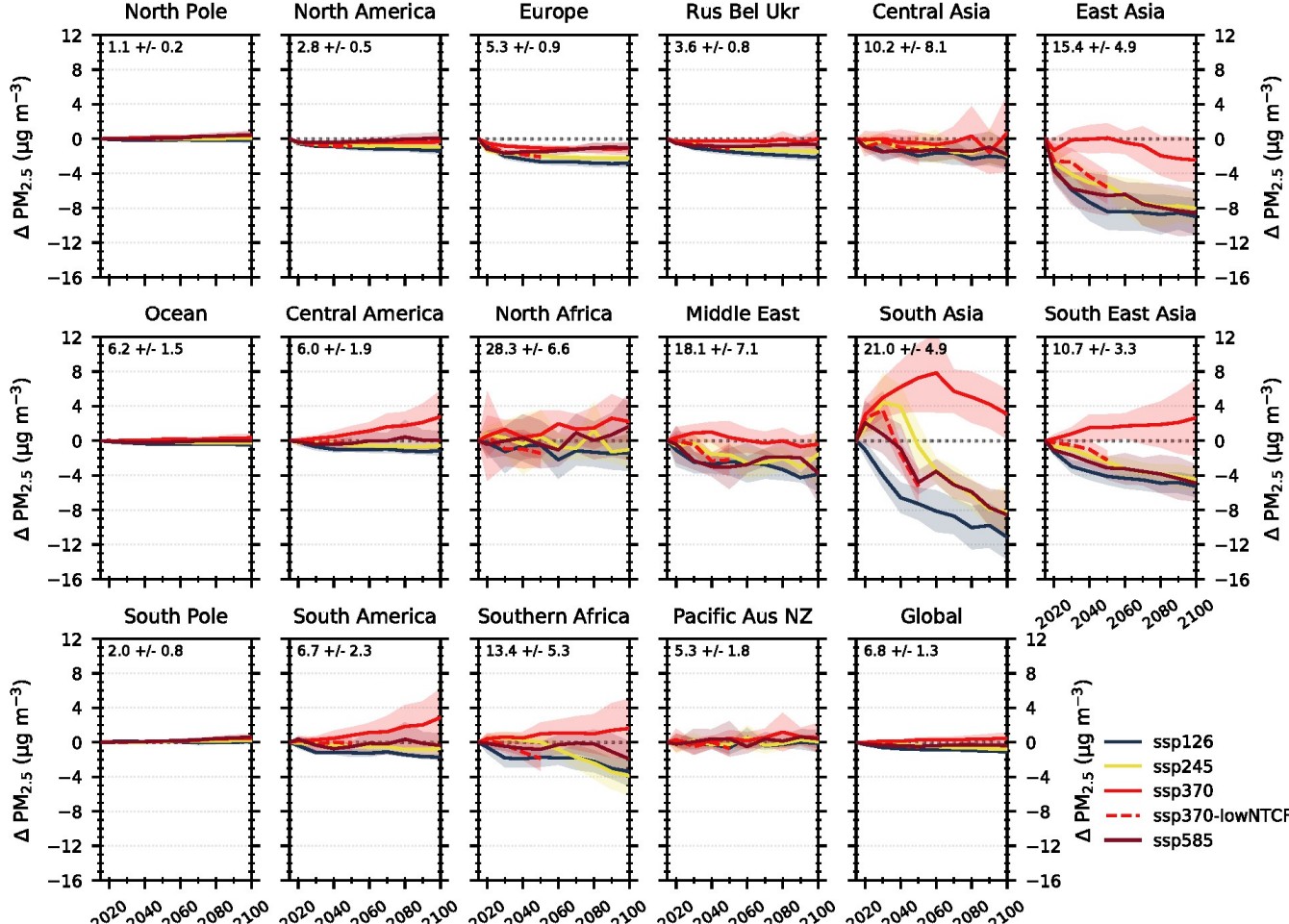

**Figure 14 – Future global and regional changes in annual mean surface PM$_{2.5}$, relative to 2005-2014 mean, for the different SSPs used in CMIP6. Each line represents a multi-model mean across the region with shading representing the +/- 1 standard deviation in the mean. See Table 1 for details of models contributing to each scenario. The multi-model regional mean value (+/- 1 standard deviation) for the year 2005-2014 is shown in the top left corner of each panel.**

In a similar analysis to that for surface O$_3$, a more detailed comparison has been undertaken of four CMIP6 models predicting changes in annual and seasonal surface PM$_{2.5}$ in 2050 and 2095 under ssp370 (Figure 15). In addition, an analysis of the relationships, in terms of correlation coefficients, between future annual mean surface PM$_{2.5}$ and other variables (total surface precipitation, surface air temperature and emissions of BVOCs, SO$_2$, BC and organic aerosol) has been undertaken for CMIP6 models in the ssp370 scenario (Figure 16). Small reductions in annual mean surface PM$_{2.5}$ concentrations (<2 µg m$^{-3}$) are simulated consistently by all CMIP6 models across North America and Europe in ssp370, with larger reductions simulated in DJF than JJA. The reductions in annual mean PM$_{2.5}$ over Europe and North America are mainly attributed to decreases in the BC and SO$_4$ components (Fig. S24 and S25), as indicated by the strong correlations with BC and SO$_2$ emissions across CMIP6 models (Figure 16). However, by 2095 a small increase (up to 2 µg m$^{-3}$) is simulated in JJA by UKESM1-0-LL and CESM2-WACCM over North America, which could be attributed to changes in climate due to the strong positive correlations in both models for temperature, precipitation and BVOCs (Figure 16).

South Asia, the region with the largest simulated future change in annual mean surface PM$_{2.5}$ of up to 12 µg m$^{-3}$, shows fairly good agreement between three CMIP6 models (UKESM1-0-LL, GFDL-ESM4 and CESM2-WACCM) as projections in 2050 and 2095 are all within the range of each of the individual models. The future increases in annual mean surface PM$_{2.5}$ appear to be strongly driven by emission changes as there are strong positive correlations between these variables across South Asia in all models (Figure 16). Across South Asia, all models simulate a larger increase in DJF mean surface PM$_{2.5}$ concentrations, of up to 18 µg m$^{-3}$ by 2050, than occurs in JJA and reflects the seasonality shown in the model evaluation. The MIROC-ES2L model predicts smaller future increases in surface PM$_{2.5}$ than the other models across South Asia of up to 5 µg m$^{-3}$ in both

2050 and 2095. This is a result of smaller changes in the BC, OA and sulphate aerosol components in the MIROC-ES2L model despite increases in aerosols and aerosol precursor emissions across South Asia in ssp370 (Figure S24-S26).

Disagreements in both the sign and magnitude of simulated future annual and seasonal mean surface $PM_{2.5}$ changes between CMIP6 models are also exhibited across East Asia. Small regional annual mean increases are predicted in 2050 due to $PM_{2.5}$ increases in JJA from all models apart from GFDL-ESM4. A larger reduction in the $SO_4$ component is simulated over East

Asia by GFDL-ESM4 than in other models (Fig S25), resulting in an overall decrease in $PM_{2.5}$. In 2095 most models , simulate a reduction in $PM_{2.5}$ concentrations in both seasons across East Asia, apart from CESM2-WACCM due to the increase in JJA. All models simulate continual reductions out to 2100 for $SO_4$ across this region, whereas BC increases in the near-term before decreasing out to 2100. For OA, CESM2-WACCM shows larger increases over East Asia in both 2050 and 2095 compared to the other models, which show a smaller increase in 2050 and a reduction by 2095 (Fig. S26). CESM2-WACCM includes a

more complex treatment of SOA formation, showing a strong response to climate and historical trends in OA (Tilmes et al., 2019). Positive correlations are shown for CESM2-WACCM between surface $PM_{2.5}$ and emissions of BVOC and temperature (Fig. 16), which are not present in other models and could explain the differences between this model and others across East Asia. The discrepancies in CMIP6 models are not as obvious over South Asia as the effect of the increase in OA over South Asia in CESM2-WACCM is masked by coincident increases in other components across other models, as indicated by the

strong correlations with emissions here. CESM2-WACCM also shows larger simulated increases in $PM_{2.5}$ over South America, Central America, Southern Africa and South East Asia than other models, which can be attributed to the larger increase in the OA fraction (Fig. S26) and the strong correlations in this model with changes in temperature and emissions (BVOCs and $SO_2$). Over Southern Africa UKESM1-0-LL shows a reduction in future $PM_{2.5}$, in contrast to other models, due to a reduction in the BC, OA and dust aerosol components (Fig. S24, S26 and S27). UKESM1-0-LL exhibits particularly strong negative

correlations for surface $PM_{2.5}$ when compared with temperature and precipitation. These relationships over Southern Africa are quite different to other CMIP6 models, which is also highlighted in the model evaluation over this region (Fig. 8) and indicates that climate change influences aerosol concentrations differently over this region in this model (Figure 16). In addition, there is a slight positive correlation of $PM_{2.5}$ with BVOC emissions in UKESM1-0-LL over Southern Africa. Future biogenic emissions (including monoterpenes) reduce here in ssp370 (Fig. S22), potentially due to land-use vegetation change

as UKESM1-0-LL has dynamic vegetation coupled to BVOC emissions (Table S1). This could also reduce $PM_{2.5}$ concentrations over this region because monoterpene emissions are the main precursor to SOA formation in UKESM1-0-LL (Mulcahy et al., 2019).

The decadal annual and seasonal mean $PM_{2.5}$ response is variable across individual CMIP6 models over regions close to natural sources of particulate matter (North Africa, Central Asia and Pacific, Australia and New Zealand). Over these regions there is

a large range in both the sign and magnitude of the annual and seasonal $PM_{2.5}$ response, which can be mainly attributed to the dust fraction (Fig. S27) and the fact that this aerosol source has a large inter-annual variability in its emission strength. There is also a lack of consistency across CMIP6 models in the correlations of $PM_{2.5}$ with any individual driver, indicating the variability of aerosol sources in these regions within models. Interestingly, the CMIP6 models do not agree in the sign and magnitude of future changes to dust concentrations in ssp370 (Fig. S27).

Across the ocean and North Pole regions all the CMIP6 models tend to simulate a small increase in $PM_{2.5}$ concentrations, which can be attributed to increases in sea salt concentrations (Fig. S28). A strong increase in sea salt concentrations is simulated in all models across the Southern Ocean (and other oceans), potentially driven by changes to meteorological conditions (reflected by the positive correlations of $PM_{2.5}$ with the climate variables temperatures and precipitation in Fig. 16), which increase wind speed and sea salt emissions. As ssp370 is a scenario with a large climate change signal, the increases in

$PM_{2.5}$ across the North Pole, particularly in 2100, can be attributed to the melting of sea ice increasing sea salt emissions, which again is reflected in the positive correlations of $PM_{2.5}$ with climate variables over this region. However, the magnitude

of this response is different in the CMIP6 models due to the underlying ECS and the response of Arctic surface temperatures within the individual model.

The differences in the simulated future PM$_{2.5}$ changes across the CMIP6 models in ssp370 highlight that it is important to consider how natural sources of aerosol respond in a future climate in addition to that from changes in anthropogenic emissions. Particular differences between models have been shown for dust, sea salt and also organic (secondary) aerosols, which should be explored further. In addition, the different representations of aerosols within individual models e.g. organic aerosols, are an important consideration as they can make a large difference to any future regional projection of PM$_{2.5}$.

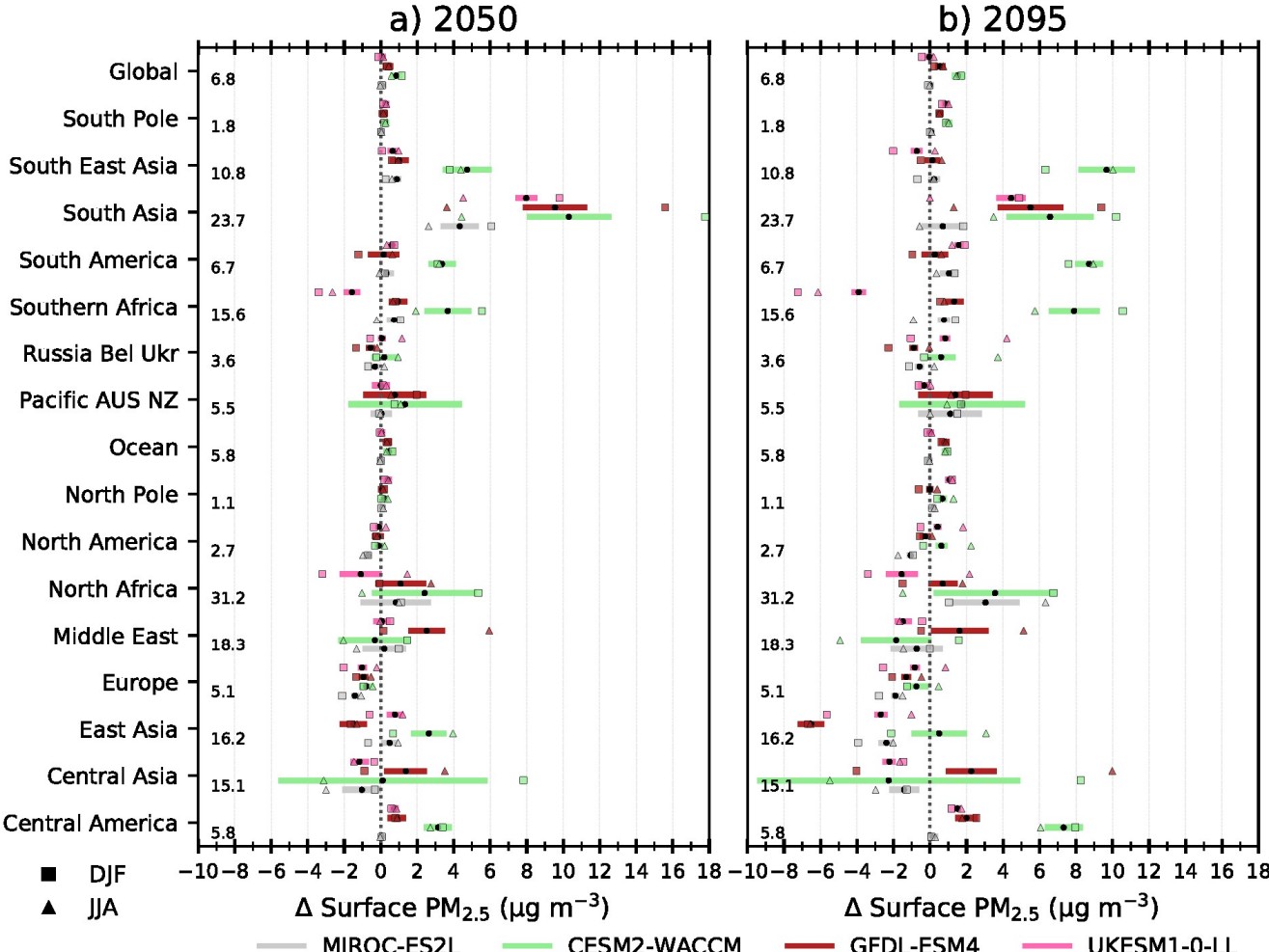

**Figure 15 – Future global and regional changes in the decadal annual and seasonal mean surface PM$_{2.5}$, relative to the 2005-2014 mean, for the ssp370 pathway used in CMIP6. Each black circle represents the decadal annual mean response for an individual model in a) 2045-2055 and b) 2090-2100, with the coloured bars showing the standard deviation across the decadal annual mean. The DJF and JJA seasonal mean response averaged over the 10 relevant period are shown by squares and triangles respectively. The multi-model regional mean over the period 2005- 2014 is given towards the left of each panel.**

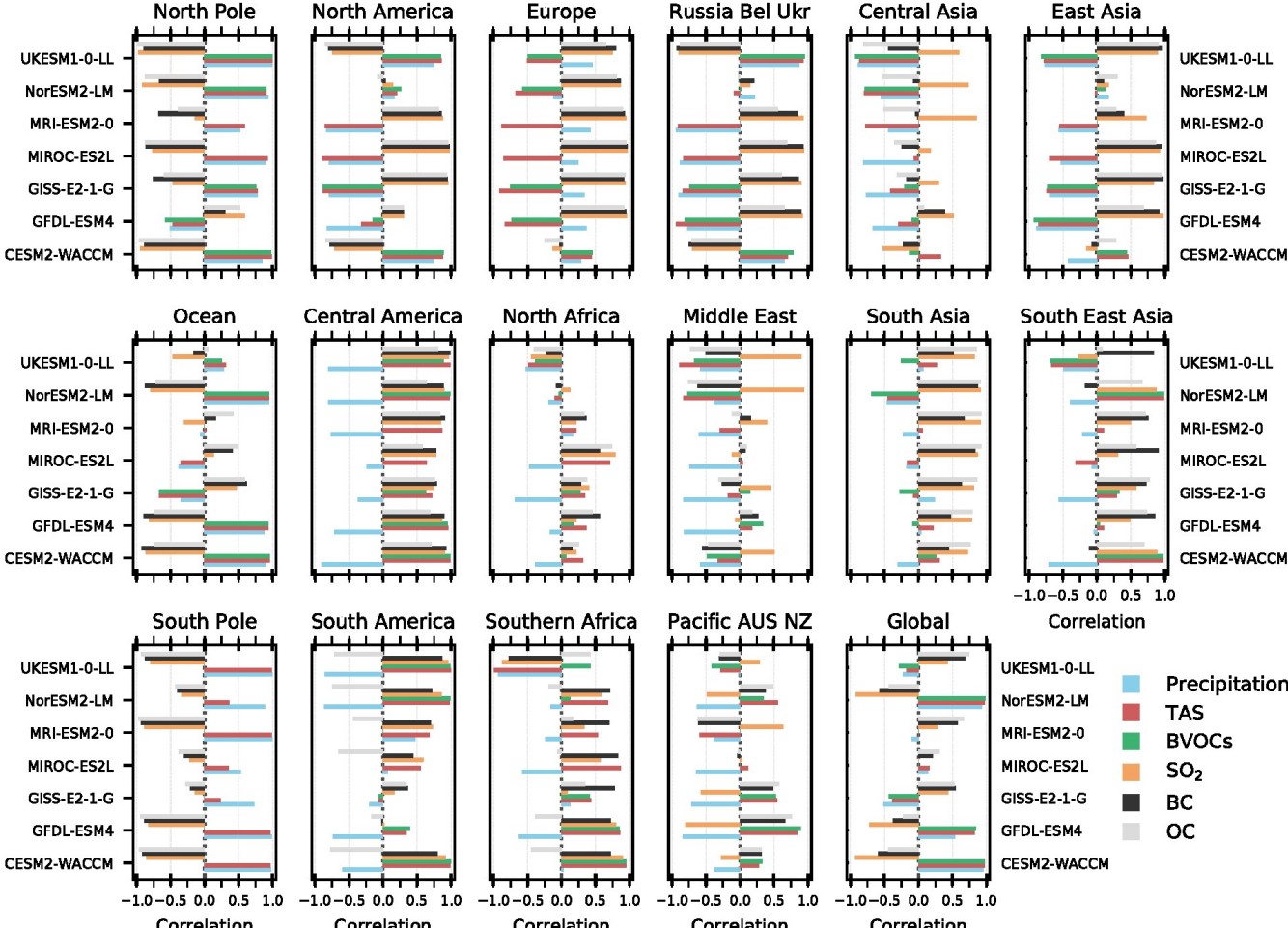

**Figure 16 – Correlation coefficients calculated when comparing future annual mean surface PM$_{2.5}$ concentrations against individual variables of precipitation, surface air temperature (TAS), emissions of biogenic volatile organic compounds (BVOCs) and emissions of SO$_2$, black carbon (BC) and organic carbon (OC) from individual CMIP6 models (that had data out to 2100) over the period 2015 to 2100 in the ssp370 scenario.**

## 6 Conclusions

In this study we have provided an initial analysis of the historical and future changes in air pollutants (O$_3$ and PM$_{2.5}$) from the latest generation of Earth system and climate models that have submitted results from experiments conducted as part of CMIP6. Data was available from the historical experiments of 6 CMIP6 models for surface O$_3$ and 11 models for surface PM$_{2.5}$. Historical changes in regional concentrations of O$_3$ and PM$_{2.5}$ are presented over the period 1850 to 2014 using data from all models. A present day model evaluation of the CMIP6 models was conducted against surface observations of O$_3$ and PM$_{2.5}$ obtained from the TOAR and GASSP databases respectively. An additional comparison was performed for simulated PM$_{2.5}$ concentrations against the MERRA-2 aerosol reanalysis product. An assessment is then made of the changes in surface O$_3$ and PM$_{2.5}$ simulated by the CMIP6 models across different future scenarios, ranging from weak to strong air pollutant and climate mitigation.

The 6 CMIP6 models simulate present day (2005-2014) surface O$_3$ concentrations that are elevated in the Northern Hemisphere summer, with lower values throughout the year across the Southern Hemisphere. However, a large model diversity is shown across the continental Northern Hemisphere due to the large simulated seasonal cycles in certain models. Compared to surface O$_3$ measurements, CMIP6 models overestimate observed annual mean values and in both summer and winter across most regions by up to 16 ppb (a similar result to previous multi-model evaluations of global chemistry-climate models in Young et al., (2018)). An exception to this is at observation locations across Antarctica where CMIP6 models tend to underestimate observed values by 5 ppb.

Large surface PM$_{2.5}$ concentrations are simulated in CMIP6 models near dust and anthropogenic emission source regions. Model diversity across the CMIP6 models is largest near the dust source regions due to their sensitivity to meteorological variability, whereas across other regions the CMIP6 models are relatively similar in their simulation of PM$_{2.5}$ concentrations. Evaluating the approximate PM$_{2.5}$ calculated from CMIP6 models (excluding nitrate aerosols) against ground based PM$_{2.5}$ observations shows an underestimation across most regions of up to 10 μg m$^{-3}$. The underestimation of observations by models is larger in the northern hemisphere winter than summer, in part due to the absence of nitrate aerosols within most CMIP6 models and also due to underrepresentation of other aerosol processes within global models (a similar result to other multi-model assessments). To improve the spatial coverage and consistency of the PM$_{2.5}$ evaluation with CMIP6 models an additional comparison was made to the MERRA-2 aerosol reanalysis product. A similar, but slightly smaller, underestimation of PM$_{2.5}$ concentrations over Europe and North America was found in the comparison of CMIP6 models and MERRA-2, providing further confidence in the result from the ground-based comparison. CMIP6 models overestimated the monthly PM$_{2.5}$ concentrations in MERRA-2 over South and East Asia by up 15 μg m$^{-3}$, in contrast to the evaluation using ground based observations. Mean annual cycles simulated by CMIP6 models and MERRA-2 tend to agree across other regions for which there are no suitable ground-based observations. The comparison of surface O$_3$ and PM$_{2.5}$ simulated by CMIP6 models to observations shows similar biases to previous generations of global composition-climate models. Further studies are required (e.g. global sensitivity or process studies) to explore uncertainties in models and the differences with observations.

Across the historical period (1850-2014), the CMIP6 models simulated a global annual increase in surface O$_3$ of between 7 and 14 ppb, with a larger increase in JJA than DJF. A global multi-model mean increase of 11.7 +/- 2.3 ppb was simulated by the CMIP6 models which agrees well with the change previously simulated by CMIP5 models. A large diversity in the historical change of surface O$_3$ was simulated by CMIP6 models across South Asia and other Northern Hemisphere regions. CMIP6 models predicted larger historical changes in surface O$_3$ than those from an emission-only driven parameterisation, indicating a potential climate change impact (Wu et al., 2008; Bloomer et al., 2009; Weaver et al., 2009; Rasmussen et al., 2013; Colette et al., 2015) on surface O$_3$ over the historical period. Small global increases in surface PM$_{2.5}$ are simulated over the historical period by CMIP6 models, with larger regional changes of up to 12 μg m$^{-3}$ on an annual mean basis and up to 18 μg m$^{-3}$ in DJF across East and South Asia. The largest diversity in the response of CMIP6 models occurs over Asian regions, with large interannual variabilities near dust source regions. CMIP6 models simulate the peak in PM$_{2.5}$ concentrations in the 1980s across Europe and North America, prior to simulating the observed decline in concentrations to present day (Leibensperger et al., 2012; Tørseth et al., 2012; Turnock et al., 2015), attributed to the implementation of air pollutant emission controls over these regions.

The CMIP6 models predict surface O$_3$ to increase across most regions in the weak mitigation scenarios (ssp370 and ssp585), particularly over South and East Asia (up to 16 ppb by 2100) due to a combination of increases in air pollutant emissions, increases in global CH$_4$ abundances and climate change. Discrepancies exist in the regional surface O$_3$ response in ssp370 between individual CMIP6 models due to differences in the response of chemistry (NOx – Fig. S17), climate (temperature) and biogenic precursor emissions (Fig. S22). Benefits to regional air quality from large reductions in surface O$_3$ are possible across all regions for scenarios that contain strong climate and air pollutant mitigation measures, including those targeting CH$_4$.

CMIP6 models predict surface PM$_{2.5}$ concentrations to decreases across all regions in both the middle-of-the-road (ssp245) and strong mitigation scenarios (ssp126) by up to 12 μg m$^{-3}$ due to the reduction in anthropogenic aerosols and aerosol precursor emissions, yielding a benefit to regional air quality. Whereas for the weak climate and air pollutant mitigation scenario (ssp370), annual and seasonal mean surface PM$_{2.5}$ is simulated to increase across a number of regions. Implementing mitigation measures specifically targeting SLCFs on top of the ssp370 scenario shows immediate improvements in PM$_{2.5}$ concentrations, restricting any changes to below present day values. The largest change in regional mean PM$_{2.5}$ concentrations, and also largest diversity across CMIP6 models, is predicted in ssp370 across South Asia, an area with already poor air quality.

Disagreements in the projection of future changes to regional surface $PM_{2.5}$ concentrations between individual CMIP6 models can be attributed to differences in the complexity of the aerosol schemes implemented within models, in particular the formation mechanisms of organic aerosols and emission of BVOCs over certain regions, along with the strength of the climate change signal (temperature and precipitation) simulated by models and the impact this has on natural aerosol emissions via Earth system couplings.

The results from CMIP6 provide an opportunity to assess the simulation of historical and future changes in air pollutants within the latest generation of Earth system and climate models using up to date scenarios of future socio-economic development. Large changes in air pollutants were simulated over the historical period, primarily in response to changes in anthropogenic emissions. Future regional concentrations of air pollutants depend on the particular trajectory of climate and air pollutant mitigation that the world follows, with important consequences for regional air quality and human health. Substantial benefits can be achieved across most world regions by implementing measures to mitigate the extent of climate change, as well as from large reductions in air pollutants emissions, including $CH_4$ which is particularly important for controlling $O_3$. In future scenarios which do not mitigate climate change and air pollutant emissions, the regional concentrations of air pollutants are anticipated to increase. Important differences between individual CMIP6 models have been identified in terms of how they simulate air pollutants from the interaction of chemistry ($O_3$ and NOx), climate (temperature and precipitation) and natural precursor emissions (BVOCs) in the future. Further research and understanding is necessary of these processes to improve the robustness of regional projections of air pollutants on climate change timescales (decadal to centennial).

**Data Availability**

CMIP6 data is archived at the Earth System Grid Federation and is freely available to download. A list of the model datasets used in this study are provided in Table 1.

**Author Contributions**

S.T.T. conducted the analysis and wrote the paper with contributions from R.J.A. T.W. and J.Z. performed BCC-ESM1 simulations. L.E. and S.T. performed CESM2-WACCM simulations. P.N. and M.M. performed CNRM-ESM2-1 simulations. L.W. H. and J.G.J. performed GFDL-ESM4 simulations. S.B. and K.T. performed GISS-E2-1-G simulations. M.A and P.G. performed HadGEM3-GC31-LL simulations. T.T. performed MIROC6-ES2L simulations. D.N. performed MPI-ESM1.2-HAM simulations. M.D. and N.O. performed MRI-ESM2-0 simulations. D.O. and M.S. performed NorESM2-LM simulations. A.S. and F.M.O'C. performed UKESM1-0-LL simulations. All co-authors have been involved in providing comments and editing the manuscript.

**Competing Interests**

The author declares that there are no conflicts of interest.

**Acknowledgements**

S.T.T. and F.M.O'C. would like to acknowledge that support for this work came from the BEIS and DEFRA Met Office Hadley Centre Climate Programme (GA01101). S.T.T. would also like to acknowledge the UK-China Research and Innovation Partnership Fund through the Met Office Climate Science for Service Partnership (CSSP) China as part of the Newton Fund. FMO'C also acknowledge the EU Horizon 2020 Research Programme CRESCENDO project, grant agreement number

641816. T.T. was supported by the supercomputer system of the National Institute for Environmental Studies, Japan, and JSPS KAKENHI Grant Number JP19H05669. K.T. and S.B. acknowledge resources supporting this work were provided by the NASA High-End Computing (HEC) Program through the NASA Center for Climate Simulation (NCCS) at Goddard Space
Flight Center. M.D. and NO were supported by the Japan Society for the Promotion of Science (grant numbers: JP18H03363, JP18H05292, and JP20K04070) and the Environment Research and Technology Development Fund (JPMEERF20172003, JPMEERF20202003, and JPMEERF20205001) of the Environmental Restoration and Conservation Agency of Japan.

For making their measurement data available to be used in this study we would like to acknowledge the providers who supplied their data to the GASSP database and TOAR database. David Neubauer acknowledges funding from the European Union's
Horizon 2020 research and innovation programme project FORCeS under grant agreement No 821205 and a grant from the Deutsches Klimarechenzentrum (DKRZ) under project ID 1051. S.S was supported by Korea Meteorological Administration Research and Development Program "Development and Assessment of IPCC AR6 Climate change scenario" under grant (KMA2018-00321)

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
