# Peer review of "Historical and future changes in air pollutants from CMIP6 models"

_Atmospheric Chemistry and Physics, 2019_

## Referee Comment (RC1) · Anonymous Referee #1 · 15 Feb 2020

This comprehensive manuscript interrogates past and future changes to surface ozone and PM2.5 air pollution in state-of-the-science multi-model simulations from AerChem-MIP/CMIP6 using updated historical and future emissions datasets. The manuscript is thorough and extremely clear and represents a very large simulation and analyses workload involving multiple international institutes. It is important to document the validation of the state of the science global Earth system models and assess the surface air quality responses to past and future global change for new updated emission scenarios. The methodology is sound and the Figures are clear. It may be possible to slightly reduce the number of Figures in the main manuscript further. The multi-model evaluation of surface ozone and PM2.5 is highly valuable to the entire chemistry-climate scientific community. The manuscript discusses changes to both emissions versus

climate, but these are mostly qualitative, and even intuitive, rather than quantitative be-cause none of the applied simulation protocols formally separate out climate change versus emissions change impacts. The authors have done an excellent job with the available datasets and from this perspective the paper is appropriate for publication. However, the results raise some challenging questions about the usage of these global models for surface air quality research. For instance, human health effects calcula-tions depend explicitly on absolute concentrations for exposure. There are some more detailed comments/questions to consider below.

1. The systematic model overestimate of surface ozone across all models is striking (e.g. Fig. 3(c) and (f)). From Fig. 4 for the NAM and EU where there is by far the most data, all models are unable to reproduce the seasonal dynamics (maximum in NH spring and gradually decreasing through the summer months). The authors offer some possible explanations: "The overestimation in the CMIP6 models analysed here could be due to the coarse resolution of the ESMs, an excess of O3 chemical production (potentially due to an overabundance of NOx and/or VOCs) and weak O3 deposition.". If possible, it would be good to have a more robust and clear explanation and understanding of the systematic overestimate and poor seasonal dynamics? Is the coarse resolution problem related to directly injecting the NOx emissions across the large spatial extent ∼2degx2deg (∼200km) grid cells? Where the ozone production regime will be highly NOx-limited at this scale? What is needed from the community to improve/address the systematic positive bias in surface ozone simulations in global models?

2. The systematic underestimate in monthly PM2.5 in NAM, EU and EAS (Fig. 6) is troubling. Can it really be explained only by the missing nitrate component? Are there other fundamental missing or misrepresented processes? Output from these models is more frequently being used to assess health impacts, for example, premature mortality due to outdoor air pollution exposure (PM2.5 and ozone) but such application would not be justified based on the model/measurement comparison here. It could be argued

from the model/measurement evaluation that the models cannot be applied as tools to study the surface air quality?

3. How reliable are the model simulations of past and future changes when the monthly mean surface air quality concentrations cannot be reproduced by the models and there are clear systematic biases?

4. Fig. 9. I find this Figure also striking in the diversity of model results for historical surface ozone evolution. Why does the GISS model have such large changes/sensitivities to the PI-PD? Esp. for Europe, S. Asia and E. Asia (but not SE Asia + less polluted SH regions)? Does the GISS model gas-phase chemistry have a larger sensitivity to NOx changes than other models and why? The GISS model is also an outlier in Fig. 10 for evolution of PM2.5 over S. Asia region specifically? What is the value of the multi-model mean in e.g. Fig. 13 when there is such large diversity of sensitivities shown in Figs. 9&10?

5. "Surface O3 increases across most world regions in this scenario can be attributed to the large increase in global CH4 abundances (80%) and the large predicted increase in surface temperatures". Why do increases in surface temperature increase surface ozone concentrations independent of emissions? What is the mechanism? Is it temperature, or co-varying stagnation or light/downward SW? How do we know it is temperature with 100% certainty as stated here?

6. "across East Asia the additional precursor emission reductions in ssp370-lowNTCF have made little difference to surface O3 concentrations predicted by the CMIP6 models, indicating that other factors are more important over this region (chemistry or climate change)." This result is critically important. So, aggressive mitigation of ozone precursors has no impact on the surface ozone concentrations in this region relative to a scenario with those precursors? What is the reason for surface ozone in East Asia to be independent of ozone precursor emission changes under this level of global change? Further explanation is needed. Are there climatic feedbacks from the precursors themselves that are offsetting the changes?

7. "Discrepancies in the magnitude of change in these emissions due to climate and *land-use change\**". Please specify in similar to Table 1 the models for which the natural emissions and atmospheric chemistry are actually dynamically coupled with the climate model's land surface scheme and vegetation cover / Plant Functional Types (that are dynamically changing in the simulations due to human land use change). Which models have the BVOC emissions actually coupled to the climate model's internal land surface scheme? If uncertainty in the changes to natural emissions is an important conclusion of the paper, there needs to be a separate Table describing the representation of those emissions in each model.

Minor comments

I find Fig. 2 challenging to look at and wonder about for other readers. I appreciate it is difficult to show this Fig. 1 type information across multiple regions.

Is it necessary to have both Fig 6 and Fig 8 i.e. for the 2000-2010 and 2005 and 2014 periods? Could one of the plots go into SI?

"Large regional historical changes are simulated for both pollutants, across East and South Asia, with an increase of up to 40 ppb for $O_3$ and 12 $\mu$g m-3 for PM2.5." and similar sentences in abstract. Need to include the temporal averaging associated with those values in abstract (annual).

"Near Term Climate Forcers (NTCFs)." IPCC AR6 uses "Short-lived Climate Forcers (SLCFs)".

"Initial assessments have been made of future changes to air pollutants in the SSPs using simplified models." Need to add references here.

"A particular climate mitigation target, in terms of an anthropogenic radiative forcing by 2100, is included on top of each SSP" What does "on top of" mean exactly?

[Figure]

"However, scenarios with large increases in global CH4 abundances, a large climate change signal and limited control of precursor emissions fail to restrict regional increases in surface O3, leading to poor future air quality and potential human health impacts (Silva et al., 2017)." Is this statement redundant/obvious? Where is the new science?

"Whilst there is disagreements" sp. there are

---

## Referee Comment (RC2) · Anonymous Referee #2 · 15 Apr 2020

This manuscript conducts an evaluation of surface PM2.5 and ozone with observations for the CMIP6 chemistry-climate models that participated in AerChemMIP. It also documents the simulated historical & future changes in annual mean ozone and PM2.5 in various regions around the globe. It's clear that an enormous amount of effort went in to preparing this manuscript. By detailing the performance of each individual model (10 for PM2.5; 5 for ozone) against the available observations, a major community service has been performed in the production of this detailed supplemental information.

The rather long paper documents the current status of O3 and PM2.5 in the latest versions of global chemistry-climate models. It does so, however, without much attempt to understand more deeply the inter-model differences, or the sources of agreement, beyond discussing qualitative links to the emission trajectories or referencing relation-

ships identified in prior work. A stronger paper would be more cohesive throughout and communicate better the novelty of the work. Below I suggest ways to strengthen the paper in each of these two directions, followed by more detailed comments. I support the points made by the other reviewer and so try to avoid repeating those points here.

First, the model evaluation presented is not tied in a clear way to the past or future projections of the models. The evaluation focuses on monthly and seasonal data but then only annual mean concentrations are presented for the historical and future trends. It seems far more relevant to evaluate regional trends in annual mean concentrations where observations allow this, or to demonstrate some relationship between seasonal cycles and future changes across the models (and should one exist, this would be an exciting finding as it would open up the possibility of identifying a "best" model from the evaluation with observations). The evaluation shown in Figures 5 and 6 of the Mortier et al. paper or in Figure 4 of Griffiths et al. in this special issue seems more relevant, although the remote sites used in Griffiths et al. are not that relevant for the polluted regions examined in this study. One could tackle a similar type of evaluation for North America and Europe where there are at least two decades of long-term observations for ozone and PM2.5, and it should be particularly straightforward to do so with the gridded MERRA reanalysis product for PM2.5. An alternative angle could be to examine if the past or future trends are strongly seasonally dependent. If so, showing some of the seasonality in the projections would connect better to the seasonal evaluation included. If the authors choose to remove any of the current figures, they should be included in the supplemental material, as the general evaluation done here will certainly be of high value to the modeling community.

Second, the authors could better demonstrate the new contributions here, perhaps by looking a bit more closely at some aspect of the inter-model differences rather than ending with qualitative and in some cases speculative statements. For example, are there clear relationships between the inter-model spread in the global or regional temperature or precipitation changes and the air pollution changes projected over time?

Could previously identified general conclusions regarding relationships between global ozone, NOx and methane (see Figure 6 of Stevenson et al. 2006, Figure 13 of Young et al., 2013) be extended to surface ozone, and regionally? Can any conclusions be made as to whether future changes in particulate matter depend most on a particular component? There is a lot of useful information in the supplement regarding aerosol components and temperature changes that could be connected more closely to the changes reported in the main text. I find Figures 12 and 14 particularly interesting and the results presented there would be even more useful if they were connected more directly to changes in regional or global temperature, precipitation, humidity, air pollutant emissions, precursor surface concentrations, or whichever quantities are available across the set of models.

Detailed comments

One of the more interesting aspects of the paper is the comparison with the parameterisation based on HTAP models to separately attribute changes to emissions versus the combined emissions and climate changes simulated by the AerChemMIP models. However, it would help to have a better summary of how the parameterisation was developed and applied. Is it one parameterisation, or an ensemble of parameterisations that were developed separately for each model? Is there any overlap in the models used in developing the parameterisation and the AerChemMIP models? If so, can that subset of models be analyzed to attribute with greater confidence the role of climate change? Would this study support future work to extend this parameterisation to include the effects of temperature, humidity, or some other changes in climate variables?

The referencing throughout the text seems to focus on more recent work rather than early papers that first identified important relationships. For example, the role of increasing water vapor in increasing ozone loss was first pointed out by Johnson et al., 1999 (text around line 65, and especially 450); the role of methane for surface ozone by Fiore et al. 2002 and Shindell et al. 2012 (text around line 65); the increase in ozone under climate change scenarios by Wu et al. 2009 and Weaver et al. 2009 (text around

line 645).

Try to quantify wherever possible in the text, such as line 29 "consistent overestimate", line 31 "consistently underestimated", by how much? Is there any improvement in biases, or worsening, relative to prior studies? Line 40 "important differences", can anything be said as to which is most important or handled most realistically? Line 44-45 should include at least one example to support this statement.

Lines 113-114. Why do this for a future scenario rather than the historical period where there might be some opportunity to evaluate with observations?

Figure 2 is difficult to digest. Why does this need to be in the main text? This is an example where more could be gleaned from the analysis if these changes in emissions could be shown to be related to the projected changes in ozone and/or PM2.5, perhaps through scatterplots.

Line 271. This can be checked and stated more confidently by examining NO2+O3 rather than just O3.

Lines 444-445 is not new as this was a major result from CMIP5 era RCP8.5. Some of that work probably deserves a citation, such as Gao et al. 2013.

The biases in Figure 3 are very hard to read. It should be stated if the color bar saturates.

Lines 494-500. These seemingly different responses may occur because of different responses in winter versus summer across the models being mixed together in the annual mean.

Lines 503-514. Can these points about sources of inter-model differences be illustrated and based on evidence rather than surmised? Same goes for lines 580-590 & 600-602, where it might be worth moving some of the supplemental information into the main text to support more strongly these points.

Lines 648-650 should be supported with observations for this conclusion to be made here.

Stronger evidence should also be included to support conclusions on lines 665-666 & 677-678.

References cited:

Gao, Y. et al. (2013), The impact of emission and climate change on ozone in the United States under representative concentration pathways (RCPs), Atmos. Chem. Phys., 13, 9607–9621, doi:10.5194/acp-13-9607-2013.

Johnson et al. (1999) Relative roles of climate and emissions changes on future tropospheric oxidant concentrations, J. Geophys. Res. – Atmos., 104 (D15) 18,631-18,645.

Fiore et al. (2002) Linking ozone pollution and climate change: The case for controlling methane, Geophys. Res. Lett., 29 (19) 1919.

Shindell et al. (2012) Simultaneously mitigating near-term climate change and improving human health and food security, Science, 335 (6065) 183-9. doi: 10.1126/science.1210026.

Stevenson et al. (2006) Multimodel ensemble simulations of present-day and near-future tropospheric ozone, J. Geophys. Res. – Atmos., 111 (D08301) doi:10.1029/2005JD006338

Weaver et al. (2009) A preliminary synthesis of modeled climate change impacts on U.S. regional ozone concentrations. Bull. Amer. Meteor. Soc., 90, 1843-1863.

Wu et al. (2009) Effects of 2000-2050 global change on ozone air quality in the United States, J. Geophys. Res. - Atmos. 113, D06302, doi:10.1029/2007JD008917.

Young et al. (2013) Pre-industrial to end 21st century projections of tropospheric ozone from the Atmospheric Chemistry and Climate Model Intercomparison Project (ACCMIP), 13, 2063-2090, doi:10.5194/acp-13-2063-2013.

---

## Author Comment (AC1) · 17 Jul 2020

**Author's response to referee comments on "Historical and Future changes in Air pollutants from CMIP6 Models"**

Steven T. Turnock et al.
Correspondence to: Steven T. Turnock
(steven.turnock@metoffice.gov.uk)

We would like to thank both of the reviewers for their helpful and constructive comments. Below we have responded to each comment in turn and made alterations to the manuscript where appropriate (shown enclosed in *"speech marks and italic font" and any deletions from the manuscript shown with a strikethrough ""*). The referee comments are shown first in grey shading and the author's response is shown below in normal font.

We would like to note that all analyses have been updated based on current model availability to aid in the response to the reviewers comments below. Minor changes to the text and figures have been made to reflect this. This includes the addition of both surface $O_3$ and $PM_{2.5}$ data from a new CMIP6 model, MRI-ESM2-0. Furthermore, the use of data from the GISS-E2-1-H model has been replaced by that from GISS-E2-1-G, as data is available from both future and historical scenarios with this configuration of the GISS model. A different ocean is coupled to the same atmosphere in the two versions of the GISS-E2-1 model, which does not make any significant changes to the regional simulation of air pollutants. Additional data has also now been included for more future scenarios from CMIP6 models already included within the manuscript. The overall results and conclusions of the paper remain unchanged.

**Response to Referee 1**

This comprehensive manuscript interrogates past and future changes to surface ozone and PM2.5 air pollution in state-of-the-science multi-model simulations from AerChemMIP/CMIP6 using updated historical and future emissions datasets. The manuscript is thorough and extremely clear and represents a very large simulation and analyses workload involving multiple international institutes. It is important to document the validation of the state of the science global Earth system models and assess the surface air quality responses to past and future global change for new updated emission scenarios. The methodology is sound and the Figures are clear. It may be possible to slightly reduce the number of Figures in the main manuscript further. The multi-model evaluation of surface ozone and PM2.5 is highly valuable to the entire chemistry-climate scientific community. The manuscript discusses changes to both emissions climate, but these are mostly qualitative, and even intuitive, rather than quantitative because none of the applied simulation protocols formally separate out climate change versus emissions change impacts. The authors have done an excellent job with the available datasets and from this perspective the paper is appropriate for publication. However, the results raise some challenging questions about the usage of these global models for surface air quality research. For instance, human health effects calculations depend explicitly on absolute concentrations for exposure. There are some more detailed comments/questions to consider below.

1. The systematic model overestimate of surface ozone across all models is striking (e.g. Fig. 3(c) and (f)). From Fig. 4 for the NAM and EU where there is by far the most data, all models are unable to reproduce the seasonal dynamics (maximum in NH spring and gradually decreasing through the summer months). The authors offer some possible explanations: "The overestimation in the CMIP6 models analysed here could be due to the coarse resolution of the ESMs, an excess of O3 chemical production (potentially due to an overabundance of NOx and/or VOCs) and weak O3 deposition.". If possible, it would be good to have a more robust and clear explanation and

understanding of the systematic overestimate and poor seasonal dynamics? Is the coarse resolution problem related to directly injecting the NOx emissions across the large spatial extent ~2degx2deg (~200km) grid cells? Where the ozone production regime will be highly NOx-limited at this scale? What is needed from the community to improve/address the systematic positive bias in surface ozone simulations in global models?

I would like to thank the reviewer for their comment on the discrepancies between models and observations, which is an ongoing topic of research within the global chemistry climate modelling community. The simulated overestimation of surface $O_3$ concentrations by CMIP6 models presented in this manuscript is consistent with those from previous work in the comparison of 15 ACCMIP models against TOAR observations by Young et al., (2018). Young et al., (2018) (and references therein to other previous model evaluation studies) also found that ACCMIP models overestimated observed surface $O_3$ concentrations, as well as simulated peak surface $O_3$ concentrations later in the year than observations, which is consistent with the seasonality simulated by the CMIP6 models (Fig. 4). The overestimation of observed surface $O_3$ concentrations in the northern hemisphere is common across global models and is a persistent feature in model evaluations across numerous different generations of models. Therefore, it is most likely that the overestimation is due to an issue that is commonplace across all models, such as uncertainties in emission inventories or processes that are represented in a similar way in models e.g. deposition. Performing additional sensitivity experiments using a range of emission inventories or deposition schemes would aid in the understanding if these were key issues in contributing to the model biases.

One such way to identify drivers of uncertainties in models is to conduct a sensitivity analysis on global chemistry-climate models by analysing the sensitivity of $O_3$ to variations in different model parameters (Wild et al., 2020). Tropospheric $O_3$ was found in the study of Wild et al. (2020) to show a large sensitivity to atmospheric water vapour, precursor emissions and dry deposition processes in three global chemistry-climate models. A more detailed global sensitivity study applied specifically to surface $O_3$ formation, would have the potential to highlight where key areas of research are needed to improve model simulations. Additionally if the sensitivity study were to be combined with observational constraint, as done in other studies for aerosols (Johnson et al., 2019), then this work has the potential to further identify the reasons for model uncertainties and highlight where improvements are needed in the simulation of ozone formation.

When considering the impact that horizontal resolution could have, model evaluation studies using regional composition models (with finer horizontal resolution than global models) across Europe, North America and East Asia have reported improvements in the simulation of surface $O_3$ when compared to observations, but also that certain models overestimate surface $O_3$ (Gao et al., 2018; Im et al., 2018). Analysing the impacts of model resolution on tropospheric $O_3$ production shows improvements in the simulation of $O_3$ going from ~600 km to 120 km horizontal resolution (reduced $O_3$ production over polluted regions), but also finds that this resolution is still too coarse to sufficiently resolve regional $O_3$ production (Wild and Prather, 2006). In addition, a comparison of the simulation of air pollutants using different model resolutions in Neal et al., (2017), highlighted benefits in using higher resolution modelling in the simulation of primary pollutants from improved emissions, but only modest improvements for secondary pollutants like $O_3$ and $PM_{2.5}$. Enhancing the relatively coarse resolution of global models, and the benefits of higher resolution emissions and other processes, might improve the simulation of surface $O_3$ but would not necessarily account for all of the model-observational discrepancies.

Whilst the aim of the current study was to highlight discrepancies in surface $O_3$ between the latest generation of CMIP6 models and observations, it was not intended to provide a detailed explanation

of the causes of this discrepancy within individual models. Further work is required to explore the reasons for the differences between individual CMIP6 models and observations, which would need to be the subject of future research, potentially using global sensitivity analysis, process studies and simulations at finer resolution.

However, a couple of changes to the manuscript have been included below to improve the description of the evaluation work.

The sentences on Page 9 line 266-268 have been amended to the following:

"*The model observational comparison of CMIP6 models to the TOAR observations are consistent across all models and with the previous evaluation of ACCMIP models (Young et al., 2018) . This indicates a common source of error within models, for example uncertainties in emission inventories, deposition processes or vertical mixing (Wild et al., 2020). In addition, the coarse resolution of the ESMs could lead to an overproduction of $O_3$ across polluted regions, with finer resolutions exhibiting improvements in the simulation of surface $O_3$ (Wild and Prather, 2006; Neal et al., 2017).*"

A new sentence has been included on page 30 line 640:

"*The comparison of surface $O_3$ and $PM_{2.5}$ simulated by CMIP6 models to observations shows similar biases to previous generations of global composition-climate models. Further studies are required (e.g. global sensitivity or process studies) to explore the uncertainties in models and the differences with observations.*"

2.  The systematic underestimate in monthly PM2.5 in NAM, EU and EAS (Fig. 6) is troubling. Can it really be explained only by the missing nitrate component? Are there other fundamental missing or misrepresented processes? Output from these models is more frequently being used to assess health impacts, for example, premature mortality due to outdoor air pollution exposure (PM2.5 and ozone) but such application would not be justified based on the model/measurement comparison here. It could be argued from the model/measurement evaluation that the models cannot be applied as tools to study the surface air quality?

The reviewer is correct in that the underestimation of observed $PM_{2.5}$ by the CMIP6 models cannot be solely attributed to the exclusion of nitrate aerosol mass but is potentially due to a number of issues, which are also found in other model evaluation studies. Figure S12 (now S13) shows that the surface mass concentrations of nitrate aerosols from two CMIP6 models could make a small contribution to the total $PM_{2.5}$ concentrations, particularly in northern hemisphere winter months (Bauer et al., 2016). On page 13 lines 326 to 328 we mention that including the nitrate aerosol mass fraction could account for some, but not all of the discrepancy in $PM_{2.5}$ between models and observations. In addition, the MERRA-2 reanalysis product has been constructed in the same way as $PM_{2.5}$ has been computed from the CMIP6 models, by not including the mass from nitrate aerosols. Page 13 lines 328 to 330 makes the point that the CMIP6 models still show an underestimation of the MERRA-2 product, and that differences are due to errors in other aerosol sources and processes.

Other studies performing single and multi-model evaluation of global and regional models against observations across North America, Europe and Asia found that simulated concentrations of fine mode aerosols tended to be underestimated due to a number of possible reasons including: errors in emissions, simulated meteorology and aerosol formation mechanisms for both inorganic and secondary organic aerosols (Tsigaridis et al., 2014; Pan et al., 2015; Glotfelty et al., 2017; Solazzo et al., 2017; Im et al., 2018). On page 12 lines 313 to 317 of the manuscript we highlighted the potential

reasons for some of the model-observational discrepancy and the similarity to other studies. This text has now been slightly modified to the following:

"*Nevertheless, the evaluation highlights that fine particulate matter (PM$_{2.5}$) is generally underrepresented in the CMIP6 models across North America, Europe and parts of Asia for which observations are available; a similar result to other studies evaluating global and regional models (Tsigaridis et al., 2014; Pan et al., 2015; Glotfelty et al., 2017; Solazzo et al., 2017; Im et al., 2018 ). Numerous reasons potentially exist for the model observation discrepancy shown here and in other studies including uncertainties in emissions inventories (e.g. local dust sources), errors in wet/dry deposition, the absence/underrepresentation of aerosol formation processes (e.g. organic aerosols) and the coarse resolution of global models leading to errors in emissions and simulated meteorology. Understanding the causes of model observational discrepancies is an area of active research and should be explored in further research, for example in a global multi-model sensitivity study that explores model uncertainties.*"

The reviewer points out that the underestimation of PM$_{2.5}$ by CMIP6 models might preclude their use for studying the health impacts of air quality. Air pollutant concentrations from coarser resolution models have previously been shown to produce lower health impacts than those from models with a finer horizontal resolution (Punger and West, 2013; Li et al., 2016; Silva et al., 2016a). However, air pollutant concentrations from global model simulations (including those of CMIP5 models) have been successfully utilised in health impact studies by using the change in concentrations between the future and present day (as they are able to reproduce the relationship between concentrations and emissions/climate, see response to point 3 below) or applying correction factors to account for the anticipated underestimations in concentrations in the present day, particularly across urban areas (Silva et al., 2016b; Butt et al., 2017; Chowdhury et al., 2018; Bauer et al., 2019). It should also be noted that there are large uncertainties in the exposure response functions used in health impact studies that relate exposure of air pollutants to human health impacts (Jerrett et al., 2009; Burnett et al., 2014). Health impact studies, including those used within the global burden of disease assessment, have used a combination of global modelling, satellite remote sensing products and ground based observations to generate PM$_{2.5}$ concentrations with greater precision, reducing some of the simulated biases from global composition models (van Donkelaar et al., 2010; Brauer et al., 2016; Jerrett et al., 2017). We would therefore recommend that any studies wanting to use output from CMIP6 models to study the future health impacts from changes in air quality consider the different techniques and methods outlined in some of the above studies.

3.  How reliable are the model simulations of past and future changes when the monthly mean surface air quality concentrations cannot be reproduced by the models and there are clear systematic biases?

Whilst there are model observational biases in the absolute magnitude of the present day simulations of both surface O$_3$ and PM$_{2.5}$ in the CMIP6 models, there is some confidence in the ability of models to simulate temporal changes when compared to long-term historical observations.

Long term changes in surface O$_3$ concentrations from CMIP6 models have been evaluated in the tropospheric O$_3$ CMIP6 companion paper of Griffiths et al., (2019) at four remote locations with the longest observational record over the second half of the twentieth century. Figure 4 of Griffiths et al., (2019) shows that the CMIP6 models are able to reproduce the observed multi-decadal changes in surface O$_3$, providing some confidence in the ability of CMIP6 models to simulate future changes. Young et al., (2018) presented a summary of the ability of the previous generation of global chemistry

climate models to simulate long term changes in surface $O_3$ based on the comparisons in Parrish et al., (2014). This showed that selected CMIP5 models had areas of agreement and disagreement with long-term measurements of $O_3$ concentrations at northern midlatitudes. However, the models were reported to underestimate the observed long-term changes in surface $O_3$ by ~50%. The evaluation highlighted a number of limitations in long-term comparisons from uncertainties in emission changes, observational records, sampling biases and low frequency variability influencing observed $O_3$ concentrations that is not simulated by models. Therefore, any future predictions of changes in surface $O_3$ by CMIP6 models could be similarly uncertain and represent a conservative estimate of change. The long-term comparison of models and observations is an area of active research and is currently being undertaken in other studies using output from CMIP6 models, with any results from this providing information on the ability of CMIP6 models to simulate changes in surface $O_3$.

The absence of long historical records of fine particulate matter concentrations at the surface have limited the ability to evaluate any changes simulated by models over a multi-decadal period. The longest records of such data exist over Europe and North America and this is where long-term evaluations have tended to be focussed. Studies evaluating global composition models over these regions and at other locations have tended to show that models are able to reproduce the multi-decadal trends in aerosol components, particularly sulphate, and aerosol optical depth (AOD) (Lamarque et al., 2010; Pozzoli et al., 2011; Leibensperger et al., 2012; Chin et al., 2014; Turnock et al., 2015; Aas et al., 2019). CMIP5 models were previously shown to have a reasonable reproduction of satellite trends in AOD since the 1980s (Shindell et al., 2013). Simulated aerosol trends in AOD, sulphate and particulate matter from global composition models, including a number of CMIP6 models, have been shown to be able to reproduce observed changes over the last two decades (Mortier et al., 2020). These studies all provide confidence in the simulation of past and future changes in fine particulate matter within global composition models, even though the magnitude of present day concentrations is underestimated.

There is some confidence in the ability of global models to reproduce long term observed changes in air pollutants from previous studies, although this will require further research using the latest generation of models contributing to CMIP6 to understand the reasons for any discrepancies. Nevertheless, multi-decadal changes in air pollutants simulated by CMIP6 models in different future scenarios provide a useful indication of future changes in air pollutants under different pathways of emissions and climate change, even if there are potential uncertainties associated with the projections.

The ability of CMIP6 models to reproduce long term changes in surface $O_3$ was mentioned on p18, lines 384-386. We have amended this sentence to that below for additional clarity.

*"An evaluation of the long-term changes in surface $O_3$ over the historical period simulated by the CMIP6 models at specific measurement locations is presented separately in the tropospheric $O_3$ CMIP6 companion paper of Griffiths et al., (2019). This shows that the CMIP6 models can reasonably represent long term changes in surface ozone since the 1960s, providing a degree of confidence in the future projection of changes in the CMIP6 scenarios. However, long term changes in simulated surface $O_3$ from the previous generation of global composition climate models (CMIP5) were found to underestimate the observed trend at northern hemisphere monitoring locations (Parrish et al., 2014). Further comparisons of long-term historical observations of surface $O_3$ with that simulated by CMIP6 models is outside the scope of the current work but will be the subject of future research."*

The sentence on page 20 lines 416-418 has been replaced with the following:

*"There is limited long-term multi-decadal observational data available to assess changes in aerosols simulated by global models. Previous studies using long-term data since the 1980s, mainly over Europe and North America, have found that global models are able to reproduce the observed multi-decadal changes in aerosols relatively well (Pozzoli et al., 2011; Leibensperger et al., 2012; Tørseth et al., 2012; Chin et al., 2014; Turnock et al., 2015; Aas et al., 2019). More recently, global composition models, including some CMIP6 models, were shown to be able to reproduce the observed changes in AOD, sulphate and particulate matter over the last two decades (Mortier et al., 2020). The ability of global composition models to reproduce historical changes in aerosols provides a degree of confidence in the future projections under the CMIP6 scenarios. Further model observational comparisons of multi-decadal changes in aerosols will need to be undertaken to improve the understanding of changing aerosol properties and processes."*

4. Fig. 9. I find this Figure also striking in the diversity of model results for historical surface ozone evolution. Why does the GISS model have such large changes/sensitivities to the PI-PD? Esp. for Europe, S. Asia and E. Asia (but not SE Asia + less polluted SH regions)? Does the GISS model gas-phase chemistry have a larger sensitivity to NOx changes than other models and why? The GISS model is also an outlier in Fig. 10 for evolution of PM2.5 over S. Asia region specifically? What is the value of the multimodel mean in e.g. Fig. 13 when there is such large diversity of sensitivities shown in Figs. 9&10?

As the reviewer points out, Figure 9 shows that there is a large diversity in the regional surface $O_3$ response over the historical period across the CMIP6 models. In the revised manuscript the regional surface $O_3$ response from an additional model (MRI-ESM2-0) has been included on Figure 9, which doesn't change the overall result but adds to the multi-model mean.

Further investigation has been undertaken into the different historical changes in surface $O_3$ concentrations from CMIP6 models with the regional annual mean absolute concentrations over the historical period shown in a new Figure S15 below and the spatial annual mean concentrations in 2005-2014, 1980-1989 and 1850-159 shown in a new Figure S14 below. Figure S14 and S15 show that there is a range of surface $O_3$ concentrations simulated by CMIP6 models over different regions particularly in 1850, with more agreement between models towards the present day. However, there is a noticeable difference in the regional change of simulated surface $O_3$ concentrations over the historical period in different models (Fig. S15). Out of all the CMIP6 models, UKESM1 tends to simulate some of the smallest changes in regional annual mean surface $O_3$ concentrations over the historical period due to having larger concentrations in the 1850s and some of the smallest concentrations in recent decades. Whilst the opposite response is true for simulated regional annual mean surface $O_3$ concentrations in the GISS-E2-1-G model (smallest 1850 and largest 2015 concentrations), resulting in some of the largest regional mean changes in annual mean surface $O_3$ shown on Figure 9.

Uncertainties in the simulation of pre-industrial $O_3$ concentrations across models is one of the contributing factors to the diversity of the response in historical surface $O_3$ across models. Figure S14 shows that there is significant diversity in the simulated pre-industrial $O_3$ concentrations across CMIP6 models due to the lack of observation data for validation purposes. Uncertainties arise in the simulation of the pre-industrial $O_3$ state due to differences in meteorology and chemical mechanisms, in particularly the simulation of NOx and natural emission sources of $O_3$ precursors (isoprene) in this period, which can dominate $O_3$ formation.

In addition, the difference in the historical simulation of surface $O_3$ concentrations across CMIP6 models could be due to the chemical sensitivity of each model to NOx concentrations in the different time periods and the change in concentrations between them. A comparison of the regional annual

mean surface $O_3$ concentrations and regional annual mean NOx (NO + $NO_2$) concentrations for three time periods (new Figure S17 as shown below) highlights the different chemical sensitivities of $O_3$ formation to NOx across models. Across most regions the higher NOx concentrations in UKESM1 have tended to result in higher surface $O_3$ concentrations in the 1850s and lower in the present day. Whereas for GISS-E2-1-G the lower NOx concentrations have tended to result in the lower surface $O_3$ concentrations in the 1850s and higher concentrations in the present day period, indicating a shift in chemical environments over time. The large sensitivity of $O_3$ formation to surface NOx concentrations in the GISS model was also shown in the global sensitivity study of Wild et al., (2020). The sensitivity of surface $O_3$ formation to different historical NOx concentrations is particularly noticeable in most models over South Asia (due to the large regional changes in NOx) but especially evident in GISS-E2-1-G, which results in its large surface $O_3$ response over this region. Additionally, the large increase in $PM_{2.5}$ over the historical period in South Asia (Fig. S18 below) could also influence the heterogeneous loss of radicals to aerosols and therefore also changes to $O_3$.

[Figure]

**Figure S14 -** Annual mean surface $O_3$ concentrations across 6 CMIP6 models over the period 2005-2014 (top row), 1980-1989 (middle row) and 1850-1859 (bottom row).

[Figure]

**Figure S15** - Regional and global annual mean surface $O_3$ concentrations across 6 CMIP6 models and the HTAP_param. The multi-model annual mean year 2005-2014 surface $O_3$ concentrations (+/- 1 standard deviation) are shown in the top left of each panel. Regions are defined in Figure S1.

[Figure]

**Figure S17 –** Annual mean regional surface $O_3$ concentrations compared to regional annual mean surface NOx (NO + $NO_2$) concentrations across 6 CMIP6 models over three ten-year periods of 1850-1859 (circles), 1980-1989 (diamonds) and 2005-2014 (triangle).

As the reviewer points out the change in historical surface $PM_{2.5}$ from GISS-E2-1-G on Figure 10 is also shown to be smaller than other CMIP6 models over South Asia. Like for surface $O_3$, a revised Figure 10 has now been produced to include the additional model results from MRI-ESM2-0, which hasn't altered the overall result. A new Figure S18 below shows the pre-industrial to present day change in total annual mean surface $PM_{2.5}$ and from each individual component. Looking at the historical change in each aerosol component highlights that over South Asia, the response in GISS-E2-1-G is the smallest from all CMIP6 models for sulphate and one of the smallest for black carbon and organic aerosol. The combination of the smaller response in all anthropogenic aerosol components from GISS-E2-1-G over South Asia results in the smaller response in historical total $PM_{2.5}$ concentrations shown on Figure 10 and below.

[Figure]

**Figure S18** – Pre-industrial (1850-1859 mean) to present day (2005-2014 mean) changes in the regional and global annual mean surface total PM$_{2.5}$ concentrations (PM) and that from each individual component (BC – black carbon, DU – dust, SU – sulphate, OA – organic aerosol and SS – sea salt). Individual circles represent each annual and seasonal mean changes from the 11 individual CMIP6 models, with the multi-model mean represented by the solid bar. The. Regions are defined in Figure S1.

Showing the diversity in response across CMIP6 models is useful as this identifies where the models (with different chemistry and meteorology) agree but also where there is disagreement and uncertainties in the simulated surface O$_3$ response. This could help identify further research priorities to understand the differences between models. The multi-model means shown on Figure 11 and 13 (now Fig. 14) also contain a shaded area which shows the diversity in the simulated response across CMIP6 models (+ 1 standard deviation of the multi-model mean). We feel that it is useful to show the multi-model mean as it provides a degree of confidence in the future projections and allows the reader to identify where there is agreement between models in the simulated future response (such as Europe) but also where there is disagreement and uncertainty in the range of potential future model responses (e.g. over South Asia in ssp370). Where there is significant model diversity, we feel that including a multi-model mean with a degree of uncertainty provides useful information on the confidence in future predictions of surface air pollutants across different CMIP6 models.

We have made the following changes to the manuscript to reflect the above discussion on historical changes in surface O$_3$ and PM$_{2.5}$.

The sentence on page 17, line 379 has been amended as follow:

"*The simulated changes in surface O$_3$ across 6 CMIP6 models and the HTAP_param are shown in Figure 9 and Figure S14-S15 over the historical period of 1850 to 2014.*"

A new sentence has been inserted on Page 18, line 388.

"*The large diversity across CMIP6 models in the surface $O_3$ response over the historical period can be attributed to the different magnitude of simulated $O_3$ concentrations in the 1850 period (Figure S14) and the rate of change in regional mean $O_3$ concentrations (Figure S15), which is related to the different chemical sensitivity of $O_3$ formation in each model to changing NOx concentrations over the historical period (Figure S17).*"

The sentence on page 18, line 391 has been amended to the following:

"*South Asia is the region with the largest diversity in simulated historical changes in surface $O_3$ of between 16 and 40 ppb, with a larger range in DJF (10-40 ppb) than in JJA (19-36 ppb). The large diversity in CMIP6 models is attributed to the large differences in simulated NOx concentrations, and hence chemical sensitivities of $O_3$ formation, occurring across South Asia (Figure S17). In addition, the large historical change in $PM_{2.5}$ over this region* (Fig. S18) *could alter the heterogeneous loss rate of radicals to aerosols and therefore also affect $O_3$ formation.*"

The sentence on Page 19, line 410-412 has been amended as follows:

"*The largest model diversity is also exhibited over the Asian regions with variations in the response between models of up to 50%,*  *with larger differences between models in DJF than JJA (Figure S16), reflecting the differences shown in the present day model evaluation (Fig. 6). The inter-model differences can be attributed to the different simulation of historical changes in the anthropogenic components sulphate, black carbon and organic aerosols (Figure S18).*"

5. "Surface O3 increases across most world regions in this scenario can be attributed to the large increase in global CH4 abundances (80%) and the large predicted increase in surface temperatures". Why do increases in surface temperature increase surface ozone concentrations independent of emissions? What is the mechanism? Is it temperature, or co-varying stagnation or light/downward SW? How do we know it is temperature with 100% certainty as stated here?

We thank the reviewer for the comment on this particular sentence, which was an attempt to identify the importance of changes in $CH_4$ and climate on regional surface $O_3$ concentrations in the ssp370 scenario, despite the reductions in precursor emissions over certain regions. Previous work has shown that climate change can have an important impact on surface and tropospheric $O_3$ concentrations; the ozone climate penalty (Rasmussen et al., 2013; Stevenson et al., 2013; Colette et al., 2015). In addition, the importance of future changes in global $CH_4$ abundance for surface $O_3$ concentrations has been previously shown (Fiore et al., 2009; Wild et al., 2012; Young et al., 2013; Turnock et al., 2019). Therefore, the purpose of the sentence mentioned by the reviewer was to highlight the continued importance of these drivers in the SSPs used in CMIP6 models, although we appreciate that the sentence needs to be clearer.

Therefore, the sentence on P21 line 443 has been amended to improve its clarity as follow:

"*Despite the reductions in $O_3$ precursor emissions across North America, Europe and East Asia by 2100 (Fig. 2) surface $O_3$ concentrations have continued to increase up to the end of this period, indicating the importance of future changes in chemistry, global $CH_4$ abundances and climate on the response of surface $O_3$ in ssp370* (Wild et al., 2012; Gao et al., 2013; Rasmussen et al., 2013; Young et al., 2013; Colette et al., 2015; Fortems-Cheiney et al., 2017; Li et al., 2017; Turnock et al., 2019)."

In addition, future model experiments utilising a fixed climate signal, as well sensitivity studies involving CH$_4$, are currently being undertaken by the Aerosol Chemistry Model Intercomparison Project (AerChemMIP). This will enable the quantification of the impact from changes in climate and CH$_4$ on future air pollutants, which will inform future studies on the importance of these processes.

6. "across East Asia the additional precursor emission reductions in ssp370-lowNTCF have made little difference to surface O3 concentrations predicted by the CMIP6 models, indicating that other factors are more important over this region (chemistry or climate change)." This result is critically important. So, aggressive mitigation of ozone precursors has no impact on the surface ozone concentrations in this region relative to a scenario with those precursors? What is the reason for surface ozone in East Asia to be independent of ozone precursor emission changes under this level of global change? Further explanation is needed. Are there climatic feedbacks from the precursors themselves that are offsetting the changes?

The apparent small change in surface O$_3$ for the ssp370-lowNTCF scenario over East Asia can be initially attributed to only having a three model ensemble of results available for this future scenario at the time of manuscript submission. One model (BCC-ESM1) shows a larger response of surface O$_3$ in 2050 in both the ssp370 (original Figure S14, now changed to S19) and ssp370-lowNTCF scenarios than the other two models, which had a disproportionate impact on the multi-model mean shown in Figure 11. Since submission of the original manuscript, surface O$_3$ concentrations have become available from an additional two CMIP6 models for ssp370-lowNTCF (MRI-ESM2-0 and UKESM1-0-LL) which have now been included in the analysis to provide additional information for the explanation of the different surface O$_3$ response over East Asia in the ssp370 and ssp370-lowNTCF scenarios.

A revised Figure S14 (shown below and now Figure S19) has been included in the manuscript along with a new Figure S20 (shown below) showing the change in surface O$_3$ in the ssp370-lowNTCF scenario from CMIP6 models. The surface O$_3$ change in the BCC-ESM1 model is larger in both of the scenarios compared to other CMIP6 models. The difference between the 2050 panels in both figures (Figure R1) shows that the more aggressive mitigation measures in ssp370-lowNTCF have reduced future increases in surface O$_3$ concentrations across most world regions, compared to the response in ssp370. The notable exception is across Eastern China, a part of the larger East Asian region defined in Figure S1, where surface O$_3$ concentrations increase in ssp370-lowNTCF consistently across all models compared to ssp370. The increase in surface O$_3$ in all models for ssp370-lowNTCF over Eastern China can be attributed to a small increase in NMVOC emissions (Fig. 2) and a large decrease in NOx emissions (from a high initial value), which reduces the NOx titration of O$_3$ over this area. The decrease in PM$_{2.5}$ concentrations over Eastern China (Figure R2) could also reduce the heterogeneous loss of radicals (e.g. N$_2$O$_5$, HO$_2$) to aerosols in ssp370-lowNTCF, compared to ssp370, and is another process that could be important in explaining the increase in surface O$_3$, but will need further investigation (Li et al., 2019). The increase in surface O$_3$ over Eastern China is responsible for the smaller benefits simulated in the ssp370-lowNTCF scenario over the larger East Asia region (where the averaging takes into account both increases and decreases across the region). Further sensitivity experiments will be required to allow for a full quantification of the impacts from changes in chemistry and climate across different models in the ssp370 and ssp370-lowNTCF over the East Asia region.

[Figure]

**Figure S19** – Annual mean surface O₃ concentrations and future response in ssp370 across 6 different CMIP6 models. Top row shows the 2005-2014 annual mean surface O₃ concentrations in each model from the historical simulations. Middle row shows the surface O₃ response in 2050, relative to 2005-2014 mean, in each model for ssp370. Bottom row shows the same as the middle but for 2100. No data is presented in 2100 for BCC-ESM1 as data for ssp370 only extended out to 2055.

[Figure]

**Figure S20** – Annual mean surface O₃ concentrations and future response in ssp370-lowNTCF across 5 different CMIP6 models. Top row shows the 2005-2014 annual mean surface O₃ concentrations in each model from the historical simulations. Bottom row shows the surface O₃ response in 2050, relative to 2005-2014 mean, in each model for ssp370-lowNTCF.

[Figure]

**Figure R1** – Difference in annual mean surface O₃ for 5 CMIP6 models between ssp370-lowNTCF and ssp370 in 2050.

[Figure]

**Figure R2** – Same as Fig. S1 but for surface PM₂.₅.

An amended version of Figure 11 has now been included in the manuscript (and shown below) using the additional available model data. This shows a regional reduction in surface $O_3$ concentrations across East Asia in the ssp370-lowNTCF scenario compared to the ssp370, highlighting the benefit, albeit small, from the additional mitigation measures to $O_3$ precursors.

[Figure]

The following changes to the manuscript text have been made to reflect the above discussion of the reasons behind the changes in surface $O_3$ across East Asia in ssp370-lowNTCF:

P21, line 454 sentence amended to:

"*However, across East Asia the additional precursor emission reductions in ssp370-lowNTCF have resulted in smaller benefits to surface $O_3$ concentrations being simulated by the CMIP6 models than in other regions (Figure S20), which is attributed to an increase in surface $O_3$ concentrations over Eastern China (a part of the larger East Asian region shown in Fig. S1). This increase in surface $O_3$ results from the slight increase in NMVOC emissions (Fig. 2) and a reduction in the NOx titration of $O_3$ due to the large decreases in NOx emissions in ssp370-lowNTCF. In addition, a reduction in the heterogeneous loss of radicals due to decreases in $PM_{2.5}$ concentrations in ssp370-lowNTCF could also lead to increased surface $O_3$ concentrations* (Li et al., 2019)."

7. "Discrepancies in the magnitude of change in these emissions due to climate and \*land-use change\*". Please specify in similar to Table 1 the models for which the natural emissions and atmospheric chemistry are actually dynamically coupled with the climate model's land surface scheme and vegetation cover / Plant Functional Types (that are dynamically changing in the simulations due to human land use change). Which models have the BVOC emissions actually coupled to the climate model's internal land surface scheme? If uncertainty in the changes to

natural emissions is an important conclusion of the paper, there needs to be a separate Table describing the representation of those emissions in each model.

We would like to thank the reviewer for this useful comment. As part of the revision we have included a new table in the supplementary material (Table S1 shown below) that provides information on the chemistry and aerosol configuration within each model used in this study. In addition, a new Figure S23 has been included to show the emissions of isoprene from each model (in a similar way to Figure S15, revised now to S22). In direct response to the reviewers comment, the CMIP6 models that have interactive chemistry and emission of BVOCs coupled to the model's land surface scheme and a dynamic vegetation model are UKESM1-0-LL, GISS-E2-1-G (isoprene only), BCC-ESM1 and CESM2-WACCM. The number of BVOCs emitted from vegetation and involved in atmospheric chemistry varies within each of the CMIP6 models, leading to discrepancies in the total BVOC emissions in Fig S15 (now S22). GISS-E2-1-G interactively emits only isoprene, with no inhibition to $CO_2$ concentrations, whereas CESM2-WACCM emits isoprene (with inhibition to $CO_2$ concentrations), monoterpenes and many other short and long chained hydrocarbons. Emissions of BVOCs in these models will depend on the future climate and how the distribution of different vegetation types changes in each CMIP6 model in response to the future scenarios. This could lead to important differences in both $O_3$ and secondary organic aerosol formation, particularly over regions with large natural sources of BVOC emissions. Further discussion of these comparisons are also made in the response to reviewer 2 but some small changes to manuscript are shown below.

Page 24 Lines 501-509 have been amended as follows:

"*Over South America and Southern Africa, particularly the tropical areas (Fig. S19), larger future changes in surface $O_3$, particularly by 2100, are predicted by GFDL-ESM4 and UKESM1 than by CESM2-WACCM. Over this region, biogenic emissions (particularly isoprene) are an important source of $O_3$ formation. Discrepancies in the future response of these BVOC emissions between models could be occurring due to the differing magnitudes of climate and land-use change and how they are coupled within individual CMIP6 models (Table S1), which could affect future surface $O_3$. Future changes in the total emissions of BVOCs and solely from isoprene obtained from five CMIP6 models (Figure S22 and S23) show that CESM2-WACCM has larger total BVOC emissions over the period 2005-2014 (due to the inclusion of more BVOCs), which then increase in the future ssp370 scenario, along with isoprene emissions, resulting in a smaller increase (and decreases over some parts of the region) in surface $O_3$. Whereas,  UKESM1-0-LL shows larger increases in $O_3$ and a reduction in BVOCs, mainly from isoprene (Fig. S23), over part of South America and tropical Africa. .*"

Page 28 Lines 586-590 have been amended as follows:

"*Over Southern Africa UKESM1-0-LL shows a reduction in future $PM_{2.5}$, in contrast to  other models,  due to a reduction in the BC, OA and dust aerosol components (Fig. S24, S26 and S27). UKESM1-0-LL exhibits particularly strong negative correlations for surface $PM_{2.5}$ when compared with temperature and precipitation. These relationships over Southern Africa are quite different to other CMIP6 models, which is also highlighted in the model evaluation over this region (Fig. 8) and indicates that climate change influences aerosol concentrations differently over this region in this model (Figure 16). In addition, there is a slight positive correlation of $PM_{2.5}$ with BVOC emissions in UKESM1-0-LL over Southern Africa. Future biogenic emissions (including monoterpenes) reduce here in ssp370 (Fig. S22), potentially due to land-use vegetation change as UKESM1-0-LL has dynamic vegetation coupled to BVOC emissions (Table S1). This could also reduce $PM_{2.5}$ concentrations over this*

*region because monoterpene emissions are the main precursor to SOA formation in UKESM1-0-LL (Mulcahy et al., 2019)."*

**Table S1** – Brief descriptions of the chemistry and aerosol set up within CMIP6 models used in this study

| CMIP6 Model | Horiz. Res. | Vert levels (top level) | Aerosol scheme | Aerosol Species | Natural Sources | Treatment of SOA | Chemistry Scheme | Chemistry reactions | BVOCs | Model Ref |
|---|---|---|---|---|---|---|---|---|---|---|
| BCC-ESM1 | 2.813° x 2.813° | L26 (2.914 hPa) | Mass-based aerosol scheme. Prescribed stratospheric aerosols. | $SO_4$, BC (hydrophilic and hydrophobic), OM (hydrophilic and hydrophobic, sea salt (4 size bins), dust (4 size bins). No nucleation or coagulation of aerosols | Prescribed DMS seawater concentrations with emissions dependent on wind speed. Online emissions of sea-salt and dust aerosols. NOx calculated from lightning. | Hydrophilic OC from anthropogenic emissions but also from natural sources calculated using a fixed yield, assumed to be equal to 10% of monoterpene emissions (from land surface model) | CAM-Chem (based on MOZART). Tropospheric only chemistry. | 66 gas-phase chemical species with 33 photolytic reactions and 135 kinetic reactions. | Online biogenic emissions from dynamically evolving vegetation computed in the land model BCC-AVIM2.0 following the algorithm of MEGANv2.1 which has a dependence on light and temperature but also inhibits isoprene emissions based on $CO_2$. | (Wu et al., 2020) |
| CESM2-WACCM | 0.9° x 1.25° | L70 ($6 \times 10^{-6}$ hPa) | MAM4 (modal scheme, simulating mass and number concentrations) with VBS-SOA | $SO_4$, BC, OM (both primary and secondary), sea salt, dust | Prescribed climatology of DMS seawater concentrations and emissions. Online emissions of sea-salt and dust aerosols. NOx calculated from lightning. Soil NOx and ocean CO, VOCs from POET | Explicit calculation of SOA using volatility basis set (VBS) where aromatic species, terpenes and isoprene are oxidised to produce a range of gas-phase SOA precursors with different volatilities. Formation of SOA linked to BVOCs emissions from interactive land surface scheme. | MOZART-TSMLT1 covering troposphere, stratosphere, mesosphere and lower thermosphere | 231 gas-phase species, 150 photolytic reactions, 403 kinetic reactions and 30 heterogeneous reactions involving ClOx, BrOx, NOx-HOx-Ox, CO, $CH_4$ and NMVOCs. | Online biogenic emissions (isoprene, monoterpenes, acetone, methanol, and other short and long-chained hydrocarbons) from dynamically evolving vegetation computed in the Community Land Model (CLM) using the MEGAN2.1 algorithm, which has dependence on light and temperature but also inhibits isoprene emissions based on $CO_2$. | (Gettelman et al., 2019; Tilmes et al., 2019; Emmons et al., 2020) |
| CNRM-ESM2-1 | 1.4° x 1.4° | L91 (80km) | TACTIC_v2. Tropospheric aerosols. Mass | $SO_4$, BC (hydrophilic and hydrophobic), OM (hydrophilic | Prescribed DMS seawater concentrations. Online | Prescribed SOA from monthly inventory | No representation of lower tropospheric | N/A | N/A | (Michou et al., 2019; Séférian et al., 2019) |

| CMIP6 Model | Horiz. Res. | Vert levels (top level) | Aerosol scheme | Aerosol Species | Natural Sources | Treatment of SOA | Chemistry Scheme | Chemistry reactions | BVOCs | Model Ref |
|---|---|---|---|---|---|---|---|---|---|---|
| | | | based aerosol scheme. | and hydrophobic), sea salt (3 size bins), dust (3 size bins) | emissions of sea-salt and dust aerosols | | chemistry so not considered here. | | | |
| GFDL-ESM4 | cubed-sphere (c96) grid, with ~100 km native resolution, regridded to 1.0° x 1.25° | L49 (0.01 hPa) | Bulk mass-based scheme. 5 size bins are used for sea salt and dust. | $NH_4$, $SO_4$, $NO_3$, $NH_4$, BC, OM, sea salt, dust | DMS and sea salt emissions calculated online as a function of wind speed (and a prescribed DMS seawater climatology). Dust emissions coupled to interactive vegetation. Lightning NOx calculated online as a function of convection. Natural emissions of NOx, CO, NMVOCs, $H_2$, and $NH_3$ from POET. $NH_3$ from seabird colonies. Two-way exchange of $NH_3$ with ocean. | SOA formed simulated using an anthropogenic source from oxidation of $C_4H_{10}$ tracer and a tracer representing BVOC emissions from vegetation | Interactive stratosphere-troposphere | 43 photolysis reactions, 190 gas-phase kinetic reationcs and 15 heterogeneous recations. NOx-HOx-Ox- chemical cycles and CO, $CH_4$ and NMVOC oxidation reactions | Online emissions of BVOCs (isoprene and monoterpenes) calculated from a prescribed vegetation cover using MEGAN2.1 algorithm, which has dependence on light and temperature but also inhibits isoprene emissions based on $CO_2$. | (Horowitz et al., 2019; Dunne et al., 2020) |
| GISS-E2-1-G | 2° x 2.5° | L40 (0.1 hPa) | OMA (one moment aerosol scheme – mass based) | $SO_4$, $NO_3$, $NH_4$, BC, OM treated as externally mixed with prescribed and constant size distribution. Sea salt has two size classes. Sectional scheme for dust with 5 size bins that can be coated with $SO_4$ and $NO_3$ to increase solubility. | Sea salt, DMS, isoprene and dust emission fluxes are calculated interactively. Online NOx calculated from lightning. Soil NOx, ocean CO, VOCs from GEIA. $NH_3$ from oceans. $SO_2$ from volcanoes as in AeroCom. | Two-product model approximation to represent SOA formation from the oxidation of biogenic VOCs, including NOx dependent chemistry yields. | Coupled troposphere-stratosphere chemistry scheme. Modified Carbon Bond Mechanism 4 (CBM-4) chemical mechanism | inorganic chemistry of Ox, NOx, HOx, CO, and organic chemistry of $CH_4$ and lumped higher hydrocarbons (only isoprene and terpenes are explicitly taken into account), along with Cl and Br stratospheric chemistry and heterogenous reactions on PSCs and $SO_4$ aerosols. | Emissions of isoprene from dynamically evolving vegetation are calculated interactively using the algorithm of Guenther et al., (1995), which has dependence on light and temperature. Terpene emissions are prescribed. | (Bauer et al., 2020) |
| HadGEM3-GC31-LL | 1.25° x 1.875° | L85 (85km) | GLOMAP-Mode. (Modal scheme, | $SO_4$, BC, OM, sea salt in 5 log- | Prescribed climatologies of DMS | Fixed yield of SOA of 26% calculated | Simplified sulphur | Oxidation for $SO_4$ and simplified | N/A | (Mulcahy et al., 2020) |

| CMIP6 Model | Horiz. Res. | Vert levels (top level) | Aerosol scheme | Aerosol Species | Natural Sources | Treatment of SOA | Chemistry Scheme | Chemistry reactions | BVOCs | Model Ref |
|---|---|---|---|---|---|---|---|---|---|---|
| | | | | normal modes and dust in 6 bins | seawater concentrations and BVOC emissions. No marine source of primary organics. Online emissions of sea-salt and dust aerosols | from gas-phase oxidation reactions involving prescribed land-based monoterpene sources | chemistry for use with aerosol scheme | oxidation scheme (monoterpenes) for SOA | | |
| MIROC6-ES2L | 2.813° x 2.813° | L40 (3.0 hPa) | SPRINTAS. | SO$_4$, BC, OM, sea salt and dust in log-normal size distributions. External mixing assumed for SO$_4$, sea salt and dust aerosols. | Online emissions of DMS, sea-salt and dust aerosols. Primary marine organic aerosol emissions coupled to ocean biogeochemistry. | Prescribed emissions of isoprene and terpenes from GEIA used to convert to secondary organic carbon. | Simplified chemistry for use with aerosol scheme | Oxidation for SO$_4$ and simplified oxidation scheme (isoprene and monoterpenes) for SOA | Prescribed emissions of isoprene and terpenes from GEIA. | (Takemura, 2012; Hajima et al., 2020) |
| MPI-ESM1.2-HAM | 1.875° x 1.875° | L47 (0.01 hPa) | HAM2.3 (Modal scheme, mass and number) | SO4, BC, OM, sea salt, dust in 7 log-normal modes | Interactive online emissions of DMS (using prescribed sea water concentrations), sea-salt and dust aerosols dependent on meteorology. | 15% of natural terpene emissions at the surface (prescribed) form SOA. SOA have identical properties to primary organic aerosols | Simplified sulphur chemistry. Other fields prescribed. | Reactions involving SO$_2$, DMS and SO$_4$, including aqueous phase. | N/A | (Tegen et al., 2019) |
| MRI-ESM2-0 | MRI-AGCM3.5: 1.125° x 1.125°, MASINGAR mk-2r4c: 1.875° x 1.875°, MRI-CCM2.1: 2.813° x 2.813° | L80 (0.01 hPa) | MASINGAR mk-2r4c | Mass-based scheme with externally mixed size distributions. SO4 (three categories), BC (hydrophilic and hydrophobic), OM (hydrophilic and hydrophobic), sea salt (10 size bins), dust (10 size bins). | Interactive online emissions of DMS (using prescribed Climatological DMS sea water concentrations), sea-salt, and dust aerosols dependent on meteorology. Online NOx calculated from lightning. Climatological soil NOx and ocean CO, VOCs emissions. | No explicit calculation: 14% of prescribed monoterpene and 1.68 % of isoprene emissions are assumed to form SOA. | Chemistry Climate Model version 2.1 (MRI-CCM2.1) covering troposphere, stratosphere, and mesosphere | 90 chemical species and 259 chemical reactions (184 gas-phase reactions, 59 photolysis reactions, and 16 heterogeneous reactions) involving HOx-NOx-CH$_4$-CO cycles and NMVOC oxidation reactions, and halogen chemistry (Cl and Br) | Climatological BVOCs emissions | (Deushi and Shibata, 2011; Yukimoto et al., 2019) |
| NorESM2-LM | 1.9° x 2.5° | L32 (3.64 hPa) | OsloAero6 | SO4, BC, OM, sea salt, dust. (log-normal modes) | Interactive emissions for sea-salt, biogenic primary OM (including | Fixed SOA formation yields of 15% and 5% from | Simplified chemistry for use in aerosol | Oxidation for SO$_4$ and simplified oxidation scheme | Online biogenic emissions from dynamically evolving | (Kirkevåg et al., 2018; |

| CMIP6 Model | Horiz. Res. | Vert levels (top level) | Aerosol scheme | Aerosol Species | Natural Sources | Treatment of SOA | Chemistry Scheme | Chemistry reactions | BVOCs | Model Ref |
|---|---|---|---|---|---|---|---|---|---|---|
| | | | | | MSA) and DMS over oceans, and interactive mineral dust and BVOC over land | oxidation of monoterpenes and isoprene | scheme. Other fields prescribed. | (isoprene and monoterpenes) for SOA | vegetation computed in the Community Land Model (CLM) using the MEGAN2.1 algorithm, which has dependence on light and temperature but also inhibits isoprene emissions based on $CO_2$. | Seland et al., 2020) |
| UKESM1-0-LL | 1.25° x 1.875° | L85 (85km) | GLOMAP-Mode. (Modal scheme, mass and number). Mass based bin scheme used for dust. | $SO_4$, BC, OM, sea salt in 5 log-normal modes and dust in 6 bins | Dynamic vegetation and interactive ocean biogeochemistry used for online emissions of DMS, sea-salt and dust aerosols, as well as emissions of primary marine organics and biogenic organic compounds. Online NOx calculated from lightning, soil NOx and ocean CO, VOCs from POET | Fixed SOA yield of 26% from gas-phase oxidation reactions involving interactive land-based monoterpene sources. | UKCA coupled stratosphere-troposphere. Interactive photolysis | 84 chemical tracers. Simulates chemical cycles of Ox, HOx and NOx, as well as oxidation reactions of CO, $CH_4$ and NMVOCs. In addition, heterogeneous processes, Cl and Br chemistry are included. | Dynamic vegetation and land surface model used to calculate interactive emissions of Isoprene and monoterpenes using light and temperature, but isoprene emissions are inhibited based on $CO_2$. Isoprene emissions coupled to chemistry and affect tropospheric $O_3$ and methane lifetime. Monoterpenes only affect SOA. | (Archibald et al., 2020; Mulcahy et al., 2020) |

Minor comments

I find Fig. 2 challenging to look at and wonder about for other readers. I appreciate it is difficult to show this Fig. 1 type information across multiple regions.

I would like to thank the reviewer for the comment on Figure 2., which has been reproduced in a different way to try and make it easy to view. The amended figure is shown below and have been used to replace the original Figure 2 in the manuscript.

[Figure]

Is it necessary to have both Fig 6 and Fig 8 i.e. for the 2000-2010 and 2005 and 2014 periods? Could one of the plots go into SI?

We thank the author for the comments but whilst it appears that Figure 6 and 8 are showing similar results, there are key differences which means it is important to include both within the main text. Figure 6 shows a comparison of model vs observations at ground based monitoring locations, which are from specific spatial points within each region. The results for MERRA on Figure 6 are also shown at only these locations for the same time period (2000-2010) in order to directly compare the MERRA product with the ground based observations and CMIP6 models. This provides additional information for the evaluation of model biases (see response to comment 2 above). In figure 8 the regional means are calculated from MERRA based on all of the grid points within a particular region. The regional meaning therefore contains many more data points (see parenthesis on Figure 8) than is possible in Figure 6, which allows for improved statistics by using the reanalysis product. The comparison of Figure 6 and 8 therefore provides additional inter-comparison between the CMIP6 models, MERRA reanalysis product and ground based observations and we feel that it warrants a separate inclusion within the main text.

"Large regional historical changes are simulated for both pollutants, across East and South Asia, with an increase of up to 40 ppb for O3 and 12 µg m-3 for PM2.5." and similar sentences in abstract. Need to include the temporal averaging associated with those values in abstract (annual).

The following sentences have been changed within the abstract to include reference to the temporal averaging period:

"*Large regional historical changes are simulated for both pollutants, across East and South Asia, with an annual mean increase of up to 40 ppb for $O_3$ and 12 µg $m^{-3}$ for $PM_{2.5}$. In future scenarios containing strong air quality and climate mitigation measures (ssp126), annual mean concentrations of air pollutants are substantially reduced across all regions by up to 15 ppb for $O_3$ and 12 µg $m^{-3}$ for $PM_{2.5}$. However, for scenarios that encompass weak action on mitigating climate and reducing air pollutant emissions (ssp370), annual mean increases of both surface $O_3$ (up 10 ppb) and $PM_{2.5}$ (up to 8 µg $m^{-3}$) are simulated across most regions.*"

"Near Term Climate Forcers (NTCFs)." IPCC AR6 uses "Short-lived Climate Forcers (SLCFs)".

Changed all references to Near Term Climate Forcers in the manuscript to Short-lived Climate Forcers (SLFCs) to be consistent with IPCC AR6.

"Initial assessments have been made of future changes to air pollutants in the SSPs using simplified models." Need to add references here.

The sentence has been changed to the following to include additional references:

"*Initial assessments have been made of future changes to air pollutants in the SSPs using simplified models (Reis et al., 2018; Turnock et al., 2018, 2019)*"

"A particular climate mitigation target, in terms of an anthropogenic radiative forcing by 2100, is included on top of each SSP" What does "on top of" mean exactly?

The sentence has been amended as follows to improve clarity on this point:

"*A particular climate mitigation target, in terms of an anthropogenic radiative forcing by 2100, and the range of emission mitigation measures associated with achieving it are included in addition to the existing policy measures within each baseline SSP scenario.*"

"However, scenarios with large increases in global CH4 abundances, a large climate change signal and limited control of precursor emissions fail to restrict regional increases in surface O3, leading to poor future air quality and potential human health impacts (Silva et al., 2017)." Is this statement redundant/obvious? Where is the new science?

Thank you to reviewer for the comment on this sentence. The sentence has been rewritten to make it more relevant to differences in the new scenarios that have been used in CMIP6.

"However, scenarios with large climate signals (ssp370 and ssp585) but different post 2050 controls on $O_3$ precursors (most notably $CH_4$ and NOx), show different long-term changes in regional surface $O_3$ concentrations, which could have important consequences for impacts on human health."

"Whilst there is disagreements" sp. there are

Corrected mistake.

**Response to Referee 2**

This manuscript conducts an evaluation of surface PM2.5 and ozone with observations for the CMIP6 chemistry-climate models that participated in AerChemMIP. It also documents the simulated historical & future changes in annual mean ozone and PM2.5 in various regions around the globe. It's clear that an enormous amount of effort went in to preparing this manuscript. By detailing the performance of each individual model (10 for PM2.5; 5 for ozone) against the available observations, a major community service has been performed in the production of this detailed supplemental information.

The rather long paper documents the current status of O3 and PM2.5 in the latest versions of global chemistry-climate models. It does so, however, without much attempt to understand more deeply the inter-model differences, or the sources of agreement, beyond discussing qualitative links to the emission trajectories or referencing relationships identified in prior work. A stronger paper would be more cohesive throughout and communicate better the novelty of the work. Below I suggest ways to strengthen the paper in each of these two directions, followed by more detailed comments. I support the points made by the other reviewer and so try to avoid repeating those points here.

First, the model evaluation presented is not tied in a clear way to the past or future projections of the models. The evaluation focuses on monthly and seasonal data but then only annual mean concentrations are presented for the historical and future trends. It seems far more relevant to evaluate regional trends in annual mean concentrations where observations allow this, or to demonstrate some relationship between seasonal cycles and future changes across the models (and should one exist, this would be an exciting finding as it would open up the possibility of identifying a "best" model from the evaluation with observations). The evaluation shown in Figures 5 and 6 of the Mortier et al. paper or in Figure 4 of Griffiths et al. in this special issue seems more relevant, although the remote sites used in Griffiths et al. are not that relevant for the polluted regions examined in this study. One could tackle a similar type of evaluation for North America and Europe where there are at least two decades of long-term observations for ozone and PM2.5, and it should be particularly straightforward to do so with the gridded MERRA reanalysis product for PM2.5. An alternative angle could be to examine if the past or future trends are strongly seasonally dependent. If so, showing some of the seasonality in the projections would connect better to the seasonal evaluation included. If the authors choose to remove any of the current figures, they should be included in the supplemental material, as the general evaluation done here will certainly be of high value to the modeling community.

We thank the reviewer for this useful comment on trying to make the manuscript more quantitative and also to improve the connections between the model evaluation and historical/future projections. As the reviewer mentions an analysis of long-term changes in surface $O_3$ and aerosol properties has already been undertaken in other manuscripts within this special issue and was therefore considered outside of the scope of the current work (see response to point 3 of reviewer 1 for more details). Further work is ongoing to analyse long term surface $O_3$ changes from CMIP6 models at northern hemisphere continental observation locations. Therefore, we have made improvements throughout the manuscript to better connect the seasonal and annual mean aspects of the present day model evaluation with the historical and future simulations. Revised versions of Figures 3, 5 and 7 have been produced to include a comparison of the annual mean surface concentrations of $O_3$ and $PM_{2.5}$ with observations, in addition to the seasonal mean comparisons originally present. Figures S2 to S7 in the supplementary material showing individual CMIP6 model comparisons have also been updated to include annual mean comparisons. Numerous minor text changes to the manuscript have been made in section 3 to reflect the inclusion of the annual mean evaluation. An example of the revised Figure 3 for surface $O_3$ is shown below:

[Figure]

**Figure 3** – Multi-model (6 CMIP6 models) annual and seasonal mean surface $O_3$ concentrations in a) Annual mean, d) December January, February (DJF) and g) June, July, August (JJA) over the 2005-2014 period. The standard deviation in the multi-model mean in b) Annual mean, e) DJF and h) JJA. The difference between the multi-model mean and TOAR observations in c) Annual mean, f) DJF and i) JJA (colour bar saturates).

We have included simulated seasonal mean changes in air pollutants over the historical and future time periods on Figures in the revised manuscript and supplementary material to connect better with the present-day evaluation work. A new Figure S16 (shown below) has been included within the supplementary material showing the annual and seasonal mean change in surface $O_3$ and $PM_{2.5}$ between 1850 and 2014.

[Figure]

**Figure S16** – Annual and seasonal regional mean changes in surface O₃ and PM₂.₅ from pre-industrial (1850-1859 mean) to present day (2005-2014 mean) across 11 CMIP6 models. Individual circles represent each annual and seasonal mean changes from individual CMIP6 models, with the multi-model mean represented by the solid bar.

The following changes to the manuscript have been made in Section 4 to include the seasonal historical changes. The following new sentence has been included on page 17 line 382:

*"Globally and over most regions there has been a larger historical increase in surface O₃ in JJA than in DJF (Figure S16)."*

A new sentence has been included on page 18 line 388

*"Larger differences between CMIP6 models are shown in the DJF mean historical changes over northern hemisphere regions than occurred in JJA (Figure S16), reflecting the differences shown in the model evaluation (Fig. 4) and the strong seasonality of the changes."*

The sentence on page 18, line 390 has been amended to the following:

*"South Asia is the region with the largest diversity in simulated historical changes in surface O₃ of between 16 and 40 ppb, with a larger range in DJF (10-40 ppb) than in JJA (19-36 ppb)."*

The sentence on page 18, line 391 has been amended to the following:

*"Surface O₃ is simulated to have increased by between 10 to 30 ppb on an annual mean basis and by a larger amount in JJA (12 to 37 ppb) over the major northern anthropogenic source regions since 1850, driven mainly by the large increases in anthropogenic precursor emissions of CH₄, NOx, CO, and NMVOCs over this period."*

The sentence on page 19 line 408 has been amended to the following:

"*Larger regional increases in surface annual mean PM$_{2.5}$ of up to 12 μg m$^{-3}$ are simulated across South and East Asia, with changes in DJF (up to 21 μg m$^{-3}$) larger than those in JJA (up to 12 μg m$^{-3}$) (Fig. S16), reflecting the strong seasonality of PM$_{2.5}$ concentrations in these regions.*"

The sentence on Page 19, line 410-412 has been amended as follows:

"*The largest model diversity is also exhibited over the Asian regions with variations in the response between models of up to 50%,  with larger differences between models in DJF than JJA (Figure S16), reflecting the differences shown in the present day model evaluation (Fig. 6).*"

In addition, we have also included simulated seasonal mean changes in air pollutants over the future time periods on Figures 12 and 14 (now Fig. 15) in the revised manuscript to try to better link the future predictions with the present-day evaluation work. An example of a revised Figure 12 (shown below) has been included within the revised manuscript, now showing both the annual and seasonal mean change in surface O$_3$ in 2050 and 2095 in the ssp370 future scenario for four CMIP6 models. A similar revised Figure has also been included within the manuscript for future surface PM$_{2.5}$ changes in ssp370.

[Figure]

**Figure 12** – Future global and regional changes in the decadal annual and seasonal mean surface O$_3$, relative to the 2005-2014 mean, for the ssp370 pathway used in CMIP6. Each black circle represents the decadal annual mean response for an individual model in a) 2045-2055 and b) 2090-2100, with the coloured bars showing the standard deviation across the decadal annual mean. The DJF and JJA seasonal mean response averaged over the relevant 10 year period is shown by squares and triangles respectively. The multi-model regional mean over the

period 2005- 2014 is given towards the left of each panel. The response from the HTAP_param in each time period is shown by the separate gold circle.

The following changes to the manuscript have been made in Section 5 to include mention to the seasonal future changes in air pollutants.

Page 23 lines 491-492 have been amended to the following:

"*Over the North Pole region all models show surface $O_3$ increases that are larger than the HTAP_param, with a larger increase in DJF than JJA.*"

A new Figure S21 showing the future DJF surface $O_3$ changes in ssp370 has been included in the supplementary material, as well as a new sentence on Page 24 line 495 and an amended sentence on line 496:

"*The lower annual mean response in UKESM1-0-LL and GFDL-ESM4 is driven by a reduction in DJF in these models (Fig. S21), which results in the DJF change in 2050 being lower than the 2005-2014 annual mean value (Fig. 12). The large increase in NOx emissions in ssp370 over South Asia (~80%) has resulted in areas of NOx titration, particularly in DJF, near the Indo-Gangetic plain in both UKESM1-0-LL and GFDL-ESM4, reducing surface $O_3$ concentrations (Fig. S19 and S21). This strong feature of NOx titration of $O_3$ in DJF is absent in both CESM2-WACCM and BCC-ESM1, resulting in larger $O_3$ production over South Asia.*"

The following new sentence is included on Page 24 line 502:

"*These changes over South America are larger in JJA in all models, with small seasonal differences over Southern Africa.*"

A new sentence is included on Page 24 line 512:

"*There are differences in simulated seasonal response across these regions, with all models showing a smaller increase in JJA than DJF across North America and Europe, whilst across East Asia there tends to a be a larger future surface $O_3$ increase in JJA than DJF.*"

The sentence on page 27, lines 566-567 has been amended as follows:

"*In a similar analysis to that for surface $O_3$, a more detailed comparison has been undertaken of four CMIP6 models predicting changes in annual and seasonal surface $PM_{2.5}$ in 2050 and 2095 under ssp370 (Figure 14).*"

The sentence on page 27, line 568-569 has been amended to:

"*Small reductions in annual mean surface $PM_{2.5}$ concentrations (<2 µg m$^{-3}$) are simulated consistently by all CMIP6 models across North America and Europe in ssp370,  with larger reductions simulated in DJF than JJA.*"

A new sentence has been included on page 27, line 571.

"*Across South Asia, all models simulate a larger increase in DJF mean surface $PM_{2.5}$ concentrations, of up to 18 µg m$^{-3}$ by 2050, than occurs in JJA, and reflects the seasonality shown in the model evaluation.*"

The sentence on page 28, line 576-577 has been amended to:

*"Small regional annual mean increases are predicted in 2050 due to PM$_{2.5}$ increases in JJA from all models apart from GFDL-ESM4. A larger reduction in the SO$_4$ component is simulated over East Asia by GFDL-ESM4 than in other models (Fig S25), resulting in an overall decrease in PM$_{2.5}$. In 2095 most models simulate a reduction in PM$_{2.5}$ concentrations in both seasons across East Asia, apart from CESM2-WACCM due to the increase in JJA."*

The sentence on page 28, line 591-594 has been amended to:

*"The decadal annual and seasonal mean PM$_{2.5}$ response is variable across individual CMIP6 models over regions close to natural sources of particulate matter (North Africa, Central Asia and Pacific, Australia and New Zealand). Over these regions there is a large range in both the sign and magnitude of the annual and seasonal PM$_{2.5}$ response, which can be mainly attributed to the dust fraction (Fig. S26) and the fact that this aerosol source has a large inter-annual variability in its emission strength."*

The sentence on page 30, line 641-642 has been amended to:

*"Across the historical period (1850-2014), the CMIP6 models simulated a global annual increase in surface O$_3$ of between 7 and 14 ppb, with a larger increase in JJA than DJF."*

The sentence on page 30, line 646-648 has been amended to:

*"Small global increases in surface PM$_{2.5}$ are simulated over the historical period by CMIP6 models, with larger regional changes of up to 12 µg m$^{-3}$ on annual mean basis and up to 18 µg m$^{-3}$ in DJF across East and South Asia."*

Second, the authors could better demonstrate the new contributions here, perhaps by looking a bit more closely at some aspect of the inter-model differences rather than ending with qualitative and in some cases speculative statements. For example, are there clear relationships between the inter-model spread in the global or regional temperature or precipitation changes and the air pollution changes projected over time?

Could previously identified general conclusions regarding relationships between global ozone, NOx and methane (see Figure 6 of Stevenson et al. 2006, Figure 13 of Young et al., 2013) be extended to surface ozone, and regionally? Can any conclusions be made as to whether future changes in particulate matter depend most on a particular component? There is a lot of useful information in the supplement regarding aerosol components and temperature changes that could be connected more closely to the changes reported in the main text. I find Figures 12 and 14 particularly interesting and the results presented there would be even more useful if they were connected more directly to changes in regional or global temperature, precipitation, humidity, air pollutant emissions, precursor surface concentrations, or whichever quantities are available across the set of models.

We thank the reviewer for this useful comment on trying to connect the changes in air pollutants better with other variables such as aerosol components, temperature, precipitation and emissions. We have conducted additional analysis by comparing regional future changes in air pollutants from individual models in ssp370 over the period 2015 to 2100 with selected variables. However, there are additional experiments being performed within AerChemMIP that will enable further quantification of the emission and climate change effect on air quality. A summary figure for both O$_3$ and PM$_{2.5}$ showing the correlation coefficients for these comparisons has now been included as a new Figure 13 and 16 within the manuscript (and shown below). Changes to the manuscript listed below have been made to reflect this new analysis.

[revised manuscript text omitted]

Detailed comments

One of the more interesting aspects of the paper is the comparison with the parameterisation based on HTAP models to separately attribute changes to emissions versus the combined emissions and climate changes simulated by the AerChemMIP models. However, it would help to have a better summary of how the parameterisation was developed and applied. Is it one parameterisation, or an ensemble of parameterisations that were developed separately for each model? Is there any overlap in the models used in developing the parameterisation and the AerChemMIP models? If so, can that subset of models be analyzed to attribute with greater confidence the role of climate change? Would this study support future work to extend this parameterisation to include the effects of temperature, humidity, or some other changes in climate variables?

The $O_3$ parameterisation is built upon models and emission perturbation experiments contributing to phase 1 and 2 of the Hemispheric Transport of Air Pollutants (HTAP) project. The models used to construct the $O_3$ parameterisation are independent of those used in CMIP6 and in the analysis presented in this manuscript. The parameterisation is based solely on emission perturbation experiments and does not account for any changes in $O_3$ due to climate or meteorology. Therefore, comparison of the results from the parameterisation with CMIP6 models provides an indication of the impact on surface $O_3$ from non-emission driven changes. Further development of the parameterisation is planned in the future to include some representation of the impact of climate change on surface $O_3$.

Based on the reviewers comment we have included more details on the development and application of the $O_3$ parameterisation in the manuscript. The following has been included on Page 6, line 206:

"*The HTAP_param was previously developed based upon the source-receptor relationships of $O_3$ derived from perturbation experiments of regional precursor emissions and global $CH_4$ abundances (Wild et al., 2012; Turnock et al., 2018). The HTAP_param applies the fractional change in global $CH_4$ abundance and regional emission precursors (NOx, CO and NMVOCs) for a particular scenario to the ozone response from each individual model used in the parameterisation. The total $O_3$ response is obtained by summing up the response from each of the individual models to all precursor changes across all source regions. The surface $O_3$ response previously calculated from the HTAP_param in both the historical and future CMIP6 scenarios is compared to that from the CMIP6 models (Turnock et al., 2019).*"

The referencing throughout the text seems to focus on more recent work rather than early papers that first identified important relationships. For example, the role of increasing water vapor in increasing ozone loss was first pointed out by Johnson et al., 1999 (text around line 65, and especially 450); the role of methane for surface ozone by Fiore et al. 2002 and Shindell et al. 2012 (text around line 65); the increase in ozone under climate change scenarios by Wu et al. 2009 and Weaver et al. 2009 (text around line 645).

Following the recommendations of the reviewer we have updated the text in the manuscript at the appropriate places to include reference to these papers.

Try to quantify wherever possible in the text, such as line 29 "consistent overestimate", line 31 "consistently underestimated", by how much? Is there any improvement in biases, or worsening, relative to prior studies? Line 40 "important differences", can anything be said as to which is most important or handled most realistically? Line 44-45 should include at least one example to support this statement.

We thank the reviewer for the suggestions and tried to make improvements throughout the text to provide more quantitative statements.

In response to the specific comments above Page 1, line 29-33 has been amended to:

"*CMIP6 models consistently overestimate observed surface $O_3$ concentrations across most regions and in most seasons by up to 16 ppb, with a large diversity in simulated values over northern hemisphere continental regions. Conversely, observed surface $PM_{2.5}$ concentrations are consistently underestimated in CMIP6 models by up to 10 µg $m^{-3}$, particularly for the northern hemisphere winter months, with the largest model diversity near natural emission source regions. The biases in CMIP6 models when compared to observations of $O_3$ and $PM_{2.5}$ are similar to those found in previous studies.*"

Page 1 Line 40 has been slightly amended to reflect that differences between models vary on a regional basis.

"*A comparison of simulated regional changes in both surface $O_3$ and $PM_{2.5}$ from individual CMIP6 models highlights important regional differences due to the simulated interaction of aerosols, chemistry, climate and natural emission sources within models.*"

Line 44 -45

"*Differences between individual models emphasises the importance of understanding how future Earth system feedbacks influence natural emission sources e.g. response of biogenic emissions under climate change.*"

Lines 113-114. Why do this for a future scenario rather than the historical period where there might be some opportunity to evaluate with observations?

The inter-model comparison of CMIP6 models for ssp370 was undertaken to explore the differences in their simulated response of air pollutants to future changes in emissions and climate. The model evaluation of simulated surface $O_3$ and $PM_{2.5}$ against observations in the present day (2004-2014) was conducted to benchmark each of the CMIP6 models, as well as identify biases and differences between CMIP6 models. The evaluation highlights particular discrepancies between CMIP6 models such as the higher present day concentrations of surface $O_3$ simulated by BCC-ESM1 and GISS-E2-1-G and the large seasonal cycle in surface $O_3$ simulated by UKESM1. In addition, higher concentrations of surface $PM_{2.5}$ are simulated by CESM2-WACCM and UKESM1 over Asia, whereas lower values are simulated by MIROC-ES2L over remote regions. We have made amendments to the text in the model evaluation section of the manuscript to try and bring out some of the inter-model differences in addition to biases against observations.

Figure 2 is difficult to digest. Why does this need to be in the main text? This is an example where more could be gleaned from the analysis if these changes in emissions could be shown to be related to the projected changes in ozone and/or PM2.5, perhaps through scatterplots.

We included Figure 2 to highlight the regional disparity in emission trajectories of air pollutants compared to the global changes presented in Figure 1. In addition, we wanted to highlight the importance of different short-term or long-term trajectories in future scenario e.g. increases in NOx emissions across East Asia in ssp370 by 2050 but then reductions out to 2100. Figure 2 has been revised based on the comments from reviewer 1 to make it easier to understand (see response to reviewer 1 above). We have also made comparisons of changes in air pollutants to emissions in the future ssp370 scenario (see above) as suggested in the initial comments by reviewer 2.

Line 271. This can be checked and stated more confidently by examining NO2+O3 rather than just O3.

The sensitivity of $O_3$ formation to NOx concentrations in each individual CMIP6 model is discussed further in the response to point 4 of reviewer 1, which highlights that UKESM1 has some of the largest regional NOx concentrations and lower surface $O_3$ concentrations. Page 9 , line 271 has been amended to include reference to the new figure. In addition, comparisons of $O_3$ and NOx concentrations are made for each model and presented in response to the initial comments by reviewer 2.

Lines 444-445 is not new as this was a major result from CMIP5 era RCP8.5. Some of that work probably deserves a citation, such as Gao et al. 2013.

This section has been amended to include references as per the response to point 5 of reviewer 1 above.

The biases in Figure 3 are very hard to read. It should be stated if the color bar saturates.

The colour bar on Figure 3 does saturate, which has now been stated in the figure caption. Figure 3 has also been amended to try and make the biases clearer, along with the inclusion of the annual comparisons in response to an earlier point by reviewer 2.

Lines 494-500. These seemingly different responses may occur because of different responses in winter versus summer across the models being mixed together in the annual mean.

The reviewer is correct in that this response in amplified on a seasonal mean basis. This section has been amended as stated in the initial response to reviewer 2 above.

Lines 503-514. Can these points about sources of inter-model differences be illustrated and based on evidence rather than surmised? Same goes for lines 580-590 & 600-602, where it might be worth moving some of the supplemental information into the main text to support more strongly these points.

Two new figures have been included in the manuscript to show correlations between future changes in air pollutants and different variables. The text of the manuscript has been edited as shown in the initial response to reviewer 2 to reflect the additional information on the reasons between differences in models.

Lines 648-650 should be supported with observations for this conclusion to be made here.

The following amendment to the text has been made to reference other studies that observe the same temporal changes in $PM_{2.5}$ concentrations.

"CMIP6 models simulate the peak in $PM_{2.5}$ concentrations in the 1980s across Europe and North America, prior to the simulating the observed decline in concentrations to present day (Leibensperger et al., 2012; Tørseth et al., 2012; Turnock et al., 2015),  attributed to the implementation of air pollutant emission controls over these regions."

Stronger evidence should also be included to support conclusions on lines 665-666 & 677-678.

Further evidence has been provided as to the reasons for the differences between CMIP6 models in the initial response to reviewer 2, along with changes to the text of the manuscript. We have slightly amended the text in the conclusion to reflect these changes.

Page 31 lines 665-66 have been amended to the following:

"*Disagreements in the prediction of future changes to regional surface PM$_{2.5}$ concentrations between individual CMIP6 models can  be attributed to differences in the complexity of the aerosol schemes implemented within models, in particular the formation mechanisms of organic aerosols and emission of BOVCs over certain regions  along with the strength of the climate change signal (temperature and precipitation)  simulated by models and  the impact this has on natural aerosol emissions via Earth system couplings *"

Page 31 lines 677-678 have been amended to the following:

"*Important differences between individual CMIP6 models have been identified in terms of how they  simulate air pollutants from the interaction of chemistry (O$_3$ and NOx), climate (temperature and precipitation) and natural precursor emissions (BVOCs) in the future.*"

[revised manuscript text omitted]

---

## Author Response (AR2)

**Author's response to Editor review on "Historical and Future changes in Air pollutants from CMIP6 Models"**

Steven T. Turnock et al.
Correspondence to: Steven T. Turnock
(steven.turnock@metoffice.gov.uk)

We would like to thank the editor and reviewer for their additional comments on the manuscript. Below we have provided a brief response to each comment in turn and made alterations to the manuscript where appropriate (shown enclosed in *"speech marks and italic font"* and any deletions from the manuscript shown with a strikethrough *""*). The referee comments are shown first in grey shading and the author's response is shown below in normal font.

The authors have thoroughly revised the paper and I find it suitable for publication. Again, I emphasize the enormous amount of work undertaken by the authors to document the current status of these models with respect to available observations and each other, and to address the concerns raised in the previous round of review. The new detailed table and figures, including those revised to increase readability from the first submission, will be extremely useful to future studies.

Below are a few minor comments the authors may wish to consider.

Line 43. Is this because emissions decrease in these regions even in SSP3-7?

Yes, across North America and Europe virtually all air pollutant emission precursors decrease even in the ssp370 scenario, which contains the largest emissions of near-term climate forcers.

Page 1 line 43 has been amended to:

*"However, for scenarios that encompass weak action on mitigating climate and reducing air pollutant emissions (ssp370), annual mean increases of both surface $O_3$ (up 10 ppb) and $PM_{2.5}$ (up to 8 µg m-3) are simulated across most regions, although, for regions like North America and Europe small reductions in $PM_{2.5}$ are simulated due to the regional reduction of precursor emissions in this scenario."*

Line 45. Prediction ⮕ projection (they are not initialized with the observed climate state).

Amended prediction to be projection here and throughout the manuscript.

Line 105. expenses ⮕ expense

Amended as suggested.

Line 304. A recent paper suggests dynamically represented dry deposition decreases simulated ozone over North America, East Asia, and elsewhere (Clifton et al. 2020: https://doi.org/10.1029/2020JD032398.) Of relevance to the oceanic regions, do any of these models include tropospheric halogen chemistry?

Included the above reference on Page 11 line 304.

No models used in this study include a representation of halogen chemistry in the troposphere but some do include halogen chemistry in stratosphere (Table S1).

Lines 325 and 364. Is there any obvious connection between models with higher organic aerosol and more detailed representations of SOA and/or precursor gas emissions?

From the analysis conducted here there is not a simple explanation of higher organic aerosol concentrations being produced from more complicated representations of SOA formation within models. CESM2-WACCM and UKESM1 both have high organic aerosol concentrations (see Fig S11) but these two models have very different levels of complexity in the SOA formation mechanisms (Table S1).

Lines 529-532. Is the point here that the difference in ozone increases is not simply following differences in NOx emissions in the 2 trajectories? Can an example be given as to what might explain this finding? Could there be a role for different biogenic emission responses?

The slightly slower increase in surface $O_3$ over South in ssp585 than ssp370 does corresponds to the different temporal changes in NOx emissions over this region in the two scenarios. The increase in NOx emissions in ssp370 (70%) is larger over South Asia by 2050, continuing to increase out to 2100, whereas NOx emissions in ssp585 peak earlier (by 2040) and continually decline afterwards out to 2100 (Fig. 2). This appears to be the main driver of the different response in surface $O_3$ between the two scenarios over South Asia. However, another key difference is that there are 6 CMIP6 models contributing surface $O_3$ concentrations for ssp370 whereas, only 4 models have data available for ssp585. The absence of the surface $O_3$ response from CESM2-WACCM and BCC-ESM1 in ssp585 could affect the multi-mean produced for this scenario, when compared to ssp370.

Page 21 lines 529-532 has been amended as follows to reflect the above:

"*Surface $O_3$ shows a slightly slower increase until  the mid 21$^{st}$ Century over South Asia in ssp585 than occurred in ssp370. This  can be attributed to a slightly different temporal evolution of NOx emissions over this region, in that they peak earlier (by 2040) and decline more rapidly in ssp585, when compared to the continual increase in NOx emissions in ssp370 (Fig. 2), which results in a different response of $O_3$ formation within CMIP6 models. In addition, there are more CMIP6 models with data available for ssp370 (6 models) than ssp585 (4 models) (Table 1), which could affect the multi-model mean response shown in Fig. 11. *"

Line 586. Are the models prescribing land use change separately? I thought this was a specified forcing? Are some models allowing natural emissions to adjust when that land use change is imposed while others are only updating physical climate variables?

There are a number of different ways that models account for natural emissions, which vary from those prescribing offline climatological BVOC emissions to models with BVOC emissions calculated from interactive land surface and vegetation schemes (see Table S1). The models with the latter set up will have BVOC emissions that respond differently to the simulated changes in climate and vegetation within their individual schemes and provide an additional area of diversity between models in their future response of BVOCs over certain areas.

Lines 614-616. A bit awkward; the factor of 2 across models doesn't have consequences for future air quality, but rather our ability to simulate it?

Page 24 line 614-618 has been amended as follows:

"*The predicted differences in models can be quite pronounced over regions like South Asia where changes in one model can be double that of another model, which could have important consequences for the ability of models to simulate future regional air quality*"

Line 802. Changes in NOx are cited here as a reason for discrepancy across the models. It's clear from Figure S17 that there are differences across the models despite using the same anthropogenic emissions. It may help to refer back to this figure here, or give an example of the range in NOx concentrations over a region. This range could reflect different natural emissions, chemical or deposition lifetimes, or different mixing and transport (e.g. convection). Is it possible to rule out any of these processes?

Reference on Page 31 line 802 has been made to Fig. S17, showing the different regional NOx concentrations, and Fig. S22, showing the difference BVOC emissions across models. Without further investigation of the CMIP6 models it is hard to rule out difference being due to any one process.

[revised manuscript text omitted]